# SafeFlowMatcher: Safe and Fast Planning using Flow Matching with Control Barrier Functions

**Jeongyong Yang**[*], **Seunghwan Jang**[* †], **SooJean Han**[†]
Korea Advanced Institute of Science and Technology (KAIST), Daejeon, Republic of Korea
`{seiryu2238, jsh991124, soojean}@kaist.ac.kr`
[*]Equal contribution. [†]Corresponding authors.

## Abstract

Generative planners based on flow matching (FM) produce high-quality paths in a single or a few ODE steps, but their sampling dynamics offer no formal safety guarantees and can yield incomplete paths near constraints. We present *SafeFlow-Matcher*, a planning framework that couples FM with control barrier functions (CBFs) to achieve *both* real-time efficiency and certified safety. SafeFlowMatcher uses a two-phase *prediction–correction* (PC) integrator: (i) a prediction phase integrates the learned FM once (or a few steps) to obtain a candidate path without intervention; (ii) a correction phase refines this path with a vanishing time-scaled vector field and a CBF-based quadratic program that minimally perturbs the vector field. We prove a barrier certificate for the resulting flow system, establishing forward invariance of a robust safe set and finite-time convergence to the safe set. In addition, by enforcing safety only on the executed path—rather than all intermediate latent paths—SafeFlowMatcher avoids distributional drift and mitigates local trap problems. Moreover, SafeFlowMatcher attains faster, smoother, and safer paths than diffusion- and FM-based baselines on maze navigation, locomotion, and robot manipulation tasks. Extensive ablations corroborate the contributions of the PC integrator and the barrier certificate. Code is available at the project page.

## 1 Introduction

Robotic path planning must simultaneously achieve real-time responsiveness and strong safety guarantees. Recently, generative models such as diffusion (Ho et al., 2020; Dhariwal & Nichol, 2021; Song et al., 2021b) and flow matching (FM) (Lipman et al., 2023) have gained attention for path planning, thanks to their expressive modeling of multi-modal action distributions (Carvalho et al., 2023; Braun et al., 2024) and low-latency inference (Qureshi et al., 2019; Liu et al., 2024) compared to classical sampling- and optimization-based planners. However, the sampling dynamics of these models are governed by implicitly learned rules and can produce paths that violate physical safety constraints, leading to task interruptions or collisions. Therefore, integrating *certified safety* into generative planning is essential for deployment in real-world robotic systems.

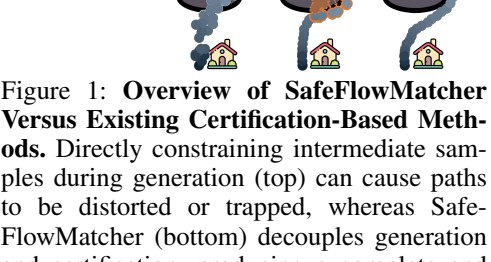

Figure 1: **Overview of SafeFlowMatcher Versus Existing Certification-Based Methods.** Directly constraining intermediate samples during generation (top) can cause paths to be distorted or trapped, whereas Safe-FlowMatcher (bottom) decouples generation and certification, producing a complete and certified-safe path.

Several approaches have attempted to enforce safety in generative planning. Safety-guidance methods regulate the sampling process through learned safety scores (often called guidance, e.g., classifier(-free) guidance(Dhariwal & Nichol, 2021; Ho & Salimans, 2021), or value/reward guid-

ance (Yang et al., 2024; Chen et al., 2024)), but their reliance on data-driven proxy prevents them from providing strong safety guarantees. More explicitly, certification-based methods incorporate functions such as Control Barrier Functions (CBFs) directly into the generative process (Xiao et al., 2025). Unlike guidance-based approaches, these methods can guarantee safety at deployment without requiring additional training. However, a key challenge in such certification-based methods is a *semantic misalignment*: certification concerns the executed physical path (its waypoints over the horizon), whereas interventions are often applied to intermediate latent states that are never executed. Constraining such latents is unnecessary for certification. As a result, repeated interventions distort the learned flow and often yield incomplete (locally trapped) paths. Finally, although diffusion samplers can be accelerated (Lu et al., 2022; Zhang & Chen, 2022; Liu et al., 2022), their SDE-based denoising requires many steps, making real-time planning expensive. In contrast, FM casts sampling as deterministic ODE integration, generating accurate paths in a *single* or a *few* steps.

To address these limitations, we propose *SafeFlowMatcher*, a planning framework that combines flow matching with CBFs, particularly for finite-time convergence CBFs, to achieve certified safety before the completion of generation, while maintaining the efficiency of FM. Our key idea is a *prediction–correction* (PC) integrator that decouples distributional drift from safety certification. In the prediction phase, we propagate the flow once (or a few steps) to obtain a candidate path without any safety intervention. In the correction phase, we refine this path by (i) compensating for integration error through a modified vector field, and (ii) enforcing safety through CBFs. Rather than constraining all intermediate samples from pure noise to the target during prediction, SafeFlowMatcher enforces safety only in the correction phase. This preserves the native FM dynamics and prevents distributional drift when generating the target path. Also, it avoids local traps caused by repeatedly pushing intermediate waypoints onto the barrier boundary and stalling near safety constraints. In summary, our main contributions are as follows:

- We introduce SafeFlowMatcher, a novel planning framework that integrates finite-time convergence CBF-based certification with flow matching to enforce hard safety constraints, while preserving the efficiency of flow matching.
- We propose a prediction–correction integrator that decouples path generation from certification: FM first generates paths without intervention, and then CBF-based corrections enforce finite-time convergence to the safe set while compensating for integration errors.
- We validate SafeFlowMatcher in maze navigation, locomotion, and robot manipulation tasks with extensive ablation studies, showing consistent improvements over both FM- and diffusion-based planners in efficiency, safety, and path quality.

## 2 RELATED WORK & PRELIMINARIES

### 2.1 FLOWMATCHER: FLOW MATCHING FOR PLANNING

FM has recently been proposed as a powerful alternative to diffusion, originally in the image generation domain (Lipman et al., 2023; Song et al., 2021b), and has shown promise for efficient path planning and robotic control (Ye & Gombolay, 2024; Zhang & Gienger, 2024; Chisari et al., 2024; Xing et al., 2025). Unlike diffusion, FM directly learns a time-varying vector field that maps noise to the target distribution via forward integration, making the sampling process efficient and flexible.

We adapt standard flow matching (FM) (Lipman et al., 2023) to the planning context. Let $H \in \mathbb{N}$ be the planning horizon and $\mathcal{H} \triangleq \{0, \ldots, H\}$. A path is a stacked vector $\boldsymbol{\tau} = (\boldsymbol{\tau}^0, \boldsymbol{\tau}^1, \ldots, \boldsymbol{\tau}^H) \in \mathcal{D}^{H+1} \subseteq \mathbb{R}^{d \times (H+1)}$, where each waypoint $\boldsymbol{\tau}^k \in \mathcal{D} \subseteq \mathbb{R}^d$ encodes the state at step $k$.

Let $v_t(\cdot; \theta) : \mathcal{D}^{H+1} \to \mathcal{D}^{H+1}$ be a time-dependent vector field. The flow $\psi : [0, 1] \times \mathcal{D}^{H+1} \to \mathcal{D}^{H+1}$ is defined as the solution of the ODE

$$\frac{d}{dt} \psi_t(\boldsymbol{\tau}) = v_t(\psi_t(\boldsymbol{\tau}); \theta), \qquad \psi_0(\boldsymbol{\tau}) = \boldsymbol{\tau}, \tag{1}$$

which transports a simple prior $p_0$ (e.g. $\mathcal{N}(0, I)$) to a target $p_1$. Following conditional flow matching (CFM), we train $v_t(\cdot; \theta)$ by regressing it to a conditional vector field that generates a fixed conditional probability path. We adopt the optimal transport (OT) path $p_t(\boldsymbol{\tau} \mid \boldsymbol{\tau}_1) = \mathcal{N}(\boldsymbol{\tau}; \mu_t(\boldsymbol{\tau}_1), \sigma_t^2 I)$, $\mu_t(\boldsymbol{\tau}_1) = t\,\boldsymbol{\tau}_1$, $\sigma_t = 1 - t$, whose generating *OT-conditional vector field* is

$$u_t(\boldsymbol{\tau} \mid \boldsymbol{\tau}_1) = \frac{\boldsymbol{\tau}_1 - \boldsymbol{\tau}}{1 - t}. \tag{2}$$

Let $q$ denote the data distribution over the target paths $\boldsymbol{\tau}_1$. Sampling $t \sim \text{Unif}[0, 1]$, $\boldsymbol{\tau}_0 \sim p_0$, $\boldsymbol{\tau}_1 \sim q$ and defining $\boldsymbol{\tau}_t \triangleq \psi_t(\boldsymbol{\tau}_0) = (1 - t)\boldsymbol{\tau}_0 + t\,\boldsymbol{\tau}_1$ (conditioned on $\boldsymbol{\tau}_1$), we have by (2) that $u_t(\boldsymbol{\tau}_t \mid \boldsymbol{\tau}_1) = \boldsymbol{\tau}_1 - \boldsymbol{\tau}_0$. Hence, we train $v_t(\cdot; \theta)$ with the CFM loss:

$$\mathcal{L}(\theta) = \mathbb{E}_{t,\, q(\boldsymbol{\tau}_1),\, p_0(\boldsymbol{\tau}_0)} \big\| v_t(\psi_t(\boldsymbol{\tau}_0); \theta) - (\boldsymbol{\tau}_1 - \boldsymbol{\tau}_0) \big\|_2^2. \tag{3}$$

Further details are in Lipman et al. (2023).

For numerical integration, we discretize $0 = t_0 < \cdots < t_T = 1$ with the sampling horizon $T \in \mathbb{N}$ (Collectively $\mathcal{T}(T) = \{t_0, \dots, t_T\}$) and define step sizes $\Delta t_i = t_{i+1} - t_i$. We define $T$-step integrator $\Psi_{0 \to 1}^{(T)} : \mathcal{D}^{H+1} \to \mathcal{D}^{H+1}$ (e.g., Euler integrator [1]) which integrates the flow matching dynamics from $\boldsymbol{\tau}_0$ to $\boldsymbol{\tau}_1$ as

$$\Psi_{0 \to 1}^{(T)}(\boldsymbol{\tau}_0) = \boldsymbol{\tau}_0 + \sum_{i=0}^{T-1} \Delta t_i\, v_{t_i}(\boldsymbol{\tau}_{t_i}; \theta). \tag{4}$$

## 2.2 Control Barrier Functions

*Safety filters* (Hsu et al., 2023; Wabersich et al., 2023) are a real-time intervention mechanism to ensure that an autonomous agent operates within some predefined safe sets, overriding its nominal behavior only when it is about to violate the sets. Various approaches exist for constructing safety filters, but among these, *control barrier functions (CBFs)* (Ames et al., 2019) are especially popular, as they provide a systematic way to guarantee forward invariance of safe sets by solving a real-time optimization problem at each control step. Additional recent works on CBFs, including non-convex safe sets and learning-based CBFs, are summarized in Appendix A.

Here, we review only the standard finite-time convergence CBF preliminaries that are necessary for the rest of this paper. To this end, we consider an arbitrary control-affine system

$$\dot{\mathbf{x}}_t = f(\mathbf{x}_t) + g(\mathbf{x}_t)\mathbf{u}_t, \tag{5}$$

where $\mathbf{x}_t \in \mathcal{D} \subset \mathbb{R}^d$, $\mathbf{u}_t \in \mathcal{U} \subset \mathbb{R}^d$, and $f : \mathbb{R}^d \to \mathbb{R}^d$ and $g : \mathbb{R}^d \to \mathbb{R}^{d \times d}$ are locally Lipschitz continuous.

Define the *safe set* $\mathcal{C}$ as the superlevel set of a continuously-differentiable ($C^1$) function $b : \mathcal{D} \to \mathbb{R}$,

$$\mathcal{C} \triangleq \{\mathbf{x}_t \in \mathcal{D} \mid b(\mathbf{x}_t) \geq 0\}. \tag{6}$$

System safety is often mathematically prescribed by ensuring that a system's state safely converges to the targeted safe set within finite time.

**Definition 1 (Finite-Time Convergence CBF)** *Given the system (5) and the safe set (6), $C^1$ function $b$ is called a finite-time convergence CBF if there exist parameters $\rho \in [0, 1)$ and $\epsilon > 0$ such that for all $\mathbf{x}_t \in \mathcal{D}$,*

$$\sup_{\mathbf{u}_t \in \mathcal{U}} \big[ L_f b(\mathbf{x}_t) + L_g b(\mathbf{x}_t)\mathbf{u}_t + \epsilon \cdot \text{sgn}(b(\mathbf{x}_t)) |b(\mathbf{x}_t)|^\rho \big] \geq 0, \tag{7}$$

*where $L_f b(\mathbf{x}_t) \triangleq \nabla b(\mathbf{x}_t)^\top f(\mathbf{x}_t)$ and $L_g b(\mathbf{x}_t) \triangleq \nabla b(\mathbf{x}_t)^\top g(\mathbf{x}_t)$ denote the Lie derivatives of $b$ along $f$ and $g$, respectively.*

**Lemma 1 (Forward Invariance of the Safe Set)** *Define CBF $b$ as in Definition 1, such that the initial state satisfies $b(\mathbf{x}_0) \geq 0$. Any Lipschitz continuous controller $\mathbf{u}_t$ that satisfies condition (7) ensures forward invariance of the safe set $\mathcal{C}$, i.e., $b(\mathbf{x}_t) \geq 0$ for all $t \geq 0$.*

Lemma 1 ensures that once the state first enters the safe set, it remains there thereafter. To select a control input that guarantees forward invariance of $\mathcal{C}$ while remaining as close as possible to a reference control input

$$\mathbf{u}_t^* = \arg\min_{\mathbf{u}_t \in \mathcal{U}} \|\mathbf{u}_t - \mathbf{u}_t^{\text{ref}}\|^2 \quad \text{subject to} \quad L_f b(\mathbf{x}_t) + L_g b(\mathbf{x}_t)\mathbf{u}_t + \epsilon \cdot \text{sgn}(b(\mathbf{x}_t)) |b(\mathbf{x}_t)|^\rho \geq 0. \tag{8}$$

This means the optimal solution $\mathbf{u}_t^*$ is the minimally modified control that guarantees the forward invariance of the safe set $\mathcal{C}$. Moreover, based on the finite-time stability theorem (Bhat & Bernstein, 2000), the finite-time convergence CBF can be used to ensure that states not only remain within the safe set but also reach it within finite-time (Li et al., 2018; Srinivasan et al., 2018).

---

[1] Alternatively, higher-order ODE solvers can be used.

## 3    SAFEFLOWMATCHER

Here, we present *SafeFlowMatcher*, a safe and fast planning framework that couples flow matching with certified safety in settings where neither the dynamics nor cost map are known. First, in Section 3.1, we introduce a two-phase *prediction–correction* (PC) integrator which decouples generation and certification. Next, in Section 3.2, we formalize safety for SafeFlowMatcher by employing control barrier functions (CBFs) and derive conditions that guarantee forward invariance and finite-time convergence to the safe set. The pseudocode of SafeFlowMatcher is in Algorithm 1, and full generation processes for two Maze environments are visualized in Appendix F.5.

We introduce a crucial problem in non-autoregressive planners, particularly for a generative-based planner. As shown in Figure 2, non-autoregressive planners may fail to generate a complete path after planning when using CBFs. Although the resulting path remains safe (does not exceed safety constraints), it may be unable to reach the goal because certain waypoints become *locally trapped* near the barrier boundaries and cannot escape within the finite sampling or integration time. We will show that SafeFlowMatcher can effectively resolve this issue using a PC integrator.

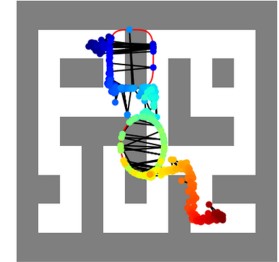

Figure 2: **Local trap.** Example of a local trap in maze environment.

**Definition 2 (Local Trap)**  *A local trap problem occurs during the planning process if there exists $k \in \mathcal{H}$ such that $\|\boldsymbol{\tau}_1^k - \boldsymbol{\tau}_1^{k-1}\| > \zeta$, where $\zeta > 0$ is a user-defined threshold depending on the planning environment.* [2]

---

**Algorithm 1** SafeFlowMatcher

---

    **Input:** learned velocity field $v_t(\cdot; \theta)$, prediction and correction horizon $T^p, T^c$, planning horizon $\mathcal{H}$, CBF parameters $(\epsilon, \rho)$, robustness parameter $\delta$, and scale constant $\alpha$
    **Output:** Safe path $\boldsymbol{\tau}_1^c$
    **Phase 1: Prediction**
1: Sample initial noise $\boldsymbol{\tau}_0^p \sim \mathcal{N}(0, I)$
2: Compute predicted path $\boldsymbol{\tau}_1^p \leftarrow \Psi_{0 \to 1}^{(T^p)}(\boldsymbol{\tau}_0^p)$ by (9)
    **Phase 2: Correction**
3: Initialize corrected path $\boldsymbol{\tau}_0^c \leftarrow \boldsymbol{\tau}_1^p$
4: **for** each correction step $t \in \mathcal{T}(T^c)$ **do**
5:     $\tilde{v}_t \leftarrow \alpha \, (1 - t) \, v_t(\boldsymbol{\tau}_t^c; \theta)$
6:     **for** each waypoint $k \in \mathcal{H}$ **do**
7:         Solve QP (17) to obtain $(\mathbf{u}_t^{k*}, r_t^{k*})$, using $\tilde{v}_t$
8:     **end for**
9:     Update velocity time-scaled flow dynamics (13) with $\mathbf{u}_t^* = \{\mathbf{u}_t^{0,*}, \ldots, \mathbf{u}_t^{H,*}\}$
10: **end for**
11: Return safe final path $\boldsymbol{\tau}_1^c$

---

### 3.1    PREDICTION–CORRECTION INTEGRATOR

SafeFlowMatcher divides the integration process into two phases: a prediction phase that generates an approximate path without considering safety, and a correction phase that refines the path by reducing integration error and adding safety constraints. Let $\boldsymbol{\tau}_t^\ell \in \mathcal{D}^{H+1} \subseteq \mathbb{R}^{d \times (H+1)}$ for $\ell \in \{p, c\}$ denote the paths in the prediction and correction phases, with waypoints $\boldsymbol{\tau}_t^{\ell,k} \in \mathcal{D} \subseteq \mathbb{R}^d$ for $k \in \mathcal{H}$. Additionally, we denote by $T = T^p + T^c$ the total sampling horizon, where $T^p$ and $T^c$ are the number of sampling (integration) steps allocated to the prediction and correction phases, respectively.

The **Prediction phase** aims to quickly approximate the target path starting from pure noise $\boldsymbol{\tau}_0^p \sim \mathcal{N}(0, I)$, without considering safety constraints. Starting from the noise, we run Euler integration to

---

    [2]The definition is slightly different from that of SafeDiffuser (Xiao et al., 2025) to capture a broader class of failure cases. See Appendix D for the details.

obtain the solution of the flow matching dynamics (1):

$$\tau_1^p = \Psi_{0\to1}^{(T^p)}(\tau_0^p) = \tau_1^\star + \varepsilon, \tag{9}$$

where $\tau_1^\star$ is the exact solution of the flow matching dynamics and $\varepsilon$ is the Euler integration (prediction) error. To balance computational efficiency and reliability, we select small $T^p$ (typically $T^p = 1$) that places $\tau_1^p$ sufficiently close to $\tau_1^\star$, making it a suitable initialization for the correction phase.

The **Correction phase** starts from the path in the prediction phase $\tau_0^c = \tau_1^p$, unlike $\tau_0^p$ in the prediction phase. In this phase, the path is refined by (i) reducing the discretization error $\varepsilon$ and (ii) enforcing safety constraints.

To achieve (i), we introduce the *vanishing time-scaled flow dynamics* (VTFD)

$$\frac{d\tau_t^c}{dt} = \alpha\,(1-t)\,v_t(\tau_t^c;\,\theta) \triangleq \tilde{v}_t(\tau_t^c;\theta), \tag{10}$$

where the factor $(1-t)$ gradually suppresses the vector field as $t \to 1$ with scaling constant $\alpha > 0$. Intuitively, this produces a contraction effect: the path is driven toward the target direction in the early correction steps, while the dynamics become increasingly stable near $t = 1$, preventing drift and allowing the prediction error to decay. This mechanism is formalized in Lemma 2 and Lemma 3.

**Lemma 2** *Assume the prediction error $\varepsilon \sim p_\varepsilon$ has a symmetric, zero-mean distribution (e.g., Gaussian) and that, in a neighborhood of $\varepsilon = 0$, the negative log-density $-\log p_\varepsilon$ is $C^2$ with a positive-definite Hessian $A \succ 0$ (i.e., locally strongly convex). In addition, assume the target log-density $\log p_1$ is $C^2$. Suppose the correction phase is initialized near the target $\tau_1^\star$:*

$$\tau_t^c = \tau_1^\star + (1-t)\,\varepsilon, \qquad \varepsilon = O(1). \tag{11}$$

*Then, $\mathbb{E}[\tau_1 \mid \tau_t^c] = \tau_1^\star + O(1-t)$.*

We empirically verify the validity of the symmetric zero-mean assumption on the prediction error $\varepsilon$ in Appendix B.1. (11) is a natural result under optimal transport, since OT path approaches $\tau_1$ as $t \to 1$. Lemma 2 ensures that the posterior expectation contracts toward the target.

**Lemma 3** *Under the assumptions of Lemma 2, let $\mathbf{e}_t \triangleq \tau_t^c - \tau_1^\star$. If the flow dynamics follow the vanishing time-scaled flow dynamics (10), then as $t \to 1$,*

$$\mathbf{e}_t = O((1-t)^2) + (\,\varepsilon + O(1)\,)e^{-\alpha t}. \tag{12}$$

Lemma 3 implies that VTFD reduces the prediction error of $\tau_1^c$. See the proofs of Lemma 2 and Lemma 3 in Appendix B.2.

## 3.2 Control Barrier Certificate for SafeFlowMatcher

To ensure the safety constraints hold during the correction phase, we introduce an additional perturbation to minimally intervene in the flow dynamics (10):

$$\frac{d\tau_t^c}{dt} = \tilde{v}_t(\tau_t^c;\theta) + \Delta\mathbf{u}_t, \tag{13}$$

where $\tilde{v}_t$ is VTFD defined in (10), and $\Delta\mathbf{u}_t = \{\Delta\mathbf{u}_t^0, \Delta\mathbf{u}_t^1, ..., \Delta\mathbf{u}_t^H\} \in \mathbb{R}^{d\times(H+1)}\,(\Delta\mathbf{u}_t^k \in \mathbb{R}^d)$ is a perturbation term that enforces safety constraints. Importantly, the safety constraint is applied in a *waypoint-wise* fashion: the CBF condition is enforced independently for each waypoint $\tau_t^{c,k}$ so that it remains within safe set $\mathcal{C}$. Thus, we can split the dynamics (13) into

$$\frac{d\tau_t^{c,k}}{dt} = \tilde{v}_t^k(\tau_t^c;\theta) + \Delta\mathbf{u}_t^k \triangleq \mathbf{u}_t^k, \tag{14}$$

where $\tilde{v}_t^k(\tau_t^c;\theta)$ denotes the $k$-th column of $\tilde{v}_t(\tau_t^c;\theta)$. For notational simplicity, we denote the right-hand side by $\mathbf{u}_t = \{\mathbf{u}_t^0, \mathbf{u}_t^1, ..., \mathbf{u}_t^H\} \in \mathbb{R}^{d\times(H+1)}(\mathbf{u}_t^k \in \mathbb{R}^d)$. [3] We now formalize the concept of safety in flow matching using finite-time flow invariance.

---

[3](14) is a control-affine system with drift $f(\tau_t^{c,k}) = \tilde{v}_t^k$ and input matrix $g = I$. Thus, at the waypoint level, the structure coincides with the standard control-affine system used in Section 2.2.

**Definition 3 (Finite-Time Flow Invariance)** *Let $b : \mathcal{D} \to \mathbb{R}$ be a $C^1$ function. The system (13) is finite-time flow invariant if there exists $t_f \in [0, 1]$ such that $b(\boldsymbol{\tau}_t^{c,k}) \geq 0$ for all $k \in \mathcal{H}$, $\forall t \geq t_f$.*

**Theorem 1 (Forward Invariance for SafeFlowMatcher)** *Let $b : \mathcal{D} \to \mathbb{R}$ be a $C^1$ function, and define the robust safe set $\mathcal{C}_\delta \triangleq \{\boldsymbol{\tau}^{c,k} \in \mathcal{D} \mid b(\boldsymbol{\tau}^{c,k}) \geq \delta\}$ for some $\delta > 0$. Suppose the system (13) is controlled by $\mathbf{u}_t$ satisfying the following barrier certificate for $0 < \rho < 1, \epsilon > 0$:*

$$\nabla b(\boldsymbol{\tau}_t^{c,k})^\top \mathbf{u}_t^k + \epsilon \cdot \mathrm{sgn}(b(\boldsymbol{\tau}_t^{c,k}) - \delta) |b(\boldsymbol{\tau}_t^{c,k}) - \delta|^\rho + w_t^k r_t^k \geq 0, \forall k \in \mathcal{H}, \forall t \in [0, 1]. \tag{15}$$

*Here, $w_t^k : [0, 1] \to \mathbb{R}_{\geq 0}$ is a monotonically decreasing function with $w_t^k = 0$ for all $t \in [t_w, 1]$ ($t_w \in [0, 1)$), and $r_t^k \geq 0$ is a slack variable. Then the flow matching (13) achieves finite-time flow invariance on $\mathcal{C}_\delta$.*

The weights $w_t^k$ serve as functions that relax the CBF constraint in the early refining phase, providing numerical stability by preventing infeasibility and reducing abrupt changes in the QP solution. Since $w_t^k$ vanishes for $t \geq t_w$, the relaxation term has no effect afterwards, ensuring that the final path satisfies certified safety.

**Proposition 1 (Finite Convergence Time for SafeFlowMatcher)** *Suppose Theorem 1 holds. Then for any initial waypoint $\boldsymbol{\tau}_{t_w}^{c,k} \in \mathcal{D} \setminus \mathcal{C}_\delta$, the waypoint $\boldsymbol{\tau}_t^{c,k}$ converges to the safe set $\mathcal{C}_\delta$ within finite time*

$$T \leq t_w + \frac{(\delta - b(\boldsymbol{\tau}_{t_w}^{c,k}))^{1-\rho}}{\epsilon(1-\rho)}, \tag{16}$$

*and remains in the set thereafter.*

Proposition 1 allows us to select parameters $\epsilon$ and $\rho$ to guarantee flow invariance on the robust safe set $\mathcal{C}_\delta$ before the time (16). The proofs of Theorem 1 and Proposition 1 are in Appendix C.

In order to enforce the invariance of the safe set $\mathcal{C}_\delta$ with minimum intervention during planning, we solve a quadratic program (QP) analogous to (8) at each sampling time $t$ and planning step $k$:

$$\mathbf{u}_t^{k*}, r_t^{k*} = \underset{\mathbf{u}_t^k, r_t^k}{\arg\min} \| \mathbf{u}_t^k - \tilde{v}_t^k(\boldsymbol{\tau}_t^c; \theta) \|^2 + r_t^{k^2} \quad \text{subject to} \quad (15), \tag{17}$$

where $\mathbf{u}_t^k$ and $\tilde{v}_t^k(\boldsymbol{\tau}_t^c; \theta)$ are defined in (14). Since the QP (17) is equivalent to a Euclidean projection problem with linear inequalities, closed-form solutions are available when it has at most two inequalities (Luenberger, 1997; Boyd & Vandenberghe, 2004). Moreover, the computational time can be reduced further by decreasing the correction horizon $T^c$ or balancing $(T^p, T^c)$, as discussed in Appendix F.1.

**Remark 1** *The PC integrator brings $\boldsymbol{\tau}_0^c$ closer to the barrier boundary after the prediction phase. By Proposition 1, this improved initialization reduces the required convergence time, allowing us a wider range of choices for $(\rho, \epsilon)$, and more stable control inputs. We empirically validate this in Appendix F.2.*

**Remark 2** *The relaxation term is mainly necessary in environments where the planner is prone to becoming locally stuck due to complex safety constraints. In particular, it is essential in Maze2D, where the safe set is highly non-convex, leading to frequent local traps. In contrast, in relatively open or convex environments such as locomotion or robot manipulation tasks in our experiments, the relaxation is typically unnecessary. In such cases, the relaxation term $w_t^k$ remains zero, and the slack variable $r_t^k$ can be removed from (15).*

## 4 EXPERIMENTS

We evaluate SafeFlowMatcher through experiments designed to answer three key questions:

1. Does SafeFlowMatcher outperform state-of-the-art generative model based safe planning baselines in terms of safety, planning performance, and efficiency?
2. Does SafeFlowMatcher really require a two-phase (prediction and correction) approach?

3. How well can SafeFlowMatcher generalize to more complex and high-dimensional tasks (e.g., robot locomotion and manipulation)?

We conduct experiments on a variety of planning domains: (i) Maze navigation (`maze-large-v1`), (ii) OpenAI Gym locomotion (`Walker2D-Medium-Expert-v2`, `Hopper-Medium-Expert-v2`) (Brockman et al., 2016; Todorov et al., 2012), and (iii) a robot manipulation task (block stacking) (Janner et al., 2022).

To fairly evaluate our proposed method, we extend *SafeDiffuser* (Xiao et al., 2025) beyond its original DDPM sampler. We introduce three additional safety-aware variants. For the first and second variants, we adapt DDIM (Song et al., 2021a) into two versions, *SafeDDIM($\eta$=0.0 & 1.0)*, which share the same weights as SafeDiffuser; here, $\eta$ controls the level of sampling randomness. The last variant we develop is *SafeFM*, a flow-matching counterpart to SafeDiffuser which uses the same weights as SafeFlowMatcher, but enforces safety directly during sampling and without the prediction–correction integrator. When safety constraints are disabled, we drop the "Safe" prefix. Additional details on experimental settings are provided in Appendix E.1.

For safety, we report *Barrier Safety* (BS) per constraint, the minimum value of the barrier function $b$ (which should remain non-negative), and *Trap Rate*, the rate of local trap occurrences. For planning quality, we measure the overall *Score*, the average path *Curvature* ($\kappa$), and the average path *Acceleration* ($a$) over the planning horizon. For efficiency, we report *S-Time*, the computation time per sampling step during generation, and *T-Time*, the total computation time to generate an entire path. Formal definitions of the metrics are provided in Appendix E.2.

## 4.1 MAIN RESULTS ON MAZE2D NAVIGATION

We first present the main performance comparison in the Maze2D setting, as shown in Figure 3, where there are two safety constraints (red circles). Our results illustrate that SafeFlowMatcher generates smooth, efficient paths that effectively avoid obstacles, whereas baselines may produce unsafe, suboptimal, or computationally-expensive paths.

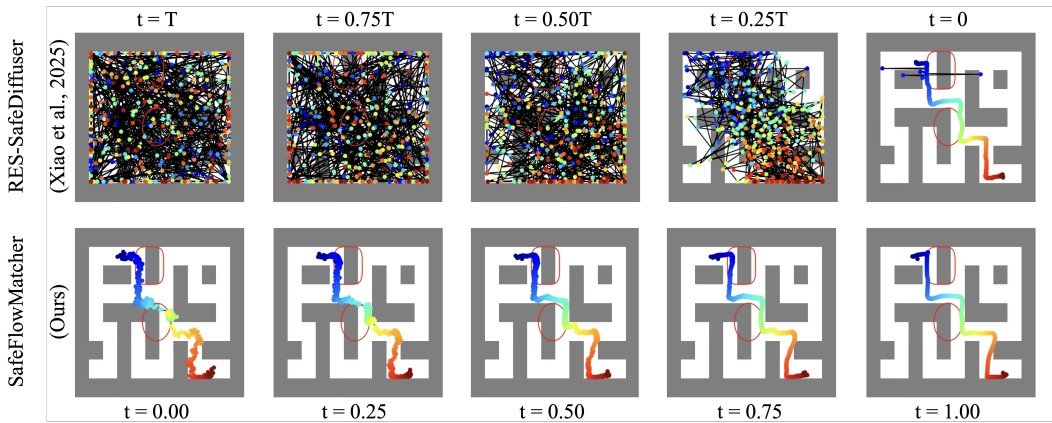

Figure 3: **Comparisons of the path generation process in Maze2D.** Red circles indicate the safety constraints the path should satisfy. (Top) RES-SafeDiffuser initializes samples all over the maze and converges to a path that has local traps. (Bottom) SafeFlowMatcher (ours) initializes from near target path after prediction phase, and converges to a higher-quality path with no local traps.[4]

As shown in Table 1, SafeFlowMatcher achieves the highest score while preserving safety, with almost no local traps. Local traps remain rare even with far more than two constraints; Appendix F.1 shows the case with six constraints.

Moreover, Figure 4 demonstrates that our method consistently outperforms all baselines, both safety-enabled and -disabled versions, across all sampling horizons. As detailed in Appendix F.3, regarding

---

[4]Diffusion-based samplers evolve backward on an interval $[0, T]$, whereas flow matching evolves forward on $[0, 1]$; a natural correspondence can be established by normalizing $T = 1$ and reversing time.

both safety and efficiency, our method also maintains $100\%$ safety even at very short sampling horizons. Especially, when $T^c = 4$ and using the closed-form solution, our method achieves **$50\times$ faster** T-Time than SafeDiffuser (0.023s vs. 1.208s), while SafeDiffuser still suffers from severe local traps that lead to incomplete paths. When using the QP solver, SafeDiffuser completes generation in 9.998s, whereas SafeFlowMatcher completes generation in just 0.157s. Notably, the QP-based Safe-FlowMatcher is still about **$8\times$ faster** than even the closed-form version of SafeDiffuser (1.208s), while achieving high task performance. We further analyze the distributional drift introduced by CBF-based corrections in Appendix F.4.

| Method | BS1 (↑) (≥ 0) | BS2 (↑) (≥ 0) | Score (↑) | S-TIME (ms) | TRAP RATE | $\kappa$ (↓) | $a$ (↓) |
|---|---|---|---|---|---|---|---|
| Diffuser (Janner et al., 2022) | -0.825 | -0.784 | 1.572±0.288 | 3.70 | 0% | 77.04±4.30 | 86.68±3.81 |
| DDIM($\eta = 0.0$) | -0.642 | -0.902 | 1.474±0.106 | 3.63 | 0% | 64.51±4.35 | 57.46±2.46 |
| DDIM($\eta = 1.0$) | -0.595 | -0.899 | 1.565±0.140 | 3.72 | 0% | 64.21±5.00 | 57.15±1.96 |
| FM | -1.000 | -1.000 | 1.422±0.359 | 3.51 | 0% | 52.09±22.02 | 33.96±22.95 |
| FlowMatcher | -0.324 | -0.904 | 1.632±0.003 | 3.51 | 0% | 73.51±1.02 | 88.45±0.60 |
| Truncation (Brockman et al., 2016) | -0.999 | -0.999 | 0.978±0.128 | 19.51 | 100% | 1118.21±1093.96 | 9.043e5±8.988e6 |
| CG (Dhariwal & Nichol, 2021) | -0.996 | -0.999 | 0.505±0.092 | 19.13 | 100% | 949.63±1103.62 | 959.71±1846.58 |
| CG-$\epsilon$ (Dhariwal & Nichol, 2021) | -0.998 | -0.999 | 0.499±0.104 | 19.87 | 100% | 1027.28±1124.70 | 1.202e9±1.1961e10 |
| ROS-SafeDiffuser (Xiao et al., 2025) | 0.010 | 0.010 | 1.435±0.502 | 4.67 | 100% | 75.15±6.67 | 422.87±86.70 |
| RES-SafeDiffuser (Xiao et al., 2025) | 0.010 | 0.010 | 1.442±0.451 | 4.72 | 72% | 80.30±13.06 | 398.17±1060.86 |
| TVS-SafeDiffuser (Xiao et al., 2025) | -0.003 | -0.003 | 1.506±0.405 | 4.78 | 69% | 78.72±7.80 | 124.51±34.22 |
| ROS-SafeDDIM($\eta = 0.0$) | 0.010 | 0.010 | 1.132±0.556 | 4.79 | 100% | 31.22±4.87 | 2073.84±1694.06 |
| RES-SafeDDIM($\eta = 0.0$) | 0.010 | 0.010 | 1.405±0.494 | 4.83 | 96% | 43.23±3.41 | 1153.81±2040.98 |
| TVS-SafeDDIM($\eta = 0.0$) | -0.026 | -0.026 | 1.522±0.295 | 4.79 | 90% | 42.56±3.39 | 575.73±371.83 |
| ROS-SafeDDIM($\eta = 1.0$) | 0.010 | 0.010 | 1.575±0.158 | 4.89 | 100% | 56.30±2.93 | 668.17±69.19 |
| RES-SafeDDIM($\eta = 1.0$) | 0.010 | 0.010 | 1.532±0.331 | 4.82 | 86% | 61.73±4.80 | 1584.00±8085.06 |
| TVS-SafeDDIM($\eta = 1.0$) | -0.026 | -0.026 | 1.549±0.304 | 4.74 | 65% | 60.29±3.41 | 27.23±43.20 |
| ROS-SafeFM | 0.010 | 0.010 | 1.138±0.556 | 4.68 | 100% | 23.57±8.34 | 1.317e4±9.931e4 |
| RES-SafeFM | 0.010 | 0.010 | 1.401±0.429 | 4.74 | 12% | 61.17±19.52 | 6724.64±5.304e4 |
| TVS-SafeFM | -0.002 | -0.002 | 1.350±0.417 | 4.73 | 41% | 60.29±3.41 | 768.71±2212.17 |
| SafeFlowMatcher w/o relaxation (ours) | 0.010 | 0.010 | 1.622±0.065 | 4.76 | 2% | 71.73±3.54 | 108.43±167.36 |
| SafeFlowMatcher (ours) | 0.010 | 0.010 | 1.632±0.003 | 4.71 | 0% | 69.19±1.02 | 91.90±0.77 |

Table 1: **Performance comparison of different methods.** We evaluated all methods over 100 independent trials under identical settings. For all safety-aware methods, we set the robustness margin to $\delta = 0.01$, meaning that a method is considered safe only if $b(\tau) \geq \delta$. This ensures robust rather than marginal safety. FlowMatcher-variants use $T^p = 1$ and $T^c = 256$, and others use $T = 256$. The closed-form CBF-QP computation takes 1.14 ms on average. All baselines are reproduced by us.

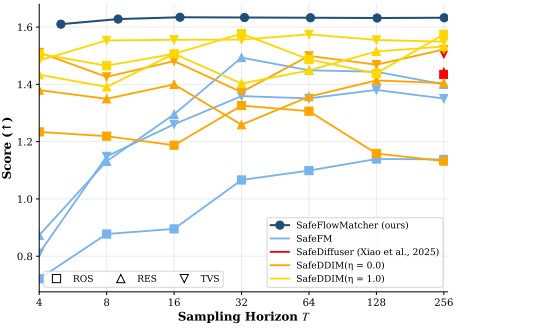 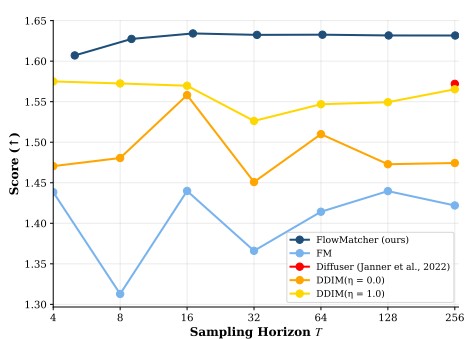

Figure 4: **Score versus sampling horizon $T$.** Left (safety on): SafeFlowMatcher attains the highest score across all sampling horizons. Right (safety off): FlowMatcher (FM + PC integrator) also remains more efficient than the other cases.

## 4.2 ABLATION STUDIES ON PC INTEGRATOR

**Effect of Using Two Phases.** To highlight the necessity of both the prediction and correction phases, we discuss the results of FlowMatcher (prediction-only), SafeFM (correction-only), and SafeFlowMatcher (PC integrator) in Table 1. The prediction-only behavior achieves good task performance but lacks safety. Conversely, the correction-only behavior enforces safety from the beginning but often fails to generate complete paths, resulting in a high trap rate. SafeFlowMatcher combines the strengths of both phases, achieving superior performance while ensuring safety.

**Effect of Prediction Horizon ($T^p$).** We analyze how the prediction horizon $T^p$ affects overall performance while keeping the correction horizon fixed at $T^c = 256$. Table 2 reports the qualities

of fully generated paths and the total computation time across different values of $T^p$, and Figure 5 visualizes how increasing $T^p$ shapes the predicted path before correction. As $T^p$ increases, the path quality remains largely unchanged, while the computation cost increases due to additional prediction steps.

Table 2: **Effect of prediction horizon $T^p$.** We compare path quality metrics (score, curvature, and acceleration) and the total computation time, measured after one full path generation.

| Prediction horizon ($T^p$) | 1 | 2 | 4 | 8 | 16 |
|---|---|---|---|---|---|
| Score($\uparrow$) | 1.632±0.008 | 1.520±0.340 | 1.468±0.434 | 1.404±0.538 | 1.632±0.003 |
| T-TIME (s) | 1.209 | 1.220 | 1.230 | 1.249 | 1.287 |
| Curvature $\kappa$($\downarrow$) | 69.19±1.02 | 68.84±3.32 | 68.70±4.62 | 68.26±4.77 | 67.73±4.97 |
| Acceleration $a$($\downarrow$) | 91.90±2.77 | 93.76±2.30 | 93.18±3.52 | 91.99±3.83 | 92.61±3.81 |

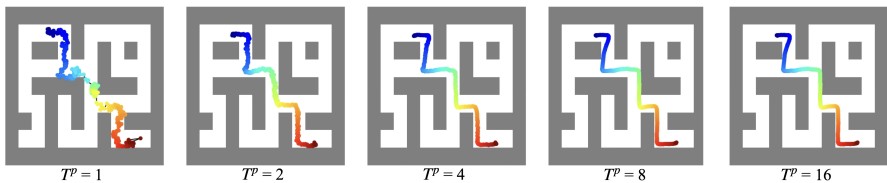

Figure 5: **Predicted paths under different prediction horizon $T^p$.** Each shows the predicted path after the prediction phase. As prediction horizon $T^p$ increases, prediction error $\varepsilon$ decreases.

**Effect of Vanishing Time-Scale.** We first analyze the role of the scaling constant $\alpha$ in VTFD (10). As shown in Table 3 and Figure 6, increasing $\alpha$ consistently reduces both curvature and acceleration, indicating that larger scaling factors suppress the prediction error more aggressively. This trend is consistent with the theoretical result from Lemma 3.

However, we observe that a larger $\alpha$ introduces bias in the final path. This effect is visible in Figure 6, where the red path region stays relatively stable up to a certain critical value but becomes increasingly distorted once $\alpha$ exceeds this threshold. In our Maze2D setup, this occurs around $\alpha \approx 2$. This shows that $\alpha$ should not simply be maximized in practice; instead, one can start from $\alpha = 1$ and increase it until we identify the point just before the sharp distortion begins.

Figure 7 shows how the score changes with increasing correction horizon $T^c$ when $T^p = 1$. With vanishing time-scale, the score remains stable even as $T^c$ grows, whereas removing the scaling causes the score to deteriorate steadily. Figure 8 provides the corresponding path visualization. With vanishing time-scale, the correction path moves from $\tau_0^c$ to $\tau_1^c$ along a straight direction. In contrast, without scaling, the path exhibits sharp drift near $t = 1$, and some segments of the path become largely distorted. These results demonstrate that vanishing time-scale is essential for preventing late-stage drift and maintaining stable refinement behavior.

Table 3: **Effect of scaling constant $\alpha$.** Path qualities are measured after full generation.

| Scaling constant $\alpha$ | 1.0 | 1.5 | 2.0 | 2.5 | 3.0 |
|---|---|---|---|---|---|
| Score ($\uparrow$) | 1.623±0.005 | 1.629±0.004 | 1.632±0.008 | 1.618±0.033 | 1.572±0.058 |
| Curvature $\kappa$ ($\downarrow$) | 85.10±3.73 | 83.91±2.00 | 69.28±1.04 | 55.16±0.83 | 44.08±0.62 |
| Acceleration $a$ ($\downarrow$) | 173.22±5.62 | 123.49±1.86 | 92.05±0.59 | 71.89±0.42 | 58.05±0.24 |

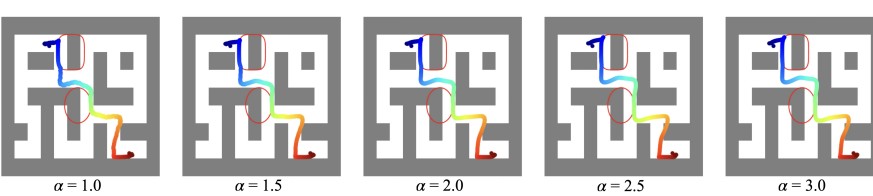

Figure 6: **Generated paths under different scaling constant $\alpha$.** Each snapshot shows the fully generated path after the two phases. As the scaling constant $\alpha$ increases, the path becomes smoother but can be distorted.

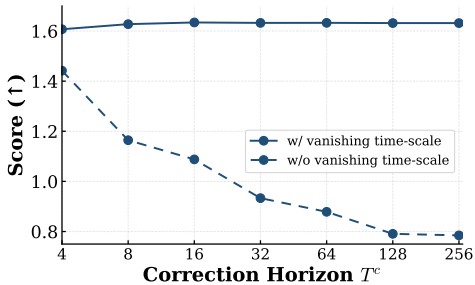

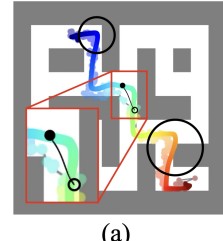 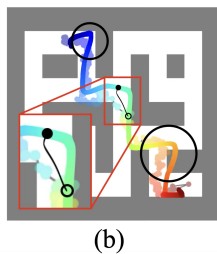

(a)  (b)

Figure 8: **Generation process with (a) and without (b) a vanishing time-scale.** The transparent path represents $\boldsymbol{\tau}_0^c$, the solid path represents $\boldsymbol{\tau}_1^c$. The black line represents the path $\boldsymbol{\tau}_t^c$ from ∘ to ● over the interval $t \in [0, 1]$. The path's segments in the black circles are largely distorted in the absence of a vanishing time-scale.

Figure 7: **Score with and without a vanishing time-scale.** When $T^p = 1$, as the correction horizon $T^c$ increases, we see that the score decreases in the absence of vanishing time-scale.

### 4.3 Generalization to High-Dimensional Robotic Tasks

We evaluate the generalization capability of SafeFlowMatcher on high-dimensional robotic tasks, including two locomotion environments (Walker2D and Hopper) and a robot manipulation task (Block Stacking). Across all three tasks, SafeFlowMatcher attains the highest score while maintaining $\mathrm{BS} \geq 0$, indicating that the PC integrator scales beyond static maze navigation. The detailed comparison across locomotion and manipulation tasks is summarized in Table 4. Note that the BS metric here is reported in a different way than in Table 1; here, BS is a binary indicator (yes or no) of whether safety is guaranteed ($\geq 0$) or not ($< 0$).

Table 4: **Performance on high-dimensional robotic tasks.** SafeFlowMatcher maintains its advantages in both locomotion and robot manipulation settings.

| Category | Environment | Method | Score (↑) | BS ($\geq 0$) |
|---|---|---|---|---|
| **Locomotion** | Walker2D | SafeDiffuser (Xiao et al., 2025) | $0.283 \pm 0.135$ | Yes |
| | | SafeFM | $0.264 \pm 0.127$ | Yes |
| | | **Ours** | $0.331 \pm 0.021$ | **Yes** |
| | Hopper | SafeDiffuser (Xiao et al., 2025) | $0.435 \pm 0.068$ | Yes |
| | | SafeFM | $0.675 \pm 0.312$ | Yes |
| | | **Ours** | $0.917 \pm 0.026$ | **Yes** |
| **Robot Manipulation** | Block Stacking | SafeDiffuser (Xiao et al., 2025) | $0.72 \pm 0.055$ | Yes |
| | | SafeFM | $0.73 \pm 0.056$ | Yes |
| | | **Ours** | $0.76 \pm 0.053$ | **Yes** |

## 5 Conclusion

We introduced *SafeFlowMatcher*, a planning framework that couples flow matching (FM) with CBF-certified safety by employing a two-phase *prediction–correction* integrator. On the path generation side, we proposed the vanishing time-scaled flow dynamics, which contracts the prediction error toward the target path. On the safety side, we established a finite-time convergence barrier certificate for the flow system to ensure forward invariance of a safe set. The approach generates a candidate path with the learned FM dynamics and then refines only the *executed* path under safety constraints. This decoupling preserves the native generative dynamics, avoids distributional drift from repeated interventions, and mitigates local trap failures near constraint boundaries. Empirically, SafeFlowMatcher attains faster, smoother, and safer paths than various diffusion- and FM-based baselines across maze navigation, locomotion, and robot manipulation tasks. Incorporating data-driven certificates is a promising direction for extending certified generative planning to more dynamic and complex environments.

## ACKNOWLEDGMENTS

We would like to thank Jiwon Park for assistance with setting up the initial experiments.

This research was supported by Brain Pool program funded by the Ministry of Science and ICT through the National Research Foundation of Korea(RS-2025-25443489).

## REPRODUCIBILITY STATEMENT

All baseline results reported in this paper are fully reproduced by us using our own implementations or publicly available code, ensuring a fair and controlled comparison on the same hardware. To facilitate reproducibility, we provide anonymized source code for training and evaluation in the supplementary material. For fair comparisons under matched computational budgets, our model architectures strictly adhere to those in prior work (Janner et al., 2022; Xiao et al., 2025) and their official implementations (Code: https://github.com/jannerm/diffuser, https://github.com/Weixy21/SafeDiffuser). Our experiments are conducted on the Maze2D environment, locomotion tasks (Hopper, Walker2D) and a robot manipulation task (block stacking). All hyperparameters for training and evaluation, including optimizer settings, learning rates, and rollout configurations, are detailed in Appendix E. For each experimental setting, we perform 100 independent trials and report the mean and standard deviation across these runs in Table 1 and Table 4. All experiments were run on a machine equipped with an AMD EPYC9354 CPU and an NVIDIA RTX4090 (24GB) GPU. Additional ablation studies are provided in Appendix F.

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

## A ADDITIONAL RELATED WORK ON CONTROL BARRIER FUNCTIONS

Control Barrier Functions have been developed and extended in a wide range of directions, and existing results show that CBF-based safety filter does not rely on perfectly known, smooth, or analytically specified safe sets. Discrete-time CBFs have been applied to hybrid locomotion and time-varying safety constraints (Agrawal & Sreenath, 2017), and duality-based DCBF methods enable safe control even with nonsmooth, polytopic, or nonconvex obstacle geometries (Liao et al., 2023). Perception noise and state-estimation uncertainty can be handled using measurement-robust and probabilistic CBF formulations (Cosner et al., 2021; Long et al., 2022). Moreover, CBFs have been extended to dynamic-obstacle environments, explicitly incorporating obstacle motion prediction and enabling real-time avoidance of moving obstacles (Jian et al., 2023).

In addition to analytic formulations, a growing line of work develops learning-based CBFs that construct safety certificates directly from data rather than hand-crafted functions. These methods learn barrier functions from RGB-D observations (Abdi et al., 2023), LiDAR scans (Srinivasan et al., 2020; Long et al., 2021; Harms et al., 2024), or expert demonstrations (Robey et al., 2020; Lindemann et al., 2024), enabling implicit representations of safe sets in dynamic and unstructured environments. While SafeFlowMatcher currently leverages analytic CBFs, its correction phase only requires evaluating a barrier constraint, making the framework compatible with these learned or perception-driven CBFs.

Beyond their theoretical development, CBF-based safety filters have also been applied across a wide range of robotic domains. They have seen successful use in autonomous driving (Ames et al., 2016), legged locomotion (Kim et al., 2023), and multi-robot coordination (Wang et al., 2017).

## B THEORETICAL AND EMPIRICAL SUPPORT FOR THE CORRECTION PHASE

### B.1 EMPIRICAL VALIDATION OF THE PREDICTION ERROR ASSUMPTION IN LEMMA 2

Lemma 2 and Lemma 3 assume that the prediction error $\varepsilon$ follows a symmetric, zero-mean distribution in a neighborhood around the target path. We empirically validate this assumption by evaluating the distribution of $\varepsilon$ under different prediction horizons $T^p \in \{1, 2, 4, 8, 16, 32\}$ in the Maze2D environment. For each configuration, we generate 1,000 predicted paths, resulting in a total of 384,000 waypoints, and evaluated the prediction error with respect to a high-accuracy FM solution $\boldsymbol{\tau}_1^\star$, which is computed using the Dormand-Prince 5(4) method (Dopri5) with 256 steps.

Figure 9 visualizes our results. Across all values of $T^p$, the distribution of $\varepsilon$ remains centered at zero and exhibits symmetry, directly supporting the symmetric zero-mean (Gaussian-like) assumption used in both lemmas. Validating whether this assumption still holds for higher-dimensional, complex tasks is a subject of future work. However, we anticipate that while the final refined path may be biased if the prediction error is biased, overall safety is still unaffected because the CBF-QP enforces forward invariance regardless of any bias. Moreover, if the bias is heavy-tailed, the local strong convexity of $-\log p_\varepsilon$ becomes weaker, which may slow down the contraction rate in the correction phase. Again, this only affects path refinement speed, not safety guarantees, and increasing $\alpha$ to introduce deliberate path distortion against the error (see Table 3 and Figure 6) might help the prediction error reduction.

### B.2 PROOFS OF LEMMA 2 AND LEMMA 3

**Proof of Lemma 2.**
Let $\phi_t(\varepsilon) = \boldsymbol{\tau}_1 + \delta\varepsilon$, where $\delta \triangleq 1 - t$. We have pushforward of $p_\varepsilon$ under $\phi_t$:

$$p_t(\boldsymbol{\tau} \mid \boldsymbol{\tau}_1) = [\phi_t]_\# p_\varepsilon(\varepsilon) = p_\varepsilon(\phi_t^{-1}(\boldsymbol{\tau}))\, det\left[\frac{\partial \phi_t^{-1}}{\partial \boldsymbol{\tau}}(\boldsymbol{\tau})\right] = \frac{1}{\delta^{d(H+1)}} p_\varepsilon\left(\frac{\boldsymbol{\tau} - \boldsymbol{\tau}_1}{\delta}\right)$$

By Bayes' rule,

$$p(\boldsymbol{\tau}_1 \mid \boldsymbol{\tau}_t^c) \propto p_1(\boldsymbol{\tau}_1)\, p_\varepsilon\left(\frac{\boldsymbol{\tau}_t^c - \boldsymbol{\tau}_1}{\delta}\right).$$

Since $-\log p_\varepsilon(z)$ is $C^2$ near 0 with Hessian $A \succ 0$ by the assumption,

$$-\log p_\varepsilon(z) = \tfrac{1}{2} z^\top A z + O(\|z\|^3).$$

Let $y = \boldsymbol{\tau}_1 - \boldsymbol{\tau}_t^c$. Substituting $z = y/\delta$ yields the posterior energy

$$\Phi_\delta(y) \ = \ \tfrac{1}{2\delta^2} y^\top A y - \log p_1(\boldsymbol{\tau}_t^c + y) + O(1).$$

The quadratic term dominates as $\delta \to 0$ ($t \to 1$), so the posterior concentrates in an $O(\delta)$ neighborhood of $\boldsymbol{\tau}_t^c$.

The stationarity condition $\nabla \Phi_\delta(y) = 0$ gives

$$\frac{1}{\delta^2} A y - \nabla \log p_1(\boldsymbol{\tau}_t^c + y) = 0.$$

Taylor expanding $\nabla \log p_1$ at $\boldsymbol{\tau}_t^c$ shows $y = O(\delta^2)$. Thus the posterior mode is

$$\hat{\boldsymbol{\tau}}_1 = \boldsymbol{\tau}_t^c + \delta^2 A^{-1} \nabla \log p_1(\boldsymbol{\tau}_t^c) + O(\delta^3).$$

Laplace's approximation then yields the same expansion for the posterior mean:

$$\mathbb{E}[\boldsymbol{\tau}_1 \mid \boldsymbol{\tau}_t^c] = \boldsymbol{\tau}_t^c + \delta^2 A^{-1} \nabla \log p_1(\boldsymbol{\tau}_t^c) + O(\delta^3).$$

Under the assumption, we have $\boldsymbol{\tau}_t^c = \boldsymbol{\tau}_1^\star + \delta\varepsilon$ with $\|\varepsilon\| = O(1)$,

$$\mathbb{E}[\boldsymbol{\tau}_1 \mid \boldsymbol{\tau}_t^c] = \boldsymbol{\tau}_1^\star + \delta\varepsilon + O(\delta^2) = \boldsymbol{\tau}_1^\star + O(\delta).$$

This proves Lemma 2.

**Proof of Lemma 3.**
If the flow dynamics follow the vanishing time-scaled flow dynamics (10), then we have:

$$\dot{\boldsymbol{\tau}}_t^c = \alpha(1 - t)\, v_t(\boldsymbol{\tau}_t^c; \theta) = \alpha(\mathbb{E}[\boldsymbol{\tau}_1 \mid \boldsymbol{\tau}_t^c] - \boldsymbol{\tau}_t^c).$$

Let $\mathbf{e}_t \triangleq \boldsymbol{\tau}_t^c - \boldsymbol{\tau}_1^\star \in \mathbb{R}^{d \times (H+1)}$, and denote its $k$-th column by $e_{k,t} \in \mathbb{R}^d$. By Lemma 2, $\mathbb{E}[\boldsymbol{\tau}_1 \mid \boldsymbol{\tau}_t^c] = \boldsymbol{\tau}_1^\star + O(1-t)$ as $t \to 1$, hence we have

$$\dot{e}_{k,t} = -\alpha e_{k,t} + O(1-t).$$

Solving with an integrating factor gives

$$e_{k,t} = e^{-\alpha t} e_{k,0} + \alpha e^{-\alpha t} \int_0^t e^{\alpha s} O(1-s)\, ds = (e_{k,0} + O(1)) e^{-\alpha t} + O((1-t)^2).$$

Combining the column vectors again yields the form

$$\mathbf{e}_t = (\mathbf{e}_0 + O(1)) e^{-\alpha t} + O((1-t)^2), \qquad \mathbf{e}_0 = \varepsilon.$$

which proves Lemma 3.

## C  PROOF OF THEOREM 1 AND PROPOSITION 1

We drop the superscript $c$ for simplicity, and choose the Lyapunov candidate function $V(\boldsymbol{\tau}_t^k) \triangleq \max(\delta - b(\boldsymbol{\tau}_t^k), 0)$. Since $w(t) = 0$ for all $t \geq t_w$, the barrier inequality (15) reduces on $[t_w, 1]$ to

$$\dot{b}(\boldsymbol{\tau}_t^k) + \epsilon \cdot \mathrm{sgn}(b(\boldsymbol{\tau}_t^k) - \delta)\, |b(\boldsymbol{\tau}_t^k) - \delta|^\rho \ \geq \ 0.$$

**Case 1:** If $\boldsymbol{\tau}_{t_w}^k \in \mathcal{C}_\delta$ (i.e., $b(\boldsymbol{\tau}_{t_w}^k) \geq \delta$), then $V(\boldsymbol{\tau}_{t_w}^k) = 0$. For all $t \geq t_w$, if $b(\boldsymbol{\tau}_t^k) > \delta$ we have $V(\boldsymbol{\tau}_t^k) = 0$. If $b(\boldsymbol{\tau}_t^k) = \delta$, the barrier inequality (15) with $\mathrm{sgn}(0) = 0$ reduces to $\dot{b}(\boldsymbol{\tau}_t^k) \geq 0$, so the path cannot exit $\mathcal{C}_\delta$ by Nagumo's principle (Nagumo, 1942)[5]. Therefore $V(\boldsymbol{\tau}_t^k) = 0$ for all $t \geq t_w$, which implies $\boldsymbol{\tau}_t^k \in \mathcal{C}_\delta$; the system stays in $\mathcal{C}_\delta$.

**Case 2:** If $\boldsymbol{\tau}_{t_w}^k \notin \mathcal{C}_\delta$ (i.e., $b(\boldsymbol{\tau}_{t_w}^k) < \delta$), then $V(\boldsymbol{\tau}_t^k) = \delta - b(\boldsymbol{\tau}_t^k) > 0$. The following finite-stability condition holds

$$\dot{V}(\boldsymbol{\tau}_t^k) = -\dot{b}(\boldsymbol{\tau}_t^k) \leq -\epsilon(\delta - b(\boldsymbol{\tau}_t^k))^\rho = -\epsilon V(\boldsymbol{\tau}_t^k)^\rho.$$

---

[5]Nagumo's theorem states that if the vector field at the boundary lies in the tangent cone of a set, then the set is forward invariant.

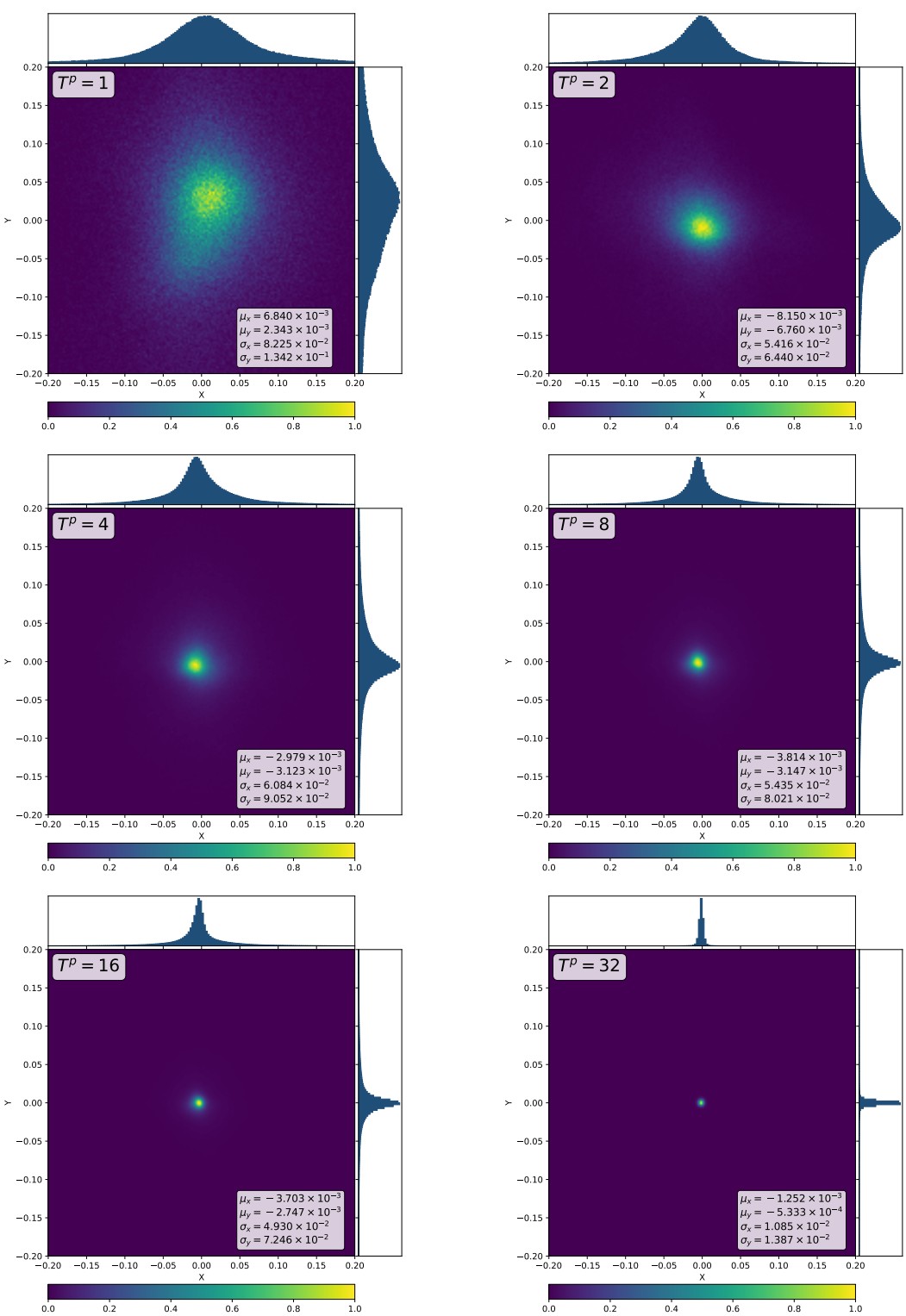

Figure 9: **Empirical distribution of the prediction error $\varepsilon$ over prediction horizon $T^p$.** The six subfigures correspond to $T^p = 1, 2, 4, 8, 16, 32$ (from top-left to bottom-right). Each subplot visualizes the joint density of $(\varepsilon_x, \varepsilon_y)$ with its marginal distributions. As $T^p$ increases, the error distribution becomes more concentrated around zero while maintaining symmetry, validating the symmetric zero-mean assumption used in Lemma 2 and Lemma 3.

Define the comparison system

$$\dot{\phi}(t) = -\epsilon\phi(t)^\rho, \ \phi(t_w) = V(\boldsymbol{\tau}_{t_w}^k).$$

By the Comparison Lemma (See Lemma 3.4 in Khalil & Grizzle (2002)), we have:

$$V(\boldsymbol{\tau}_t^k) \leq \phi(t), \ \forall t \geq t_w.$$

The solution $\phi(t)$ is

$$\phi(t) = \left(V(\boldsymbol{\tau}_{t_w}^k)^{1-\rho} - (1-\rho)\epsilon(t-t_w)\right)^{\frac{1}{1-\rho}}, \ \text{for } t \geq t_w.$$

Thus,

$$V(\boldsymbol{\tau}_t^k) \leq \left(V(\boldsymbol{\tau}_{t_w}^k)^{1-\rho} - (1-\rho)\epsilon(t-t_w)\right)^{\frac{1}{1-\rho}}.$$

Hence, $\boldsymbol{\tau}_t^k$ reaches the robust safe set $\mathcal{C}_\delta$ in finite time $T$ that satisfies $V(\boldsymbol{\tau}_t^k) \leq \phi(T) = 0$. Moreover, we get the finite convergence time,

$$T = t_w + \frac{V(\boldsymbol{\tau}_{t_w}^k)^{1-\rho}}{\epsilon(1-\rho)} = t_w + \frac{(\delta - b(\boldsymbol{\tau}_{t_w}^k))^{1-\rho}}{\epsilon(1-\rho)}.$$

Therefore, for all $t \geq T$, we have $V(\boldsymbol{\tau}_t^k) \leq 0$, implying $\mathbf{x} \in \mathcal{C}_\delta$. This completes the proofs of both Theorem 1 and Proposition 1.

## D    DIFFERENCES IN LOCAL TRAP DEFINITIONS

We clarify the difference between the local trap definition used in our SafeFlowMatcher and that of the baseline method SafeDiffuser (Xiao et al., 2025).

**Definition 4 (Local Trap in SafeDiffuser)** *A local trap problem occurs during the planning process if there exists $k \in \mathcal{H}$ such that $b(\boldsymbol{\tau}_1^k) = 0$ and $\|\boldsymbol{\tau}_1^k - \boldsymbol{\tau}_1^{k-1}\| > \zeta$, where $\zeta > 0$ is a user-defined threshold depending on the planning environment.*

In contrast, our definition of a local trap in SafeFlowMatcher removes the condition $b(\boldsymbol{\tau}_1^k) = 0$ and instead considers only the abrupt discontinuity in the path. The reason for relaxing the condition is illustrated in Figure 10. In this example, the generated path is incomplete due to overly strong or early intervention of the CBF. However, since the waypoints do not strictly lie on the boundary (i.e., $b(\boldsymbol{\tau}_1^k) \neq 0$), the original SafeDiffuser definition fails to detect this failure as a local trap. Therefore, we generalize the definition to capture a wider class of failure cases.

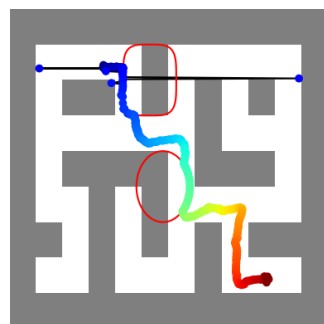

Figure 10: **Local trap occurring away from the safety boundary.** Although some waypoints do not violate constraints (i.e., $b(\boldsymbol{\tau}_t^k) > 0$), it fails to reach the goal. Our definition considers such cases as local traps, while the original definition does not.

## E    EXPERIMENTAL DETAILS

### E.1    EXPERIMENTAL SETUP

All CBF constraints are enforced via the closed-form projection of the CBF-QP in (17). For each model family, the safety-enabled variants reuse the same trained weights as their safety-disabled counterparts. Specifically, SafeDiffuser and SafeDDIM share the weights trained for Diffuser and DDIM, respectively, while SafeFM and SafeFlowMatcher share the weights trained for FM and FlowMatcher. All experiments are run using an AMD EPYC 9354 CPU and an NVIDIA RTX 4090 GPU (24GB).

**Maze2D.**    To match the total amount of training data used in Diffuser (Janner et al., 2022), we first swept across several batch sizes while fixing the total number of samples processed during training to $6.4 \times 10^7$. As shown in Table 5, both Diffuser and FM performed best or on par at batch size 128, so for all models and Maze2D experiments, we used batch size 128. Other training

and inference hyperparameters are shown in Tables 6. For the correction phase, we set the scaling constant to $\alpha = 2$, and use $(\delta, \epsilon, \rho) = (0.01, 0.5, 0.9)$ for the CBF parameters. Additionally, for the relaxation schedule, $t_w$ is chosen according to the correction horizon $T^c$. Specifically, we use $t_w \in \{0, 0.5, 0.75, 0.9, 0.9, 0.9, 0.99\}$ for $T^c \in \{4, 8, 16, 32, 64, 128, 256\}$, respectively. The relaxation function is defined as $w_t^k = 200(1 - e^{3(t/t_w - 1)})$ for $t \leq t_w$, and $w_t^k = 0$ otherwise. For Maze2D, the planner is conditioned on the start and goal observations, which are provided as the condition for each rollout.

Table 5: Scores by batch size for Maze2D for both Diffuser and FM.

| Method | 16 | 32 | 64 | 128 | 256 |
|---|---|---|---|---|---|
| **FlowMatcher** | $1.631 \pm 0.003$ | $1.628 \pm 0.002$ | $1.615 \pm 0.031$ | $1.631 \pm 0.003$ | $1.523 \pm 0.196$ |
| **Diffuser** (Janner et al., 2022) | $1.503 \pm 0.424$ | $1.438 \pm 0.500$ | $1.516 \pm 0.316$ | $1.537 \pm 0.265$ | $1.536 \pm 0.338$ |

**Locomotion.** Following the observations from Maze2D, we also train all locomotion models using a batch size of 128. SafeFlowMatcher, SafeFM, and SafeDiffuser share the same hyperparameter settings, summarized in Table 7. To provide score-based guidance to all flow-matching-based methods, including SafeFM and SafeFlowMatcher, we apply a simple covariance-aware guidance $g^{\text{cov-A}}$ with scale $1.0$, following prior work (Feng et al., 2025). During planning, we condition the model at each environment step on the current state observation and use the task score as a guidance signal to encourage forward progress.

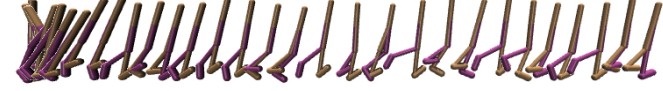

(a) Walker2D planning result with SafeFlowMatcher.

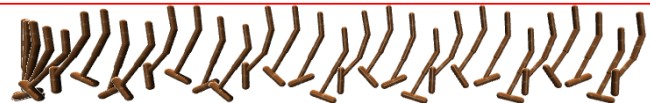

(b) Hopper planning result with SafeFlowMatcher.

Figure 11: **SafeFlowMatcher on locomotion tasks.** Planning results for Walker2D (top) and Hopper (bottom). In both figures, the red horizontal line indicates the roof height $h_r$ in the CBF barrier function ($z \leq h_r$) used in the BS metric (Appendix E.2).

**Robot Manipulation (Block-Stacking).** For the block stacking task, we followed the training parameters from Diffuser Janner et al. (2022) (batch size 32 with 2-step gradient accumulation, equivalent to batch size 64 without accumulation), rather than using a batch size of 128, while maintaining the number of training steps for training SafeFlowMatcher(SafeFM sharing weights with SafeFlowMatcher) and SafeDiffuser. Other hyperparameter values and conditions are shown in Table 8. For the block stacking task, the condition includes the initial robot joint configuration together with the observed states of the four blocks.

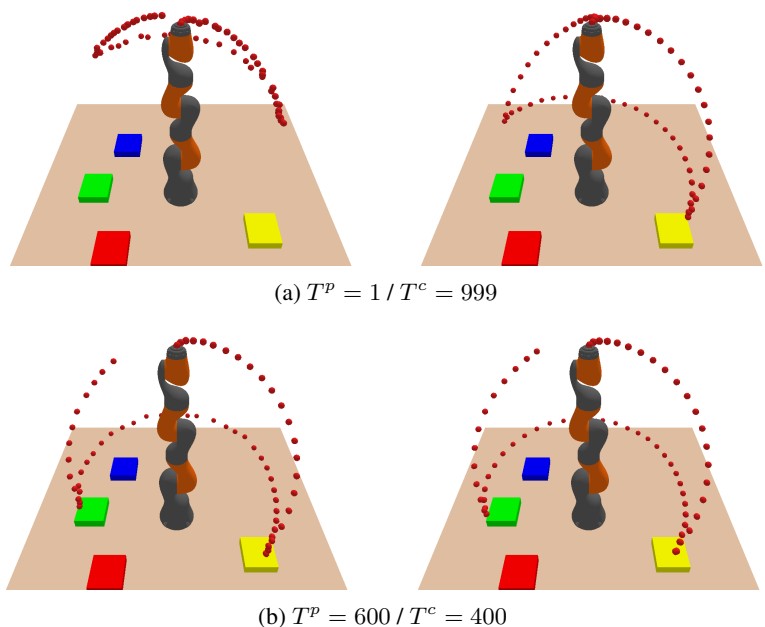

(a) $T^p = 1 / T^c = 999$

(b) $T^p = 600 / T^c = 400$

Figure 12: **Block stacking visual comparison based on the prediction and correction horizons.** For each subfigure, the left shows the predicted path $\boldsymbol{\tau}_1^p$ and the right shows the corrected path $\boldsymbol{\tau}_1^c$. Under the same planning horizon $H = 128$, we compare different allocations of $T^p$ and $T^c$. In (a), using $T^p = 1$ and $T^c = 999$ leads to poor prediction quality due to the short prediction phase, resulting in a large prediction error and ultimately a failed path. In contrast, (b) uses $T^p = 600$ and $T^c = 400$ which yields a small prediction error and successfully produces a safe and complete path, where the yellow block is being stacked on top of the green block.

Table 6: Maze2D's training and evaluation hyperparameters

| **Training** | |
| --- | --- |
| Loss type | L2 |
| Training steps $n_{\text{train}}$ | $5.0 \times 10^5$ |
| Steps per epoch | 2500 |
| Batch size | 128 |
| Learning rate | $3 \times 10^{-4}$ |
| EMA decay | 0.995 |
| **Evaluation Others** | |
| Planning Horizon $H$ | 384 |
| Sampling Horizon $T$ | 256 |
| **Evaluation SafeFlowMatcher** | |
| Planning Horizon $H$ | 384 |
| Prediction Horizon $T^p$ | 1 |
| Correction Horizon $T^c$ | 256 |

### E.2 PERFORMANCE METRICS

**BS** quantifies the degree of safety constraint satisfaction using CBFs for each safety constraint in the environment. For each rollout, we evaluate the minimum barrier value over all waypoints, and then take the worst case across all $N$ test episodes:

$$\min_{i=1,2,\ldots,N} \min_{k \in \mathcal{H}} b(\boldsymbol{\tau}_1^k).$$

Table 7: Locomotion (Walker2D/Hopper) hyperparameters

| **Training** | |
|---|---|
| Loss type | L2 |
| Training steps $n_{\text{train}}$ | $2.5 \times 10^5$ |
| Steps per epoch | 2500 |
| Batch size | 128 |
| Learning rate | $2 \times 10^{-4}$ |
| EMA decay | 0.995 |
| **Value Network Training** | |
| Loss type | L2 |
| Training steps $n_{\text{train}}$ | $5.0 \times 10^4$ |
| Steps per epoch | 2500 |
| Batch size | 128 |
| Learning rate | $2 \times 10^{-4}$ |
| EMA decay | 0.995 |
| **Evaluation Others** | |
| Planning Horizon $H$ | 600 |
| Sampling Horizon $T$ | 20 |
| **Evaluation SafeFlowMatcher** | |
| Planning Horizon $H$ | 600 |
| Prediction Horizon $T^p$ | 1 |
| Correction Horizon $T^c$ | 20 |

Table 8: Robot manipulation task (block stacking) hyperparameters

| **Training** | |
|---|---|
| Loss type | L2 |
| Training steps $n_{\text{train}}$ | $7.0 \times 10^5$ |
| Batch size | 64 |
| Learning rate | $2 \times 10^{-5}$ |
| EMA decay | 0.995 |
| **Evaluation Others** | |
| Planning Horizon $H$ | 128 |
| Sampling Horizon $T$ | 1000 |
| **Evaluation SafeFlowMatcher** | |
| Planning Horizon $H$ | 128 |
| Prediction Horizon $T^p$ | 600 |
| Correction Horizon $T^c$ | 400 |

A value $\text{BS} \geq 0$ indicates that the path remains entirely within the safe set. Maze2D contains two obstacle-based safety constraints, given by the barrier functions:

$$\textbf{BS1}: \left(\frac{x - x_0}{a}\right)^2 + \left(\frac{y - y_0}{b}\right)^2 \geq 1, \qquad \textbf{BS2}: \left(\frac{x - x_0}{a}\right)^4 + \left(\frac{y - y_0}{b}\right)^4 \geq 1.$$

where $(x, y) \in \mathbb{R}^2$ denotes the agent's 2D state, $(x_0, y_0) \in \mathbb{R}^2$ specifies the center of the obstacle, and $a, b > 0$ are scaling parameters that shape the corresponding safety region. For locomotion tasks (Walker2D, Hopper), the barrier function is defined as $z \leq h_r$, where $h_r > 0$ denotes the roof height. For robot manipulation task (block stacking), the safety constraints enforce joint limits. The barrier functions are defined as $\mathbf{q}_{\min} \leq \mathbf{q} \leq \mathbf{q}_{\max}$, where $\mathbf{q} \in \mathbb{R}^7$ denotes the joint-angle and $\mathbf{q}_{\min}, \mathbf{q}_{\max} \in \mathbb{R}^7$ are the per-joint limits.

**Score** is a normalized, undiscounted performance metric that reflects task success. In Maze2D, episodes last up to 800 environment steps while planning is performed over a horizon of $H = 384$; once the agent enters a goal neighborhood, it receives a unit reward for each remaining step, making the score proportional to the remaining horizon. For locomotion tasks, the score is proportional to forward displacement and normalized such that reaching the target position $x = 1$ yields a score of 1. We evaluate locomotion in a receding-horizon manner, continuing until the agent either reaches $x = 1$, falls, or reaches the maximum episode limit of 1000 steps. For a robot manipulation task (block stacking), planning is performed with horizon $H = 128$, and each episode attempts a single stacking action. An episode receives a score of 1 upon a successful stack and 0 otherwise.

**Trap Rate** measures the rate of local traps, i.e., the percentage of episodes in which the generated path becomes stuck against barrier constraints; see Definition 2.

**T-Time & S-Time.** We report two timing metrics: the total computation time (T-Time) and the per-step sampling time (S-Time), both of which include all computational overheads such as CBF-QP evaluations. T-Time measures the total wall-clock time required to generate an entire path, including all prediction and correction steps when applicable. S-Time reports the average wall-clock time per sampling step, computed as S-Time = T-Time$/T$, where $T$ is the total sampling horizon.

**Curvature** ($\kappa$) measures local path bending using the Menger curvature computed from triplets of consecutive points. We report the average curvature along the path.

**Acceleration** ($a$) captures the change in velocity across consecutive time steps. It is computed as the mean squared acceleration magnitude along the path. We approximate it via the second-order finite difference of the 2D position and define the metric as the average acceleration magnitude along the path.

# F    ADDITIONAL ABLATION STUDIES

## F.1    HANDLING MULTIPLE CBF CONSTRAINTS AND MITIGATING COMPUTATION BOTTLENECK

We considered only two CBF constraints so far. When more than two constraints are present, no closed-form solution is available, and a QP solver must be used to compute the CBF-QP at every step. This inevitably increases computational overhead and can become a bottleneck.

Figure 13 presents the path generation results with six CBF constraints, under a fixed sampling horizon $T = 256$, while varying the allocation between prediction and correction horizon $(T^p, T^c)$. The corresponding T-TIME and Trap Rate for each configuration are also reported. We observe that SafeFlowMatcher maintains a trap rate of 0% across all settings. However, as $T^c$ increases, the T-TIME grows due to repeated CBF-QP solves during the correction phase.

A key advantage of SafeFlowMatcher is that the PC integrator naturally mitigates this computational bottleneck. Since CBF-QP computations occur only in the correction phase, $T^p$ and $T^c$ can be adjusted to reduce the number of QP evaluations while maintaining safety. In contrast, SafeDiffuser and SafeFM require CBF-QP computations at every generation step, resulting in significantly higher overhead when many constraints are present. Moreover, SafeDiffuser becomes unstable in high-constraint settings. As presented in Table 1, local traps are already problematic in the two constraints setting. They occur even more frequently as the number of safety constraints increases. In the six CBF constraints setting, SafeDiffuser required T-TIME = 10.269 s and exhibited a 100% trap rate over 100 runs. These observations highlight that the PC integrator enables SafeFlowMatcher to scale efficiently and robustly to environments with many CBF constraints, both in terms of computational latency and safe path generation.

## F.2    EXPLORING THE FEASIBLE RANGE OF CBF HYPERPARAMETERS $\rho$ AND $\epsilon$

We examine the sensitivity of SafeFlowMatcher and SafeFM to the CBF hyperparameters $\rho$ and $\epsilon$. Smaller $\rho$ or larger $\epsilon$ induce more aggressive safety corrections, which can help fast convergence to a safe set but may also increase curvature. When excessively strong, these corrections can even lead to unstable or oscillatory behavior.

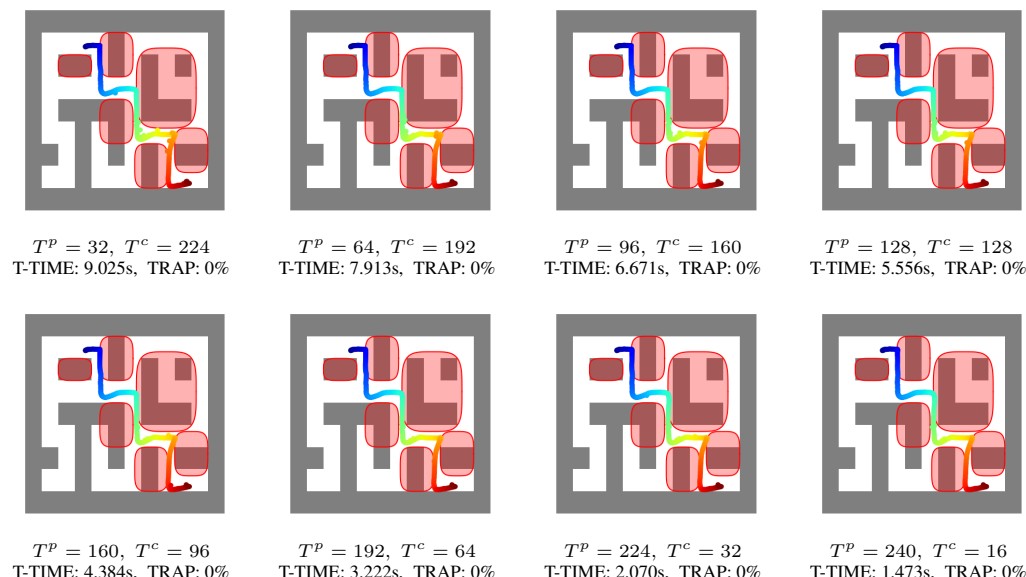

$T^p = 32,\ T^c = 224$
T-TIME: 9.025s, TRAP: 0%

$T^p = 64,\ T^c = 192$
T-TIME: 7.913s, TRAP: 0%

$T^p = 96,\ T^c = 160$
T-TIME: 6.671s, TRAP: 0%

$T^p = 128,\ T^c = 128$
T-TIME: 5.556s, TRAP: 0%

$T^p = 160,\ T^c = 96$
T-TIME: 4.384s, TRAP: 0%

$T^p = 192,\ T^c = 64$
T-TIME: 3.222s, TRAP: 0%

$T^p = 224,\ T^c = 32$
T-TIME: 2.070s, TRAP: 0%

$T^p = 240,\ T^c = 16$
T-TIME: 1.473s, TRAP: 0%

Figure 13: **Balancing prediction and correction horizon in narrow-corridor setting.** Visualization of the prediction–correction trade-off under a fixed total sampling horizon $T = T^p + T^c = 256$. Each result shows the resulting path for a different allocation of prediction steps $T^p$ and correction steps $T^c$ in the narrow-corridor setting.

Across a sweep of $\rho \in \{0.1, 0.3, 0.5, 0.7, 0.9\}$ and $\epsilon \in \{0.25, 0.50, 1.00, 2.50, 5.00, 10.00\}$, Safe-FlowMatcher, which include PC integrator, remains stable over a significantly broader hyperparameter range than SafeFM (the correction-only variant), making it substantially easier to tune in practice. This behavior is consistent with Remark 1, which explains that the prediction phase places the path closer to a region where safety enforcement is feasible and well-conditioned, resulting in more robust behavior under different CBF strengths.

Table 9: **Comparison between SafeFlowMatcher and SafeFM on CBF hyperparameters.** Subset of the $(\rho, \epsilon)$ hyperparameter grid in Maze2D, comparing SafeFlowMatcher (ours) and SafeFM (w/o PC). Each entry reports mean $\pm$ std over 100 rollouts for Score, Trap Rate, curvature ($\kappa$), acceleration ($a$), and minimum barrier values (BS1, BS2).

| $\rho$ | $\epsilon$ | Score | | Trap Rate | | $\kappa\ (\downarrow)$ | | $a\ (\downarrow)$ | | BS1 | | BS2 | |
|---|---|---|---|---|---|---|---|---|---|---|---|---|---|
| | | Ours | w/o PC | Ours | w/o PC | Ours | w/o PC | Ours | w/o PC | Ours | w/o PC | Ours | w/o PC |
| 0.1 | 0.25 | $1.632 \pm 0.003$ | $0.446 \pm 0.649$ | 0% | 100% | $76.640 \pm 1.446$ | $1.766 \pm 0.209$ | $94.579 \pm 1.116$ | $2.180e{+}04 \pm 1.083e{+}04$ | 0.010 | -0.058 | 0.009 | -0.101 |
| 0.1 | 0.50 | $1.633 \pm 0.003$ | $0.526 \pm 0.673$ | 0% | 100% | $77.919 \pm 1.570$ | $1.961 \pm 0.257$ | $96.953 \pm 1.408$ | $2.713e{+}04 \pm 4.833e{+}04$ | 0.010 | -0.189 | 0.009 | -0.211 |
| 0.1 | 1.00 | $1.632 \pm 0.003$ | $0.639 \pm 0.691$ | 0% | 100% | $78.349 \pm 1.561$ | $2.275 \pm 0.317$ | $102.139 \pm 3.267$ | $2.530e{+}04 \pm 4.340e{+}04$ | 0.010 | -0.041 | 0.010 | -0.200 |
| 0.1 | 2.50 | $1.633 \pm 0.005$ | $0.709 \pm 0.699$ | 0% | 100% | $79.337 \pm 1.898$ | $3.041 \pm 0.413$ | $109.303 \pm 6.840$ | $1.755e{+}04 \pm 1.035e{+}04$ | 0.010 | -0.022 | 0.010 | -0.383 |
| 0.1 | 5.00 | $1.633 \pm 0.004$ | $1.025 \pm 0.613$ | 5% | 100% | $80.416 \pm 1.605$ | $3.871 \pm 0.556$ | $141.126 \pm 16.164$ | $1.418e{+}04 \pm 2.735e{+}03$ | 0.010 | -0.111 | 0.009 | -0.107 |
| 0.1 | 10.00 | $1.633 \pm 0.003$ | $1.215 \pm 0.539$ | 50% | 100% | $81.395 \pm 1.435$ | $5.147 \pm 0.581$ | $174.368 \pm 31.192$ | $1.160e{+}04 \pm 2.497e{+}03$ | 0.010 | -0.888 | 0.010 | -0.111 |
| 0.3 | 0.25 | $1.632 \pm 0.003$ | $0.628 \pm 0.685$ | 0% | 100% | $73.121 \pm 1.286$ | $2.205 \pm 0.334$ | $92.501 \pm 0.775$ | $1.885e{+}04 \pm 4.554e{+}03$ | 0.010 | -0.036 | 0.008 | 0.014 |
| 0.3 | 0.50 | $1.632 \pm 0.004$ | $0.702 \pm 0.727$ | 0% | 100% | $75.438 \pm 1.267$ | $2.765 \pm 0.386$ | $93.140 \pm 0.740$ | $1.895e{+}04 \pm 1.503e{+}04$ | 0.010 | -0.056 | 0.009 | 0.017 |
| 0.3 | 1.00 | $1.632 \pm 0.003$ | $0.812 \pm 0.669$ | 0% | 100% | $77.971 \pm 1.377$ | $4.105 \pm 0.561$ | $93.835 \pm 0.949$ | $4.505e{+}04 \pm 1.266e{+}05$ | 0.010 | -0.050 | 0.009 | 0.021 |
| 0.3 | 2.50 | $1.634 \pm 0.003$ | $1.136 \pm 0.588$ | 0% | 100% | $78.486 \pm 1.510$ | $7.826 \pm 1.181$ | $95.123 \pm 1.614$ | $1.333e{+}04 \pm 3.713e{+}04$ | 0.010 | -0.045 | 0.009 | 0.029 |
| 0.3 | 5.00 | $1.633 \pm 0.003$ | $1.323 \pm 0.503$ | 0% | 100% | $79.055 \pm 1.453$ | $15.566 \pm 3.422$ | $101.603 \pm 3.888$ | $3.710e{+}03 \pm 8.006e{+}02$ | 0.010 | -0.122 | 0.009 | 0.067 |
| 0.3 | 10.00 | $1.633 \pm 0.003$ | $1.323 \pm 0.438$ | 0% | 100% | $79.345 \pm 1.786$ | $21.111 \pm 4.519$ | $113.198 \pm 8.455$ | $3.093e{+}03 \pm 1.456e{+}03$ | 0.010 | -0.240 | 0.010 | 0.071 |
| 0.5 | 0.25 | $1.632 \pm 0.003$ | $1.083 \pm 0.581$ | 0% | 100% | $70.276 \pm 1.101$ | $4.186 \pm 0.607$ | $92.016 \pm 0.787$ | $1.953e{+}04 \pm 3.482e{+}04$ | 0.009 | -0.005 | 0.008 | -0.005 |
| 0.5 | 0.50 | $1.631 \pm 0.007$ | $1.318 \pm 0.481$ | 0% | 100% | $72.149 \pm 1.218$ | $8.378 \pm 1.172$ | $92.333 \pm 0.676$ | $4.498e{+}04 \pm 2.718e{+}05$ | 0.010 | -0.044 | 0.008 | -0.047 |
| 0.5 | 1.00 | $1.632 \pm 0.005$ | $1.356 \pm 0.418$ | 0% | 100% | $74.715 \pm 1.251$ | $25.105 \pm 8.403$ | $92.021 \pm 0.760$ | $5.606e{+}03 \pm 1.862e{+}04$ | 0.010 | -0.190 | 0.009 | -0.181 |
| 0.5 | 2.50 | $1.632 \pm 0.004$ | $1.404 \pm 0.363$ | 0% | 94% | $77.517 \pm 1.436$ | $62.906 \pm 16.457$ | $91.634 \pm 0.676$ | $1.370e{+}03 \pm 5.066e{+}03$ | 0.010 | -0.550 | 0.009 | -0.623 |
| 0.5 | 5.00 | $1.633 \pm 0.003$ | $1.424 \pm 0.419$ | 0% | 100% | $78.220 \pm 1.389$ | $48.093 \pm 6.626$ | $92.980 \pm 1.315$ | $1.015e{+}03 \pm 2.653e{+}02$ | 0.010 | -0.529 | 0.010 | -0.813 |
| 0.5 | 10.00 | $1.632 \pm 0.009$ | $1.334 \pm 0.474$ | 0% | 100% | $78.742 \pm 1.554$ | $31.623 \pm 4.874$ | $96.749 \pm 2.255$ | $2.011e{+}03 \pm 1.126e{+}03$ | 0.010 | -0.478 | 0.010 | -0.634 |
| 0.7 | 0.25 | $1.632 \pm 0.005$ | $1.416 \pm 0.423$ | 0% | 97% | $69.277 \pm 1.121$ | $46.769 \pm 16.247$ | $92.030 \pm 0.821$ | $3.148e{+}03 \pm 9.812e{+}03$ | 0.010 | 0.075 | -0.039 | 0.004 |
| 0.7 | 0.50 | $1.632 \pm 0.003$ | $1.318 \pm 0.511$ | 0% | 29% | $70.150 \pm 1.057$ | $63.292 \pm 20.617$ | $92.006 \pm 0.832$ | $4.445e{+}03 \pm 2.939e{+}04$ | 0.009 | 0.129 | 0.008 | -0.002 |
| 0.7 | 1.00 | $1.632 \pm 0.003$ | $1.381 \pm 0.453$ | 0% | 49% | $71.967 \pm 1.215$ | $70.206 \pm 21.172$ | $91.808 \pm 0.642$ | $1.733e{+}04 \pm 1.123e{+}05$ | 0.010 | 0.042 | 0.008 | -0.083 |
| 0.7 | 2.50 | $1.632 \pm 0.004$ | $1.389 \pm 0.450$ | 0% | 90% | $74.581 \pm 1.317$ | $69.241 \pm 15.931$ | $90.505 \pm 0.637$ | $4.953e{+}04 \pm 4.876e{+}05$ | 0.010 | -0.037 | 0.008 | -0.552 |
| 0.7 | 5.00 | $1.633 \pm 0.003$ | $1.277 \pm 0.525$ | 0% | 100% | $76.222 \pm 1.362$ | $52.326 \pm 9.061$ | $90.312 \pm 0.615$ | $9.065e{+}02 \pm 3.554e{+}02$ | 0.010 | -0.064 | 0.009 | -0.767 |
| 0.7 | 10.00 | $1.632 \pm 0.007$ | $1.363 \pm 0.394$ | 0% | 100% | $77.335 \pm 1.329$ | $35.424 \pm 5.417$ | $91.406 \pm 0.738$ | $1.606e{+}03 \pm 3.131e{+}02$ | 0.010 | -0.094 | 0.009 | -0.716 |
| 0.9 | 0.25 | $1.631 \pm 0.011$ | $1.321 \pm 0.526$ | 0% | 17% | $71.163 \pm 1.132$ | $59.639 \pm 20.682$ | $91.879 \pm 0.629$ | $1.688e{+}04 \pm 1.575e{+}05$ | -0.010 | -0.055 | -0.101 | -0.047 |
| 0.9 | 0.50 | $1.632 \pm 0.004$ | $1.421 \pm 0.380$ | 0% | 25% | $69.218 \pm 0.897$ | $58.677 \pm 20.838$ | $92.114 \pm 0.786$ | $1.040e{+}03 \pm 3.773e{+}03$ | 0.010 | 0.010 | 0.010 | 0.011 |
| 0.9 | 1.00 | $1.632 \pm 0.006$ | $1.335 \pm 0.473$ | 0% | 26% | $70.471 \pm 0.955$ | $63.532 \pm 21.366$ | $91.757 \pm 0.777$ | $4.596e{+}03 \pm 3.320e{+}04$ | 0.010 | 0.002 | 0.010 | -0.011 |
| 0.9 | 2.50 | $1.632 \pm 0.003$ | $1.406 \pm 0.428$ | 0% | 79% | $72.022 \pm 1.198$ | $70.335 \pm 17.711$ | $90.333 \pm 0.703$ | $433.667 \pm 572.690$ | 0.010 | -0.514 | 0.008 | -0.553 |
| 0.9 | 5.00 | $1.633 \pm 0.003$ | $1.391 \pm 0.410$ | 0% | 100% | $72.987 \pm 1.284$ | $57.745 \pm 10.115$ | $89.935 \pm 0.688$ | $743.566 \pm 262.536$ | 0.010 | -0.530 | 0.008 | -0.691 |
| 0.9 | 10.00 | $1.632 \pm 0.003$ | $1.419 \pm 0.361$ | 0% | 100% | $74.981 \pm 1.450$ | $40.196 \pm 5.018$ | $89.995 \pm 0.664$ | $1.350e{+}03 \pm 255.700$ | 0.010 | -0.434 | 0.009 | -0.759 |

## F.3 Efficiency of SafeFlowMatcher Across Correction Horizons

Table 10: **Closed-Form CBF Solver: Computation time across correction horizons.** Comparison of RES-SafeDiffuser with a fixed sampling horizon $T = 256$ and SafeFlowMatcher for a fixed prediction horizon $T^p = 1$ and varying correction horizons $T^c \in \{4, 8, 16, 32, 64, 128, 256\}$ in Maze2D, when using the closed-form solution of CBF-QP. Each entry reports mean $\pm$ std over 100 rollouts.

| Method (Closed-Form CBF) | Score ($\uparrow$) | T-Time (s) | Trap Rate | $\kappa$ ($\downarrow$) | $a$ ($\downarrow$) | BS1&BS2 ($\geq 0$) |
|---|---|---|---|---|---|---|
| RES-SafeDiffuser (Xiao et al., 2025) | $1.442 \pm 0.451$ | 1.208 | 72% | $80.30 \pm 13.06$ | $398.17 \pm 1060.86$ | Yes |
| SafeFlowMatcher ($T^c$=4) | $1.610 \pm 0.029$ | 0.023 | 17% | $79.26 \pm 2.29$ | $252.01 \pm 18.19$ | Yes |
| SafeFlowMatcher ($T^c$=8) | $1.627 \pm 0.018$ | 0.042 | 0% | $75.72 \pm 1.64$ | $114.03 \pm 6.33$ | Yes |
| SafeFlowMatcher ($T^c$=16) | $1.634 \pm 0.002$ | 0.078 | 0% | $67.96 \pm 1.11$ | $89.29 \pm 0.96$ | Yes |
| SafeFlowMatcher ($T^c$=32) | $1.634 \pm 0.003$ | 0.155 | 0% | $67.48 \pm 1.09$ | $87.33 \pm 0.85$ | Yes |
| SafeFlowMatcher ($T^c$=64) | $1.633 \pm 0.003$ | 0.299 | 0% | $68.03 \pm 1.01$ | $89.09 \pm 0.73$ | Yes |
| SafeFlowMatcher ($T^c$=128) | $1.632 \pm 0.003$ | 0.617 | 0% | $69.72 \pm 0.98$ | $91.01 \pm 0.90$ | Yes |
| SafeFlowMatcher ($T^c$=256) | $1.632 \pm 0.003$ | 1.215 | 0% | $69.19 \pm 1.02$ | $91.90 \pm 0.77$ | Yes |

Table 11: **QP-Based CBF Solver: Computation time across correction horizons.** Comparison of RES-SafeDiffuser with a fixed sampling horizon $T = 256$ and SafeFlowMatcher for a fixed prediction horizon $T^p = 1$ and varying correction horizons $T^c \in \{4, 8, 16, 32, 64, 128, 256\}$ in Maze2D, when using the QP solver solution of CBF-QP. Each entry reports mean $\pm$ std over 100 rollouts.

| Method (QP CBF Solver) | Score ($\uparrow$) | T-Time (s) | Trap Rate | $\kappa$($\downarrow$) | $a$($\downarrow$) | BS1&BS2 ($\geq 0$) |
|---|---|---|---|---|---|---|
| RES-SafeDiffuser (Xiao et al., 2025) | $1.468 \pm 0.353$ | 9.998 | 85% | $76.06 \pm 38.73$ | $4776.45 \pm 2430.48$ | Yes |
| SafeFlowMatcher ($T^c$=4) | $1.606 \pm 0.029$ | 0.157 | 12% | $77.31 \pm 2.52$ | $276.02 \pm 39.01$ | Yes |
| SafeFlowMatcher ($T^c$=8) | $1.632 \pm 0.004$ | 0.315 | 4% | $76.93 \pm 1.11$ | $137.31 \pm 11.00$ | Yes |
| SafeFlowMatcher ($T^c$=16) | $1.634 \pm 0.003$ | 0.613 | 0% | $67.94 \pm 1.31$ | $118.72 \pm 9.65$ | Yes |
| SafeFlowMatcher ($T^c$=32) | $1.632 \pm 0.007$ | 1.247 | 0% | $68.54 \pm 1.22$ | $154.20 \pm 14.70$ | Yes |
| SafeFlowMatcher ($T^c$=64) | $1.632 \pm 0.003$ | 2.464 | 2% | $69.54 \pm 1.44$ | $158.70 \pm 23.97$ | Yes |
| SafeFlowMatcher ($T^c$=128) | $1.631 \pm 0.004$ | 4.892 | 11% | $70.59 \pm 1.46$ | $174.57 \pm 34.67$ | Yes |
| SafeFlowMatcher ($T^c$=256) | $1.630 \pm 0.004$ | 9.957 | 13% | $67.80 \pm 1.42$ | $183.47 \pm 28.39$ | Yes |

Across both the closed-form and QP-based CBF solvers, SafeFlowMatcher exhibits exceptionally low generation time (T-Time), even when the correction horizon is small. The tables show that Safe-FlowMatcher remains effective and safe over a wide range of $T^c$ values, whereas RES-SafeDiffuser is much slower and frequently suffers from severe local traps. At $T^c = 4$, SafeFlowMatcher may exhibit minor oscillations near constraint boundaries due to overcorrections caused by the small correction steps (see Figure 14). Although this falls under the definition of the local trap in 2, its impact is minimal, in contrast to SafeDiffuser, whose early safety enforcement often leads to hard traps and incomplete paths (see Figure 16).

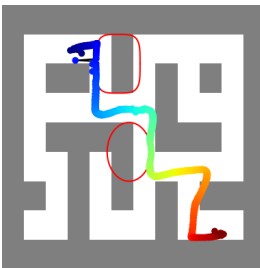

Figure 14: **Local trap of SafeFlowMatcher at $T^c = 4$.** Local traps observed at small correction horizons $T^c = 4$ in Maze2D. These traps manifest as mild boundary oscillations near obstacles, yet the path remains complete and reaches the goal. Unlike SafeDiffuser depicted in Figure 16, which often fails with incomplete paths under early safety enforcement, SafeFlowMatcher maintains path completeness despite minor oscillations.

### F.4   ENERGY-DISTANCE ANALYSIS OF DISTRIBUTIONAL DRIFT INDUCED BY CONTROL BARRIER FUNCTIONS

We quantify how much each perturbation $\Delta\mathbf{u}_t^k$ affects the generative process by measuring an energy distance between paths with and without safety intervention. For each model pair (FlowMatcher vs. SafeFlowMatcher, FM vs. SafeFM, Diffuser (Janner et al., 2022) vs. SafeDiffuser (Xiao et al., 2025)), we generate $N = 100$ paths from both the baseline and the corresponding safe variants, starting from the same initial conditions. We define the distance between two paths as the average waypoint-wise Euclidean distance

$$\delta(\boldsymbol{\tau}, \boldsymbol{\tau}') = \frac{1}{H+1} \sum_{k=0}^{H} \left\| \boldsymbol{\tau}^k - \boldsymbol{\tau'}^k \right\|_2.$$

Given $\{\boldsymbol{\tau}_{1,i}^{\text{base}}\}_{i=1}^N$ and $\{\boldsymbol{\tau}_{1,j}^{\text{safe}}\}_{j=1}^N$, where $\boldsymbol{\tau}_{1,(\cdot)}$ denotes the final generated path[6], the (sample) energy distance between the two path distributions is

$$\widehat{\mathcal{D}}_E = \frac{2}{N^2} \sum_{i=1}^{N} \sum_{j=1}^{N} \delta(\boldsymbol{\tau}_{1,i}^{\text{base}}, \boldsymbol{\tau}_{1,j}^{\text{safe}}) - \frac{1}{N^2} \sum_{i=1}^{N} \sum_{j=1}^{N} \delta(\boldsymbol{\tau}_{1,i}^{\text{base}}, \boldsymbol{\tau}_{1,j}^{\text{base}}) - \frac{1}{N^2} \sum_{i=1}^{N} \sum_{j=1}^{N} \delta(\boldsymbol{\tau}_{1,i}^{\text{safe}}, \boldsymbol{\tau}_{1,j}^{\text{safe}}).$$

Larger values indicate stronger distributional drift between the baseline and safe path distributions. For each waypoint $k$, we similarly define a per-waypoint energy distance $\widehat{\mathcal{D}}_E^k$ by replacing $\delta(\boldsymbol{\tau}, \boldsymbol{\tau}')$ with $\delta^k(\boldsymbol{\tau}, \boldsymbol{\tau}') = \|\boldsymbol{\tau}^k - \boldsymbol{\tau}^{k'}\|_2$ in the above definition of $\widehat{\mathcal{D}}_E$.

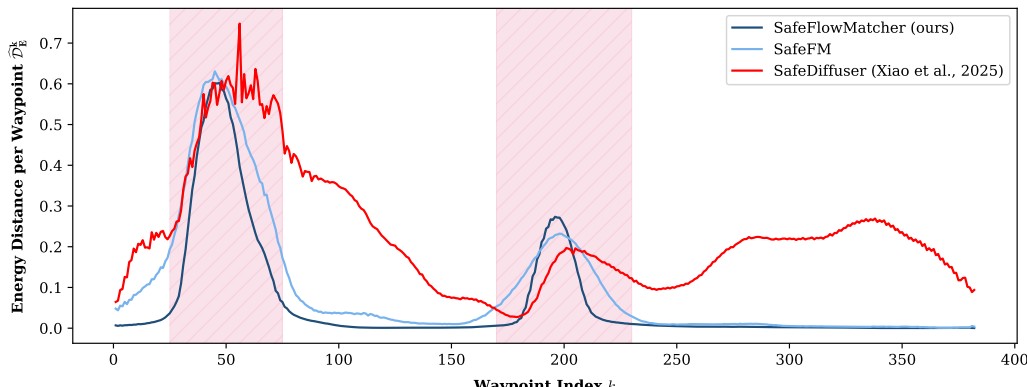

Figure 15: **Per-waypoint drift between baseline and safe path.** For each model pair (Flow-Matcher/SafeFlowMatcher, FM/SafeFM, Diffuser/SafeDiffuser), the plot shows the mean per-waypoint deviation between paths produced by the baseline and its corresponding safe variant. The pink band marks the region where the baseline path violates the CBF constraint at the final time; for clarity, a single common band is shown, although the exact violation interval may differ by several steps across models.

Figure 15 plots the per-waypoint energy distance $\widehat{\mathcal{D}}_E^k$ between the baseline and safe paths. Across the three model pairs, the resulting energy distances $\widehat{\mathcal{D}}_E$ are 0.061 (SafeFlowMatcher), 0.097 (SafeFM), and 0.229 (SafeDiffuser), showing that SafeFlowMatcher induces the smallest distributional drift while still enforcing safety. The pink band indicates the segment in which the baseline path violates the CBF constraint at the final time. Outside this safety-critical region, SafeDiffuser shows large drift, and SafeFM still exhibits noticeable spillover, suggesting that their safety interventions propagate to parts of the path that do not require correction. In contrast, SafeFlowMatcher keeps the drift close to zero outside the pink band.

In SafeFM and SafeDiffuser, the perturbation is applied not only at $t = 1$, but also to intermediate generative states $\boldsymbol{\tau}_t^k$ for $t \in [0, 1)$ that are never executed. Once these perturbed intermediate states

---

[6]For diffusion-based samplers, the final path is obtained at $t = 0$ rather than $t = 1$, but we use the unified notation $\boldsymbol{\tau}_{1,(\cdot)}$ for consistency.

are fed back into the velocity field and integrated forward, the resulting deviations can accumulate and propagate through the generative dynamics, producing drift at waypoints far outside the final-time violation interval. However, PC integrator in SafeFlowMatcher naturally separate correction from the prediction. It can mitigate this kind of drift effectively.

## F.5 Visualization and Qualitative Analysis of Local Traps

Following the Definition 2, a path is locally trapped if the corrected waypoint exhibits a large discontinuity between two successive corrected path:

$$\|\boldsymbol{\tau}_1^k - \boldsymbol{\tau}_1^{k-1}\| > \zeta,$$

for some threshold $\zeta > 0$. Intuitively, this corresponds to path that get stuck near safety boundaries and consequently produce a large *jump* to escape, often resulting in incomplete paths.

**SafeDiffuser** applies the CBF constraint to each waypoint at every sampling step, starting from pure noise. Because the initial waypoints are sampled i.i.d., neighboring waypoints $\boldsymbol{\tau}_t^k$ and $\boldsymbol{\tau}_t^{k-1}$ often differ significantly. Since CBFs depend on the waypoint, such large discrepancies cause the resulting CBF corrections to vary greatly across waypoints. Although the diffuser aims to generate a continuous path (i.e., $\|\boldsymbol{\tau}_t^k - \boldsymbol{\tau}_t^{k-1}\| \le \zeta$), applying the CBF constraint independently at each waypoint can break this continuity, pushing different waypoints toward different constraint boundaries and creating local traps. This behavior is clearly visualized in Figures 16 and 17.

**SafeFlowMatcher**, in contrast, begins the correction phase from a semi-continuous path $\boldsymbol{\tau}_0^c$ (i.e., $\|\boldsymbol{\tau}_0^{c,k} - \boldsymbol{\tau}_0^{c,k-1}\| \le \eta$ for some small $\eta \ge \zeta$). Because neighboring waypoints are already close to each other, the resulting CBF corrections vary smoothly across the path. This keeps all waypoints moving in a consistent direction, preserving the path's continuity and preventing local traps. As visualized in Figures 18 and 19, the path maintains forward progress without stalling.

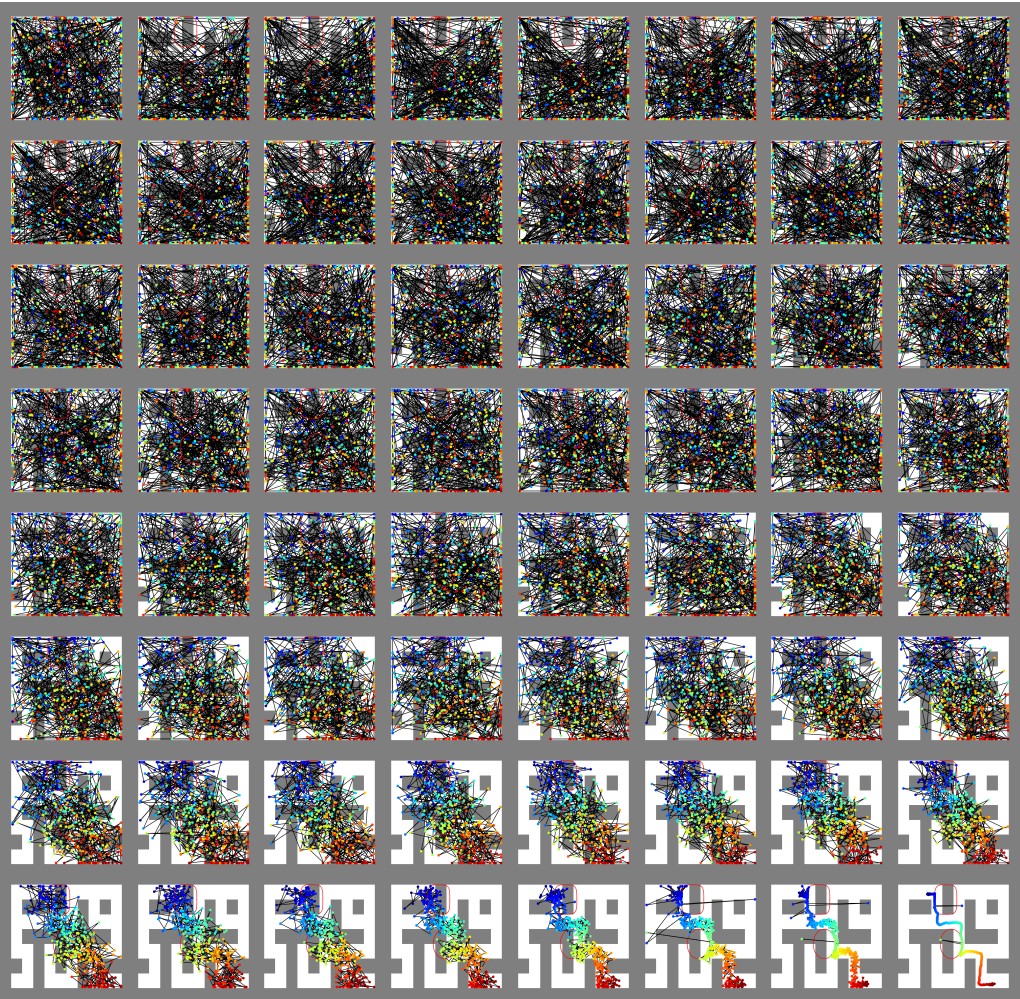

Figure 16: **Path Generation Process of SafeDiffuser (Xiao et al., 2025) in Maze2D environment with two constraints.** From the top-left to the bottom-right, we visualize $\tau_t$ on a uniform time discretization of $[T, 0]$, excluding the midpoint $t = 0.5T$.

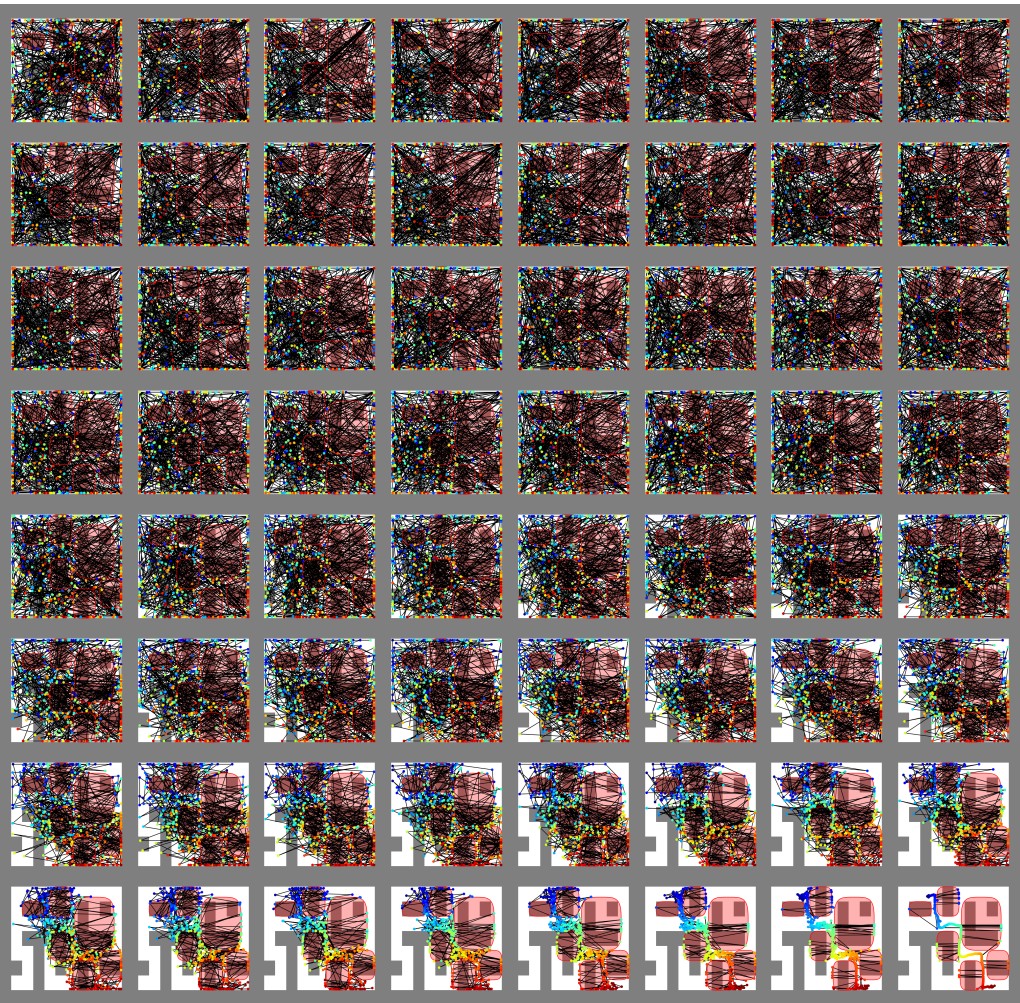

Figure 17: **Path Generation Process of SafeDiffuser (Xiao et al., 2025) in Maze2D environment with six constraints.** From the top-left to the bottom-right, we visualize $\tau_t$ on a uniform time discretization of $[T, 0]$, excluding the midpoint $t = 0.5T$.

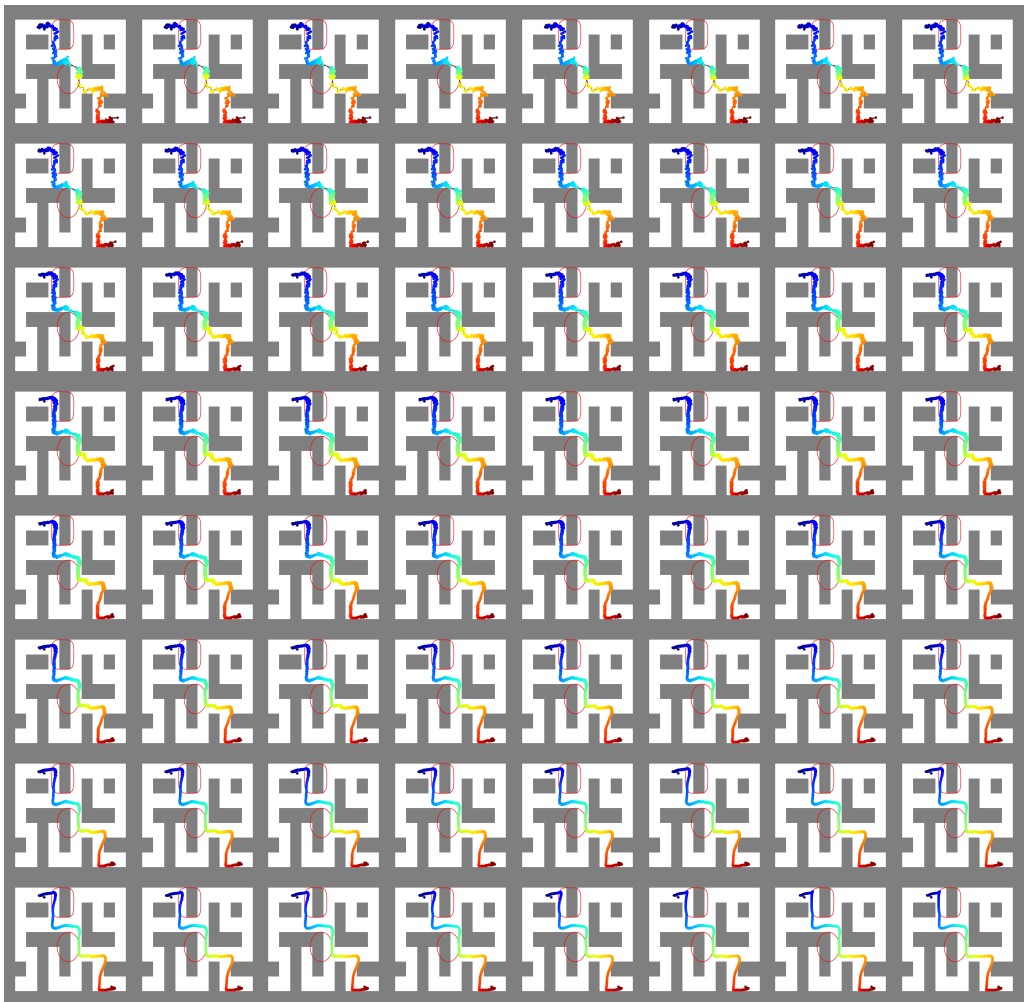

Figure 18: **Path Generation Process of SafeFlowMatcher in Maze2D environment with two constraints.** Top-left presents the predicted path $\tau_1^p = \tau_0^c$ from a noise sample. From the top-left to the bottom-right, we visualize $\tau_t^c$ on a uniform time discretization of $[0, 1]$, excluding the midpoint $t = 0.5$.

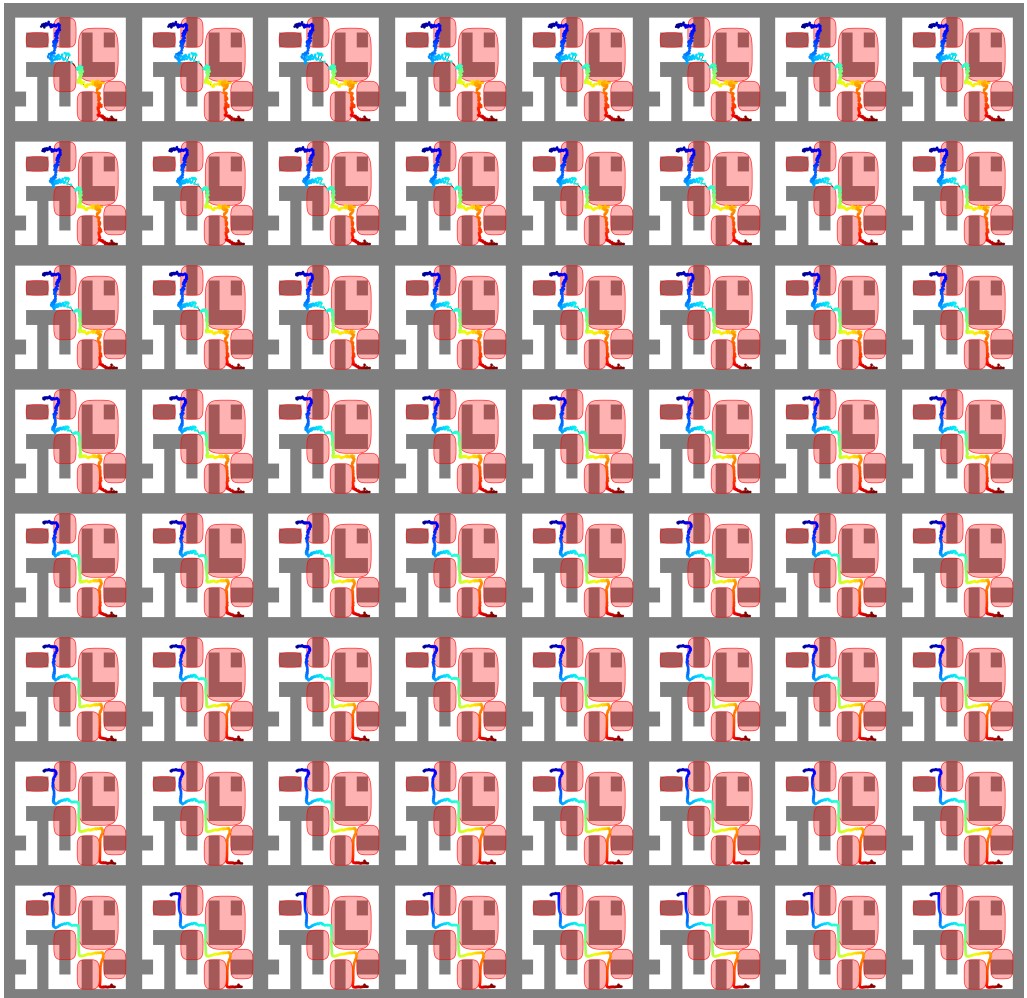

Figure 19: **Path Generation Process of SafeFlowMatcher in Maze2D environment with six constraints.** Top-left presents the predicted path $\tau_1^p = \tau_0^c$ from a noise sample. From the top-left to the bottom-right, we visualize $\tau_t^c$ on a uniform time discretization of $[0, 1]$, excluding the midpoint $t = 0.5$.

