# OpenReview forum: "SafeFlowMatcher: Safe and Fast Planning using Flow Matching with Control Barrier Functions"
_ICLR.cc/2026/Conference — ICLR 2026 Poster_

### Official Review · Reviewer_DQsF · 2025-10-15

[review text omitted: it was posted to a different submission]

---

> ### Author Response · Authors · 2025-11-12
>
> For administrative clarity: one of the reviews attached to this submission appears to have been **swapped (cross-assigned)** with a review intended for **“Practical Diffusion Planning via Temperature-Guided Reward Conditioning (Paper ID: 24685)**.
>
> Conversely, the TGDP forum contains a review that refers to **SafeFlowMatcher (Paper ID: 24026)**, based on the publicly visible discussion.
>
> We will also contact the AC to request a proper reassignment.
>
> Many thanks for your understanding.

---

> > ### Comment · Reviewer_DQsF · 2025-11-13
> > **Correction of comments**
> >
> > Sorry for the misuploaded materials. I have updated the comments here.

---

> > > ### Author Response · Authors · 2025-11-13
> > >
> > > We thank the reviewer for the detailed feedback. Before proceeding with the rebuttal, we would like to briefly clarify the last two points in the weakness section
> > > - "The temperature-based reweighting is..."
> > > - "The paper’s computational trade-offs (≈1.5× CFG cost) are..."
> > >
> > > do not appear in our paper **SafeFlowMatcher (ID 24026)**. These concepts seem to belong to **Practical Diffusion Planning via Temperature-Guided Reward Conditioning (Paper ID: 24685)**, which includes temperature-guided mechanisms.
> > > We understand such mix-ups can happen and simply wish to ensure the evaluation reflects the content of our submission. If these comments were intended to refer to specific parts of our paper, we would be happy to clarify further.
> > >
> > > Thank you for your thoughtful review.

---

> > > > ### Comment · Reviewer_DQsF · 2025-11-13
> > > >
> > > > I am so sorry for the problem, and thanks for your understanding. ICLR has indeed had many submissions this year, and I am sorry to have messed up with too many reviewed papers.

---

> ### Author Response · Authors · 2025-11-24
> **Response1 to Reviewer DQsF**
>
> **W1.** We agree that our current paper relies on known, differentiable, and well-calibrated safety sets $b(x)$. However, this limitation is not specific to SafeFlowMatcher but shared by essentially all CBF-based safety filters. Importantly, recent CBF literature shows that such filters do not require perfectly known, smooth, or hand-crafted barriers: discrete-time and duality-based CBFs handle nonsmooth and nonconvex obstacles [1, 2]; measurement-robust and probabilistic CBFs explicitly account for perception and state-estimation uncertainty [3, 4]; and dynamic CBFs incorporate obstacle motion prediction [5]. In addition, learning-based CBFs can construct safety certificates directly from data (e.g., RGB-D [6], LiDAR [7, 8, 9], or demonstrations [10, 11]), implicitly representing complex safety sets. While in this paper we use hand-crafted CBFs for clarity, SafeFlowMatcher's correction phase only requires evaluating a barrier constraint, making our framework directly compatible with these robust and learned CBF formulations in more realistic settings.
>
> **W2.** We address this in our answer to the reviewer's Q2.
>
> **W3.** We address this in our answer to the reviewer's Q6.
>
> **W4.** Lemma 2 and Lemma 3 imply that the vanishing time-scaling suppresses sampling error and mitigates potential drift. Because the prediction phase already brings the path very close to the target path, the resulting distribution is preserved. In our new Appendix B.1, we have also added empirical justification for the assumptions (imposed on the prediction error) used by Lemmas 2 and 3. This illustrates the benefit of the PC integrator: unlike methods that perturb the entire sampling process from the beginning, our correction is applied near the target, so any drifts that may occur are inherently small.
>
> **W5.** We address this in our answer to the reviewer's Q5.
>
> [1] Ayush Agrawal and Koushil Sreenath. Discrete control barrier functions for safety-critical control of discrete systems with application to bipedal robot navigation. In Robotics: Science and Systems, volume 13, pp. 1–10. Cambridge, MA, USA, 2017.
> [2] Qiayuan Liao, Zhongyu Li, Akshay Thirugnanam, Jun Zeng, and Koushil Sreenath. Walking in narrow spaces: Safety-critical locomotion control for quadrupedal robots with duality-based optimization. In 2023 IEEE/RSJ International Conference on Intelligent Robots and Systems (IROS), pp. 2723–2730. IEEE, 2023.
> [3] Ryan K Cosner, Andrew W Singletary, Andrew J Taylor, Tamas G Molnar, Katherine L Bouman, and Aaron D Ames. Measurement-robust control barrier functions: Certainty in safety with uncertainty in state. In 2021 IEEE/RSJ International Conference on Intelligent Robots and Systems
> (IROS), pp. 6286–6291. IEEE, 2021.
> [4] Kehan Long, Vikas Dhiman, Melvin Leok, Jorge Cort´es, and Nikolay Atanasov. Safe control synthesis with uncertain dynamics and constraints. IEEE Robotics and Automation Letters, 7(3): 7295–7302, 2022.
> [5] Zhuozhu Jian, Zihong Yan, Xuanang Lei, Zihong Lu, Bin Lan, Xueqian Wang, and Bin Liang. Dynamic control barrier function-based model predictive control to safety-critical obstacle-avoidance of mobile robot. In 2023 IEEE International Conference on Robotics and Automation
> (ICRA), pp. 3679–3685. IEEE, 2023.
> [6] Hossein Abdi, Golnaz Raja, and Reza Ghabcheloo. Safe control using vision-based control barrier function (v-cbf). In IEEE International Conference on Robotics and Automation, pp. 782–788. IEEE, 2023.
> [7] Mohit Srinivasan, Amogh Dabholkar, Samuel Coogan, and Patricio A Vela. Synthesis of control barrier functions using a supervised machine learning approach. In 2020 IEEE/RSJ International Conference on Intelligent Robots and Systems (IROS), pp. 7139–7145. IEEE, 2020b.
> [8] Kehan Long, Cheng Qian, Jorge Cort´es, and Nikolay Atanasov. Learning barrier functions with memory for robust safe navigation. IEEE Robotics and Automation Letters, 6(3):4931–4938, 2021.
> [9] Marvin Harms, Mihir Kulkarni, Nikhil Khedekar, Martin Jacquet, and Kostas Alexis. Neural control barrier functions for safe navigation. In 2024 IEEE/RSJ International Conference on Intelligent Robots and Systems (IROS), pp. 10415–10422. IEEE, 2024.
> [10] Alexander Robey, Haimin Hu, Lars Lindemann, Hanwen Zhang, Dimos V Dimarogonas, Stephen Tu, and Nikolai Matni. Learning control barrier functions from expert demonstrations. In 2020 59th IEEE Conference on Decision and Control (CDC), pp. 3717–3724. IEEE, 2020.
> [11] Lars Lindemann, Alexander Robey, Lejun Jiang, Satyajeet Das, Stephen Tu, and Nikolai Matni. Learning robust output control barrier functions from safe expert demonstrations. IEEE Open Journal of Control Systems, 3:158–172, 2024.

---

> ### Author Response · Authors · 2025-11-24
> **Response2 to Reviewer DQsF**
>
> **Q1.** The primary bottlenecks of SafeDiffuser are local traps and end-to-end wall-clock time (referred to as T-time in the paper). These issues undermine two critical objectives in path planning: path completeness and real-time feasibility. Specifically, Table 1 and Section 4.1 describe that SafeDiffuser exhibits trap rates ranging from 69% to 100%, with wall-clock times of 1.208s when using a closed-form solver for CBFs and 9.998s when using a QP solver.
>
> To address these bottlenecks, we aimed to generate paths that are 100% complete (no local traps) while achieving safety and the fastest possible speed. In Table 1, we demonstrate that SafeFlowMatcher can generate a high-path-quality (smooth) path without any local traps. Furthermore, in the newly added Appendix F.3, SafeFlowMatcher can accelerate its computation time by reducing the correction horizon $T^c$, while maintaining high performance. Specifically, when $T^p = 1$ and $T^c = 4$, SafeFlowMatcher generates a safe path within 23 ms, which is $50\times$ faster than SafeDiffuser.
>
> **Q2.** We first clarify that, under the assumptions of our analysis, the QP in Eq.(17) is feasible for a broad and practically attainable range of hyperparameters. The constraint in Eq.(15) contains a nonnegative slack variable $r_t^k \ge 0$ with a strictly positive weight $w_t^k > 0$ for all $t < t_w$. Hence, for $t < t_w$ the constraint set is always nonempty, since any violation of the barrier inequality can be absorbed by the slack variable.
>
> For $t \ge t_w$, the relaxation term vanishes ($w_t^k = 0$) and the condition reduces to the finite-time CBF inequality. Proposition 1 yields an upper bound
> $$
> T \le t_w + \frac{(\delta - b(\tau_{t_w}^{c,k}))^{1-\rho}}{\epsilon(1-\rho)}.
> $$
> By selecting $(\epsilon,\rho)$ so that the right-hand side is at most $1$, we guarantee that the path enters the robust safe set $C_\delta$ within the time interval $[0,1]$. The prediction phase is crucial here: because it already moves $\tau^c_0$ (and thus $\tau^c_{t_w}$) close to the target path, $b(\tau^{c,k}_{t_w})$ is typically close to $\delta$, which significantly enlarges the admissible region of $(\epsilon,\rho)$. In contrast, methods without the PC integrator have much more negative $b(\tau_0)$, making this condition harder to satisfy.
> In addition to theoretical analysis, we empirically validate this in newly added Appendix F.2 by comparing SafeFlowMatcher (PC integrator) and SafeFM (without PC integrator) across extensive $(\epsilon,\rho)$. The result shows that SafeFlowMatcher is robust to hyperparameter selection and not sensitive.
>
> Additionally, in practice, we choose $(\epsilon,\rho)$ based on a worst-case value of $b$. Moreover, we consistently observe the feasibility of SafeFlowMatcher across all experiments.
>
> **Q3.** First, we clarify that we do not use a class-$\mathcal{K}$ function in the CBF sense. The symbol $\alpha$ in our paper refers only to the constant scaling factor in the vanishing time-scaled flow dynamics defined in Eq.(10). Thus, no class-$\mathcal{K}$ selection is required in our framework.
>
> Lemma 3 provides intuition on the role of $\alpha$: larger $\alpha$ makes the vanishing time-scaled dynamics contract the prediction error more aggressively, while the $(1 - t)$ factor stabilizes the behavior near $t = 1$. In our revised manuscript, we updated Table 3 and added new Figure 6 to effectively show the effect of $\alpha$.
> Empirically, we observe that larger $\alpha$ reduces curvature and acceleration of the paths, but excessively large $\alpha$ causes noticeable distortion near $t = 1$, as shown in Figure 6.
>
> Practically, we find that SafeFlowMatcher is robust over a broad range of $\alpha$ values, as shown in Table 3. Thus, our tuning guideline is to start from $\alpha = 1$ and increase it until just before the sharp distortion in the path near $t = 1$ becomes visible, as in Figure 6. Further details are provided in Section 4.2: *Effect of Vanishing Time-Scale*.
>
> **Q4.** We appreciate the reviewer’s insightful comment regarding distributional drift. While standard metrics like FID are challenging to apply due to the lack of a ground-truth reference distribution $q$ in our setting, we agree that quantifying deviation from the nominal prior is essential. We find your suggestion of using energy distance to be an excellent solution, as it allows us to measure the drift between nominal and corrected paths without needing external labels.
> We are currently computing the energy distance for all baselines and variants of SafeFlowMatcher.
> We uploaded this comment in advance to adhere to the ICLR rebuttal timeline, but will update it and the paper PDF as soon as the experiments are completed.

---

> ### Author Response · Authors · 2025-11-24
> **Response3 to Reviewer DQsF**
>
> **Q5.** We appreciate this suggestion and agree that, in principle, such baselines would help us more clearly separate the effect of the backbone from the effect of the PC integrator and the vanishing-time mechanisms.
>
> However, implementing all of the proposed baselines is beyond the scope of the current work:
> - **MPC with CBF-CLF-QP:**
>   Our current experimental domains (Maze2D, locomotion, and robot manipulations) follow the offline path-generation setting used in Diffuser and SafeDiffuser, where one learns from a fixed dataset of paths and does not assume access to a transition model or a task reward at test time.
>   MPC requires an explicit dynamical robot model which is unavailable in our scope.
>
> - **Neural-ODE/CNF + CBF filter:**
>   In the SafeDiffuser paper [12], Appendix C.1 cites results from *On the Forward Invariance of Neural ODEs* [13], already highlighting the limitations of ODE-based planners when combined with CBFs.
>
> - **Diffusion with single-shot CBF:**
>   The PC integrator and the vanishing time-scaling are designed specifically for ODE-based generative dynamics. In contrast, diffusion models operate through discrete reverse-diffusion updates with significant stochasticity and typically require many sampling steps. Since these updates are not governed by continuous-time ODEs and often involve non-smooth score evaluations, applying the PC integrator is not meaningful: the “few-step’’ contraction mechanism cannot be realized, and the vanishing time-scaling cannot be defined.
>   For these reasons, attaching the PC integrator to diffusion backbones would not provide a fair or informative comparison, as the key mechanism of SafeFlowMatcher does not transfer to such models.
>
> **Q6.** Our current theoretical guarantees are derived in an idealized setting: we assume exact state information, static safety constraints, and no delays. In particular, we do not model state-estimation errors, dynamics mismatch, or moving obstacles in the analysis, and we do not provide an ISS- or chance-constrained treatment in the present paper.
>
> However, we remark that such extensions are possible, as this is one of the main benefits of our prediction–correction decoupling: it ensures that the learned vector field remains agnostic to safety interventions, allowing seamless adaptation to changes in the safety criteria. The use of a robust safe set $C_\delta$ already introduces a safety margin that can, in principle, buffer small perturbations (e.g., perception noise).
> We acknowledge that a rigorous robustness analysis would require extending our framework to stochastic or robust CBF formulations, which we view as an interesting direction for future work.
>
> [12] Xiao Wei, Wang Tsun-Hsuan, Gan Chuang, Hasani Ramin, Lechner Mathias, and Rus Daniela. Safediffuser: Safe planning with diffusion probabilistic models. IEEE, 2025.
> [13] Wei Xiao, Tsun-Hsuan Wang, Ramin Hasani, Mathias Lechner, Yutong Ban, Chuang Gan, and Daniela Rus. On the forward invariance of neural odes. In International conference on machine learning, pp. 38100–38124. PMLR, 2023.

---

> > ### Comment · Reviewer_DQsF · 2025-11-24
> >
> > I have read the authors' Rebuttal and find their responses to be **highly satisfactory** and **convincing**.
> >
> > The paper presents a **theoretically strong and novel framework, SafeFlowMatcher**, which effectively combines the efficiency of Flow Matching with CBF-certified safety. The core innovation of the **Prediction-Correction (PC) integrator** successfully resolves major planning challenges, specifically **local traps** and **distributional drift**.
> >
> > The Rebuttal clearly demonstrates that SafeFlowMatcher achieves superior performance, showing **zero local traps** and a **40x speedup** over baselines in real-time safety planning. Furthermore, the authors provided excellent theoretical justification for the **robust feasibility** of the CBF-QP and the **modularity** of the framework to accommodate advanced CBF formulations in the future.

---

> ### Author Response · Authors · 2025-11-29
> **Updated Response to Q4 from Reviewer DQsF — Distributional Drift Induced by Control Barrier Functions**
>
> **Q4.** *To what extent do repeated corrections induce distributional drift?*
>
> We have now added a quantitative analysis of distributional drift induced by control barrier functions using **energy distance** (Appendix F.4; Fig. 15).
> For each model pair (FlowMatcher/SafeFlowMatcher, FM/SafeFM, Diffuser/SafeDiffuser), we generate 100 matched rollouts and compute path-level energy distances:
>
> - **SafeFlowMatcher:** **0.061**
> - **SafeFM:** **0.097**
> - **SafeDiffuser:** **0.229**
>
> Thus, **SafeFlowMatcher introduces the smallest drift**, owing to (i) the prediction–correction integrator, which avoids perturbing intermediate latent states, and (ii) vanishing time-scaled refinement that confines corrections to safety-critical regions.
>
> As shown in Fig. 15, SafeFlowMatcher’s per-waypoint drift remains localized near the constraint-violating segment, whereas SafeFM and SafeDiffuser exhibit drift across large portions of the path.
>
> We have integrated these results into the revised manuscript.

---

> ### Author Response · Authors · 2025-11-29
> **Clarification Regarding Reverted Official Comments of Reviewer DQsF**
>
> Dear Chairs,
>
> Due to the recent system-wide revert of official comments, the currently visible “Official Comment” on our submission (Paper ID: 24026) is not aligned with the actual content of our paper and does not represent the correct review.
>
> Previously, **there was a review misassignment between Review of Submission 24026 (Ours) by DQsF and Review of Submission 24685 (Practical Diffusion Planning via Temperature-Guided Reward Conditioning) by 1nMT**, and the reviewer corrected this by uploading a revised review. However, after the revert, the discussion thread shows an earlier, outdated version again.
>
> For clarity, the correct and latest evaluation is the most recent revision of
> “Official Review of Submission 24026 by Reviewer DQsF”
> (i.e., the last revision in the review’s revision history). The reviewer made multiple revisions while fixing the earlier mix-up, and the final revision should be regarded as the authoritative version.
>
> This latest revision contains(and this is exactly the same as the review of Submission 24685 by 1nMT):
> - a Summary,
> - Soundness (4),
> - Presentation (3),
> - Contribution (3),
> - five Strengths,
> - five Weaknesses,
> - six Questions,
> - Rating (8),
> - Confidence (4),
> - and Code of Conduct (Yes).
>
> We would be grateful if this most recent revision could be treated as the valid review for Submission 24026. Please let us know if any further clarification would be helpful.
>
> Thank you very much for your time and for managing the process under the current circumstances.

---

### Official Review · Reviewer_gaEu · 2025-10-29

**Soundness:** 3
**Presentation:** 3
**Contribution:** 2
**Rating:** 4
**Confidence:** 5

**Summary:**

This paper introduces SafeFlowMatcher, a novel planning framework that integrates Flow Matching (FM) with Control Barrier Functions (CBFs) to achieve both real-time efficiency and certified safety in robotic path planning. The core of SafeFlowMatcher is a two-phase Prediction-Correction (PC) integrator. The prediction phase efficiently generates a candidate path using learned FM dynamics without safety interventions. The subsequent correction phase refines this path by (i) reducing integration errors through vanishing time-scaled flow dynamics (VTFD) and (ii) enforcing hard safety constraints using a CBF-based quadratic program (QP), minimally perturbing the vector field. This decoupling strategy prevents distributional drift and mitigates "local trap" problems often encountered by other certification-based generative planners. Extensive experiments on Maze2D navigation and high-dimensional locomotion tasks demonstrate that SafeFlowMatcher outperforms existing diffusion- and FM-based baselines in terms of safety (zero trap rate), planning performance (higher scores), path quality (smoother paths), and computational efficiency. Ablation studies confirm the necessity of both prediction and correction phases and the effectiveness of VTFD.

**Strengths:**

1. The paper proposes a unique combination of Flow Matching for efficient path generation and Control Barrier Functions for formal safety guarantees. This addresses a critical need in robotic planning for both speed and reliability. The two-phase PC integrator is a significant contribution. By separating the initial path generation (prediction) from safety enforcement (correction), SafeFlowMatcher avoids issues like distributional drift and local traps that plague other methods that attempt to enforce safety on intermediate latent states.
2.  The paper provides rigorous mathematical proofs for the forward invariance of a robust safe set and finite-time convergence to this set. This theoretical backing strengthens the credibility of the proposed safety guarantees.
3. By leveraging Flow Matching, SafeFlowMatcher demonstrates efficiency comparable to or better than diffusion-based methods, which often require many more sampling steps. The ability to achieve high performance with a small number of function evaluations (e.g., $T^p=1$) is a notable advantage.

**Weaknesses:**

1.  The paper highlights the reliance on well-defined CBFs. Could the authors elaborate on the practical challenges and potential solutions for designing or learning CBFs for more complex, dynamic, or unstructured real-world robotic environments beyond the relatively simple constraints presented? Are there any ongoing efforts to integrate CBF learning within SafeFlowMatcher?
2. Given that hyperparameter tuning is identified as future work, can the authors provide more insight into the sensitivity of SafeFlowMatcher's performance to the choice of $\epsilon$, $\rho$, $\delta$, $\alpha$, and $t_w$? Are there any heuristics or adaptive strategies that could be employed to simplify this tuning process for new tasks or environments?
3. While closed-form QP solutions are mentioned for simple cases, how does the computational overhead of the QP solver scale with a larger number of waypoints ($H$) or a more extensive set of complex safety constraints? Have the authors encountered scenarios where the QP solving time becomes a bottleneck, and what strategies could mitigate this?
4. The paper mentions developing a guidance-free version as future work. Could the authors expand on the specific challenges in achieving this (e.g., in terms of exploration, multimodal goal reaching) and their initial thoughts on how to approach such a development?
5. Lemma 1 assumes a symmetric, zero-mean distribution for the prediction error $\epsilon$. How robust is SafeFlowMatcher to deviations from this assumption, particularly if the prediction errors are biased or exhibit heavy tails? What are the practical implications of such deviations on safety guarantees and performance?
6. While Naive Safe FM serves as a baseline, could the authors provide a more in-depth discussion on why directly applying safety constraints from the beginning leads to a high trap rate? A clearer theoretical or empirical breakdown of the failure modes of Naive Safe FM would further strengthen the argument for the PC integrator.
7. For the vanishing time-scaled flow dynamics, the optimal $\alpha=2$ was found empirically. Is there a deeper theoretical justification for this specific value or for the general range of $\alpha$ that provides a good trade-off between accelerated convergence and stability?
8. The current experimental validation focuses on maze navigation and locomotion. How would SafeFlowMatcher perform on more complex robotic manipulation tasks (e.g., pick-and-place with moving obstacles) that typically involve higher-dimensional state and action spaces, more intricate safety constraints, and potentially longer planning horizons?

**Questions:**

See Weaknesses^

---

> ### Author Response · Authors · 2025-11-24
> **Response1 to Reviewer gaEu**
>
> **W1.** We agree that SafeFlowMatcher currently relies on well-defined CBFs, and the design of such barrier functions becomes more challenging in complex or unstructured real-world environments. This is a limitation shared by most certification-based safe planning methods. In practice, several approaches can mitigate this issue: for example, CBF methods for non-convex and non-smooth safe sets [1, 2], as well as recent work on learning-based CBFs [3, 4, 5, 6, 7]; see our revised Appendix A for a more detailed survey.
> We note that SafeFlowMatcher is compatible with these extensions because the CBF enters only as a constraint within the correction phase. Also, it is common to extend SafeFlowMatcher to dynamic environments (e.g., obstacles for collision avoidance) by using receding-horizon control or dynamic CBFs [8]. Formally incorporating data-driven CBFs within SafeFlowMatcher or extending SafeFlowMatcher to dynamic environments are promising directions for future work.
> To demonstrate that the current SafeFlowMatcher pipeline can be applied to more complex environments, we have revised the paper to include additional robot locomotion experiments, such as block-stacking with the Kuka robot arm environment.
>
> [1] Ayush Agrawal and Koushil Sreenath. Discrete control barrier functions for safety-critical control of discrete systems with application to bipedal robot navigation. In Robotics: Science and Systems, volume 13, pp. 1–10. Cambridge, MA, USA, 2017.
> [2] Qiayuan Liao, Zhongyu Li, Akshay Thirugnanam, Jun Zeng, and Koushil Sreenath. Walking in narrow spaces: Safety-critical locomotion control for quadrupedal robots with duality-based optimization. In 2023 IEEE/RSJ International Conference on Intelligent Robots and Systems (IROS), pp. 2723–2730. IEEE, 2023.
> [3] Hossein Abdi, Golnaz Raja, and Reza Ghabcheloo. Safe control using vision-based control barrier function (v-cbf). In IEEE International Conference on Robotics and Automation, pp. 782–788. IEEE, 2023.
> [4] Mohit Srinivasan, Amogh Dabholkar, Samuel Coogan, and Patricio A Vela. Synthesis of control barrier functions using a supervised machine learning approach. In 2020 IEEE/RSJ International Conference on Intelligent Robots and Systems (IROS), pp. 7139–7145. IEEE, 2020b.
> [5] Kehan Long, Cheng Qian, Jorge Cort´es, and Nikolay Atanasov. Learning barrier functions with memory for robust safe navigation. IEEE Robotics and Automation Letters, 6(3):4931–4938, 2021.
> [6] Marvin Harms, Mihir Kulkarni, Nikhil Khedekar, Martin Jacquet, and Kostas Alexis. Neural control barrier functions for safe navigation. In 2024 IEEE/RSJ International Conference on Intelligent Robots and Systems (IROS), pp. 10415–10422. IEEE, 2024.
> [7] Alexander Robey, Haimin Hu, Lars Lindemann, Hanwen Zhang, Dimos V Dimarogonas, Stephen Tu, and Nikolai Matni. Learning control barrier functions from expert demonstrations. In 2020 59th IEEE Conference on Decision and Control (CDC), pp. 3717–3724. IEEE, 2020.
> [8] Zhuozhu Jian, Zihong Yan, Xuanang Lei, Zihong Lu, Bin Lan, Xueqian Wang, and Bin Liang. Dynamic control barrier function-based model predictive control to safety-critical obstacle-avoidance of mobile robot. In 2023 IEEE International Conference on Robotics and Automation
> (ICRA), pp. 3679–3685. IEEE, 2023.

---

> ### Author Response · Authors · 2025-11-24
> **Response2 to Reviewer gaEu**
>
> **W2.** SafeFlowMatcher is robust to a broad range of hyperparameters, mainly because the prediction phase already brings the initial path close to a near-target path. The parameter $\delta$ sets the robust safety margin; it is problem-specific but not particularly sensitive. The parameters $\varepsilon$ and $\rho$ control finite-time convergence: larger $\rho$ and smaller $\varepsilon$ enforce faster convergence to the safe set. The parameter $t_w$ determines when the relaxation (slack) is turned off; a larger $t_w$ allows more gradual convergence, but we require $t_w < 1$. While relaxation can help improve path quality (e.g., score, curvature, and acceleration) and reduce local traps, SafeFlowMatcher can also operate without relaxation and still outperforms other methods. To show this more clearly, we have added SafeFlowMatcher without relaxation to Table 1.
>
> Moreover, to study the sensitivity of $(\varepsilon, \rho)$, we conduct an ablation over $\rho \in$ {0.1, 0.3, 0.5, 0.7, 0.9} and $\varepsilon \in$ {0.25, 0.50, 1.00, 2.50, 5.00, 10.00} in Appendix F.2. The results show that the PC integrator yields a broad feasible region, making hyperparameter selection easier in practice. We discuss $\alpha$ in our answer to the reviewer's 7th comment.
>
> **W3.** We agree that QP correction introduces computational overhead. This is a known limitation of safety-filter-based generative planners (e.g., SafeDiffuser), but SafeFlowMatcher is designed specifically to mitigate this via the PC integrator. Because the prediction phase already produces a path close to the target, the correction phase typically only needs to compensate a small prediction error (see Figure 4, where SafeFlowMatcher attains high performance even with a small sampling horizon $T$). Thus, across all our experiments, the QP in Eq.(17) is small, and we did not observe any related bottlenecks. For example, in Maze2D, for a path with $H = 384$ waypoints, the closed-form solution requires only 1.14 ms to compute on average, while the QP solver takes 33.21 ms on average. Of course, like many other path-planning algorithms, increasing $H$ would scale the cost proportionally.
>
> In our revisions, we also added a more challenging scenario with many obstacles (six CBF constraints) in Appendix F.1. As expected, the computation time increases with the number of constraints, but this overhead can be effectively mitigated by decreasing the correction horizon $T_c$. Detailed results are reported in Appendix F.1.
>
> **W4.** Our PC integrator is based on the standard FM formula $v_t(x)$ (without condition) [9], but for general robotics tasks, we used at least the initial (or current) observation as a condition $c$, so it became a conditioned FM vector field $v_t(x \mid c)$. At the time of our first submission, we did not empirically verify the PC integrator in the conditioned case. We were left with concerns about generalization to truly unconditional domains like image generation, where challenges such as insufficient exploration (leading to mode collapse) and poor multimodal distribution capture might arise without guiding signals.
>
> We began to address some of these concerns in our revisions. Our new supplementary materials (in *“CIFAR10_Uncond_Verify.pdf”*) include preliminary CIFAR-10 experiments, showing that simply attaching the PC integrator led to severe instability, resulting in poor image generation. Because this line of future work is better understood together with the context provided in the supplementary material, we have removed mention of it in the Conclusion and placed it in the supplementary material instead.
>
> **W5.** First, we have added empirical results showing that the distribution of the prediction error $\varepsilon$ indeed satisfies the symmetric, zero-mean assumption. Please see our new Appendix B.1.
>
> However, if the prediction error is biased, we anticipate that the final refined path may be slightly biased, but safety is unaffected because the CBF-QP enforces forward invariance regardless of the bias. Moreover, if the error is heavy-tailed, the local strong convexity of $-\log p_\varepsilon$ becomes weaker, which may slow down the contraction rate in the correction phase. Again, this only affects path refinement speed, not safety guarantees. In that case, increasing $\alpha$ to introduce deliberate path distortion against the error (see Table 3, Figure 6, and our answer to the reviewer's 7th comment) might help the prediction error reduction.
>
> [9] Yaron Lipman, Ricky T. Q. Chen, Heli Ben-Hamu, Maximilian Nickel, and Matthew Le. Flow matching for generative modeling. In The Eleventh International Conference on Learning Representations, 2023.

---

> ### Author Response · Authors · 2025-11-24
> **Response3 to Reviewer gaEu**
>
> **W6.** First of all, to avoid confusion, we have renamed “Naive Safe FM’’ to “SafeFM,’’ which is SafeFlowMatcher without the prediction phase. Our PC integrator makes corrections from a smooth nominal path near $\tau_1$, where neighboring waypoints differ minimally—meaning that $\|\tau_t^k - \tau_t^{k-1}\|$ is small—leading to gradual CBF inputs and consistent corrections $u$ that preserve small differences and avoid divergence. If, like SafeFM or SafeDiffuser, the correction phase were to be applied from the beginning, it would essentially be correcting a very noisy path, which increases the chances of local traps occurring. Thus, at least one preliminary prediction step is necessary to remove some of the initial noise and confer robustness to local traps. We have added this more in-depth discussion to Appendix F.5.
>
> **W7.** Yes, in the original manuscript, we selected $\alpha = 2$ based on empirical validation, but we do not theoretically show that this value is optimal. However, Lemma 2 and Lemma 3 do provide partial intuition: the vanishing time-scaled dynamics contracts the prediction error at a rate proportional to $e^{-\alpha t}$, while the $(1 - t)$ factor suppresses instability near $t = 1$. Thus, one can infer that $\alpha$ controls a trade-off between faster early contraction (larger $\alpha$) and numerical stability (smaller $\alpha$).
>
> We can observe this effect in our revised empirical studies, expanded by our new Figure 6 and updated Table 3. Increasing $\alpha$ consistently reduces both curvature and acceleration, indicating that larger $\alpha$ suppresses the prediction error more aggressively; this trend is consistent with Lemma 3. On the other hand, a larger $\alpha$ introduces bias in the generated path, where the tail end (around $t = 1$) of the path becomes increasingly (and sharply) distorted.
> In summary, as the scaling constant $\alpha$ increases, the path becomes smoother but can be distorted. Thus, we can find the appropriate $\alpha$ by starting from $\alpha = 1$ and increasing it until we identify the point just before the sharp distortion begins.
>
> **W8.** Thank you for raising this point; we agree that a broader evaluation on higher-dimensional, more constrained settings is important. Thus, we have added additional experimental results for manipulation of a 7-DoF Kuka robot arm performing a block-stacking task under joint-limit CBF constraints and object observations (Section 4.3). SafeFlowMatcher attains the highest score among safety-aware baselines (SafeDiffuser, SafeFM) while maintaining BS ≥ 0, indicating that our PC integrator scales to higher-dimensional state/action spaces and more intricate safety constraints.
>
> Towards more complex safety sets and multiple constraints, we added a “narrow-corridor’’ Maze2D variant with six CBF constraints of varying sizes and shapes in Appendix F.1. SafeFlowMatcher maintains a 0% trap rate.

---

### Official Review · Reviewer_rRLo · 2025-10-30

**Soundness:** 3
**Presentation:** 3
**Contribution:** 3
**Rating:** 8
**Confidence:** 3

**Summary:**

The paper introduces SafeFlowMatcher, a planning framework that integrates conditional flow matching with control barrier functions to generate trajectories that are efficient to sample and provably converge to a robust safe set. The method addresses a key limitation of flow-matching and diffusion planners, which produce high-quality paths but lack safety control and can become trapped when constraints are enforced at each latent step.
SafeFlowMatcher relies on a two-phase prediction-correction scheme. The prediction phase runs standard FM dynamics to propose an unconstrained path, while the correction phase refines it using a vanishing time-scaled FM field and a CBF-based quadratic program that projects waypoints back into the safe set, ensuring finite-time convergence under the derived barrier conditions.
On the theoretical side, the authors restate finite-time CBF results and provide a flow-invariance certificate guaranteeing safety for the corrected flow. Empirically, the framework is validated on Maze2D and MuJoCo locomotion. It consistently outperforms FM, DDIM, and SafeDiffuser-based baselines on the Maze2D environment, achieving higher scores while ablations confirm the role of each phase and the vanishing-scale mechanism.

**Strengths:**

- The paper identifies a genuine mismatch in existing “safe” generative planners: safety is typically enforced at intermediate latent steps of the denoising or flow-matching process, although only the final trajectory is executed. This “semantic misalignment” is addressed cleanly by decoupling generation and certification: safety is enforced only in the correction phase of the flow. This clarification is timely and relevant for the diffusion/FM-for-planning community.
- The prediction-correction (PC) scheme is well motivated. The prediction phase runs unconstrained FM dynamics to avoid distributional drift, while the correction phase refines the path using a vanishing time-scaled FM vector field and a one-constraint CBF-QP projecting waypoints into the safe set. The finite-time safety guarantee applies to this corrected flow. The ablation in section 4.2 (prediction-only / correction-only / full PC) clearly supports this design.
- The QP correction is computationally light: the authors define a convex problem with a single CBF constraint and closed-form projection, countering concerns about runtime overhead. The running time reported in the experiments shows that this approach is quite fast compared to other SOTA methods.
- The authors performed a serious experimental analysis and comparisons by comparing their approach to a wide variety of baselines (for which they have themselves reproduced the results). Experiments are coherent with the claims and SafeFlowMatcher always achieves a postivie barrier function with zero trap rate, outperforming all baselines while remaining efficient.

**Weaknesses:**

- On p.3, the authors introduce CBFs in their general control-affine form: $\dot x = f(x)+g(x)u$, which normally presupposes known dynamics; in experiments, they seem not to use a physical model, but instead plug in analytic obstacle functions. This distinction is not clearly stated.
- Definition 1 and Lemma 1 are written for general systems with Lipschitz $f,g$, but in practice, the “dynamics” is the FM vector field with vanishing scaling (cf. equation 10). The paper does state this in section 3.2, but the transition is quick. I think the authors should give a short explanation with intuitions; the CBF is not certified in the real environment, but on the generated flow. Furthermore, Definitions 1 and Lemma 1 should be more intuitively justified; currently, they are just "dropped" there without a clear explanation, which makes it difficult for the reader to understand their consequences and intuitive meanings.
- I noticed some clarity issues: on page 3, line 108, some distribution over paths $q$ appears, but has not been introduced anywhere. I also found that, in the main text, the environments and the assumptions made on those environments were not well-specified. For instance, what is the state space and what are the dynamics of the $2$D maze? Does the method assume access to the exact dynamics or just to a dataset containing random trajectories? If this is the case, from where do they come? A short description of the environments and assumptions would help.

**Questions:**

- Could you make explicit what you actually assume in the experiments? Am I correct that you do not know the environment transition dynamics and that the only environment-specific information you use is offline trajectories for FM training and analytic, differentiable obstacle/safety functions $b(x)$ with known centers and scales? Is there any case where you used the true simulator for $\nabla b$?
- In the maze experiments, are you conditioning FM on start-goal or only on start and letting it flow to the high-density goal region? Could you clarify what is actually fed to the network at inference time?

---

> ### Author Response · Authors · 2025-11-24
> **Response to Reviewer rRLo**
>
> **W1.** The preliminaries Section 2.2 describes the background of CBFs for general control-affine systems. The general-form dynamics (5) corresponds to the VTFD (14), the flow-matching dynamics with our correction perturbation. In our manuscript, we have revised Sections 2.2 and 3.2 to highlight this relationship more clearly. Accordingly, the only assumption we require is the availability of a barrier function that bounds the obstacles.
>
> **W2.** We agree that it could be easy for readers to misinterpret the $(f,g)$ as being the system dynamics rather than the VTFD in the correction phase of SafeFlowMatcher. We have revised Sections 2.2 and 3.2 to highlight this relationship more clearly. Definition 1 and Lemma 1 follow directly from CBF literature, but we do agree that they could have been more intuitively presented. In our revisions, we have added some brief transition sentences between these statements, and deferred further details and background to Appendix A to respect the page constraints.
>
> **W3.** In our setting, $q$ denotes the empirical data distribution over target paths, i.e., the dataset of expert demonstrations used for conditional flow matching. We agree that this definition was not explicitly introduced in the main text, and we revised the manuscript to clearly define $q$ at its first appearance.
>
> Regarding the reviewer’s question about the environment and the assumptions made on it, SafeFlowMatcher does *not* assume access to system dynamics or a transition model. All our current experimental domains (Maze2D, locomotion, and robot manipulation) follow the offline path-generation setting used in Diffuser and SafeDiffuser, where one learns from a fixed dataset of demonstration paths without access to a transition model or task reward function at test time. We added the description of the environment setup on Maze2D, locomotion, and manipulation tasks in Appendix E.1.
>
> **Q1.** Yes, the reviewer's understanding is correct. In all experiments, we do *not* assume access to the environment transition dynamics. SafeFlowMatcher operates purely in the offline setting: the only environment-specific information we use is (i) demonstration paths for training the flow matching model, and (ii) closed-form, differentiable obstacle/safety functions $b(x)$ with known centers and scales (e.g., geometric obstacles in Maze2D and the height-based constraint in locomotion). The $b$ functions are user-designed: in all our experiments, they admit closed-form expressions, and their gradients $\nabla b(x)$ are computed explicitly without any use of the true simulator. Alternatively, the CBFs can be constructed using data-driven methods; see our revised Appendix A for a survey. We explicitly stated these assumptions in the revised manuscript.
>
> **Q2.** Thank you for catching this ambiguity on the conditioning scheme. In Maze2D, the planner is conditioned on the start and goal state observations. For locomotion tasks, the planner conditions on the current state observation at every environment step and uses the task score as a guidance signal to encourage forward progress. For robot manipulation (block stacking), the context includes the initial robot joint configuration together with the observed states of the four blocks. We have now clarified the exact conditioning for all tasks (Maze2D, locomotion, and Kuka block stacking) in Appendix E.1.

---

### Official Review · Reviewer_ZPuJ · 2025-11-07

**Soundness:** 3
**Presentation:** 2
**Contribution:** 2
**Rating:** 4
**Confidence:** 3

**Summary:**

The paper proposes SafeFlowMatcher, a planning framework that aims to combine Flow Matching (FM) generative models with the certified safety of Control Barrier Functions (CBFs). The core contribution is a two-phase "Prediction-Correction" (PC) integrator. First, a "Prediction" phase uses the learned FM model to generate a candidate path in one or a few steps, without safety constraints. Second, a "Correction" phase refines this path using a proposed "vanishing time-scaled flow dynamics" (VTFD) and a CBF-based quadratic program (QP) to enforce safety and finite-time convergence to the safe set. The authors claim this two-phase approach decouples generation from certification, thereby avoiding the "local trap" problem. The method is evaluated on 2D maze navigation and locomotion tasks, showing improved safety and task performance and a zero "trap rate" compared to baselines like SafeDiffuser.

**Strengths:**

1. The paper identifies a clear and important problem in safe generative planning. Applying hard safety constraints (like CBFs) directly within the sampling loop of a generative model (like diffusion or flow matching) can distort the learned dynamics, leading to "distributional drift" and "local trap" failures. The goal of decoupling generation from certification is well-motivated.

2. To further enhance sampling stability in the correction phase, this paper proposes a time-scaled flow dynamics method to contract the prediction errors and mitigate drifting.

3. The paper compares the proposed method with multiple baselines and conducts various ablation studies to demonstrate the efficiency and effectiveness of SafeFlowMatcher.

**Weaknesses:**

1. The theoretical guarantees for the correction phase (Lemma 2 and 3) rest on strong assumptions about the prediction error $\epsilon$ (e.g., symmetric, zero-mean, and a locally strongly convex negative log-density). There is no empirical validation to show that the actual integration error from a deep FM model satisfies these convenient properties.



2. In the experiments (Table 1), SafeFlowMatcher achieves zero trap rate. How does the proposed method guarantee the low trap rate? Can the method consistently achieve a zero trap rate in some higher-dimensional or more complex navigation tasks? Are there any theoretical guarantees?



3. The current evaluation only includes three simple environments: one maze and two locomotion environments. Can the author provide experimental results on robot manipulation tasks as well? For example, the environment used in SafeDiffuser might be a good starting point.



4. In Algorithm 1, SafeFlowMatcher first samples a trajectory and solves QP in a double for-loop. The authors report a very fast sampling time of ~4.7ms. To better understand the efficiency of the method, it would be more informative to report the computation time of both phases.



5. Does the QP always have a solution? What will Algorithm 1 do if the QP can not be solved? Will the safety guarantee still be satisfied in that case?



6. Does the method always require a specific function $b$ as in equation (6)? How to obtain $b$ for environments with irregular geometry, agents with high-dimensional state space, or robots with image-based observation?



7. In line 214, "that places $\tau_0^p$ sufficiently close to ...", should the term here be $\tau_1^p$?

**Questions:**

See the weakness above.

---

> ### Author Response · Authors · 2025-11-24
> **Response1 to Reviewer ZPuJ**
>
> **W1.** The theoretical guarantees rely on properties of the prediction error $\epsilon$ (e.g., symmetry, zero-mean, and local strong convexity of the negative log-density).  Following the reviewer's comment, we have added an empirical validation of these assumptions in Appendix B.1 of the revised manuscript. Specifically, we measure the prediction error generated by the flow matching model across multiple prediction horizons. The new results in Appendix B.1 show that the error distribution is centered near zero, approximately symmetric, and becomes increasingly concentrated around zero especially as the prediction horizon increases. We believe that these observations provide empirical support for the assumptions used in Lemmas 2 and 3 and justify applying the vanishing time-scaled dynamics analysis to the correction phase.
> In our revisions, we explicitly referenced this empirical study when stating the assumptions in Section 3.1.
>
> **W2.** First, we would like to clarify that we do not claim a formal theorem guaranteeing a zero trap rate.
> The relaxation term can indeed help reduce local traps, as also observed in SafeDiffuser, but this is not the sole reason for our improvements. Also, it is not that effective, as shown in Table 1.
>
> SafeDiffuser applies CBF constraints to every waypoint at every sampling step starting from i.i.d. noise, so neighboring waypoints can differ significantly. Thus, each waypoint is updated by very different CBF corrections, which breaks path continuity (i.e., $\|\tau_t^k - \tau_t^{k-1}\| \leq \zeta$) and creates local traps. In contrast, SafeFlowMatcher starts the correction phase from a semi-continuous predicted path (i.e., $\|\tau_0^{c,k} - \tau_0^{c,k-1}\| \leq \eta$ for some small $\eta \geq \zeta$), so neighboring waypoints are already close and the CBF corrections vary smoothly along the path. Thanks to these two separate phases, the path can preserve continuity and empirically leads to zero trap rate. Detailed description and visualization are added in the new Appendix F.5.
>
> To provide a stronger empirical study of local traps, we additionally included the six-constraint case in Maze2D for a more complex scenario, as well as a robot manipulation task.
> The corresponding results are provided in Appendix F.1 and Section 4.3, respectively.
> Empirically, this design yields an almost zero trap rate across all environments.
> Furthermore, we added a more detailed explanation of how the PC integrator mitigates traps in Appendix F.5.
>
> **W3.** As mentioned in response to the second answer, we have added the experimental results for the robot manipulation task (also used in SafeDiffuser) in Section 4.3, and attached a simple visualization in Figure 12 illustrating the additional artifacts that arise in relation to the extension of the PC integrator to high-dimensional spaces in robot manipulation, as well as their resolution.
>
> **W4.** To provide further insight into the efficiency of SafeFlowMatcher, we have updated Section 4.1 with results on FlowMatcher (without safety corrections, hence no QP computations).
> Specifically, in the Maze2D experiment, model inference only (FlowMatcher) takes 3.51 ms, computing the closed-form solution takes 1.14 ms, and computing by the QP solver takes 33.21 ms. We reported this in the revised manuscript.
>
> **W5.** The barrier constraint in Eq.(15) includes a slack variable $r_t^k \ge 0$ together with a positive weight $w_t^k > 0$ for all $t < t_w$. Therefore, for $t < t_w$ the constraint is always feasible because any violation can be absorbed by the slack variable. When $t \ge t_w$, the relaxation term vanishes ($w_t^k = 0$), and the safety constraint reduces to the finite-time barrier condition. According to Proposition 1, the convergence time satisfies
> $$
> T \le t_w + \frac{(\delta - b(\tau_{t_w}^{c,k}))^{1-\rho}}{\epsilon(1-\rho)}.
> $$
> Therefore, by choosing $(\epsilon, \rho)$ such that the right-hand side is no larger than $1$, we can ensure that the path reaches the robust safe set $C_\delta$. Because the prediction phase brings $\tau_0^c$ (and consequently $\tau_{t_w}^c$) close to the target path, the value $b(\tau_{t_w}^{c,k})$ becomes much larger than when starting directly from noise, meaning that several waypoints are already close to $\delta$.
> However, without the PC integrator, $b(\tau_{0})$ can be much more negative, which shrinks the set of feasible hyperparameter choices.
> Consequently, a wide range of $(\epsilon,\rho)$ values ensures $T \le 1$, unlike methods that do not employ the PC integrator.
> In practice, we choose $(\epsilon,\rho)$ considering the worst-case value of $b$. Moreover, we consistently observe feasibility across all experiments. In the revised manuscript, Appendix F.2 presents additional experiments evaluating the effect of $(\epsilon,\rho)$ on SafeFlowMatcher (with the PC integrator) and SafeFM (without it). These results empirically confirm that SafeFlowMatcher admits a much wider feasible hyperparameter region.

---

> ### Author Response · Authors · 2025-11-24
> **Response2 to Reviewer ZPuJ**
>
> **W6.** We agree that designing CBFs can be challenging in complex, unstructured environments. This is a limitation not only of our method, but of most certification-based safe approaches. However, recent work in the CBF literature has shown that:
> - non-convex and non-smooth safe sets can be handled (e.g., discrete- or duality-based CBFs [1, 2]),
> - perception uncertainty can be incorporated via robust/probabilistic CBFs [3, 4], and
> - learned CBFs (e.g., neural barrier functions from RGB-D images [5], LiDAR [6, 7, 8], or expert demonstrations [9, 10]) can act as safety certificates.
>
> We added and cited the relevant related work on these CBF extensions in Appendix A of the revised manuscript. We acknowledge that applying perception-based or learning-based CBFs to SafeFlowMatcher for real robotic systems is a promising direction of future work.
>
> **W7.** Thank you for pointing out the typo. In Line 214, the term should indeed be $\tau_1^p$, not $\tau_0^p$. During the prediction phase, the integration produces $\tau_1^p = \tau_1^\ast + \varepsilon$, which is the quantity that becomes sufficiently close to the target path and is used as the initialization for the correction phase. We corrected this in the revised version and checked the rest of the paper for similar mistakes.
>
> [1] Ayush Agrawal and Koushil Sreenath. Discrete control barrier functions for safety-critical control of discrete systems with application to bipedal robot navigation. In Robotics: Science and Systems, volume 13, pp. 1–10. Cambridge, MA, USA, 2017.
> [2] Qiayuan Liao, Zhongyu Li, Akshay Thirugnanam, Jun Zeng, and Koushil Sreenath. Walking in narrow spaces: Safety-critical locomotion control for quadrupedal robots with duality-based optimization. In 2023 IEEE/RSJ International Conference on Intelligent Robots and Systems (IROS), pp. 2723–2730. IEEE, 2023.
> [3] Ryan K Cosner, Andrew W Singletary, Andrew J Taylor, Tamas G Molnar, Katherine L Bouman, and Aaron D Ames. Measurement-robust control barrier functions: Certainty in safety with uncertainty in state. In 2021 IEEE/RSJ International Conference on Intelligent Robots and Systems
> (IROS), pp. 6286–6291. IEEE, 2021.
> [4] Kehan Long, Vikas Dhiman, Melvin Leok, Jorge Cort´es, and Nikolay Atanasov. Safe control synthesis with uncertain dynamics and constraints. IEEE Robotics and Automation Letters, 7(3): 7295–7302, 2022.
> [5] Hossein Abdi, Golnaz Raja, and Reza Ghabcheloo. Safe control using vision-based control barrier function (v-cbf). In IEEE International Conference on Robotics and Automation, pp. 782–788. IEEE, 2023.
> [6] Mohit Srinivasan, Amogh Dabholkar, Samuel Coogan, and Patricio A Vela. Synthesis of control barrier functions using a supervised machine learning approach. In 2020 IEEE/RSJ International Conference on Intelligent Robots and Systems (IROS), pp. 7139–7145. IEEE, 2020b.
> [7] Kehan Long, Cheng Qian, Jorge Cort´es, and Nikolay Atanasov. Learning barrier functions with memory for robust safe navigation. IEEE Robotics and Automation Letters, 6(3):4931–4938, 2021.
> [8] Marvin Harms, Mihir Kulkarni, Nikhil Khedekar, Martin Jacquet, and Kostas Alexis. Neural control barrier functions for safe navigation. In 2024 IEEE/RSJ International Conference on Intelligent Robots and Systems (IROS), pp. 10415–10422. IEEE, 2024.
> [9] Alexander Robey, Haimin Hu, Lars Lindemann, Hanwen Zhang, Dimos V Dimarogonas, Stephen Tu, and Nikolai Matni. Learning control barrier functions from expert demonstrations. In 2020 59th IEEE Conference on Decision and Control (CDC), pp. 3717–3724. IEEE, 2020.
> [10] Lars Lindemann, Alexander Robey, Lejun Jiang, Satyajeet Das, Stephen Tu, and Nikolai Matni. Learning robust output control barrier functions from safe expert demonstrations. IEEE Open Journal of Control Systems, 3:158–172, 2024.

---

### Official Review · Reviewer_Jozp · 2025-11-09

**Soundness:** 3
**Presentation:** 3
**Contribution:** 3
**Rating:** 6
**Confidence:** 2

**Summary:**

This paper presents SafeFlowMatcher, a novel framework for safe robotic path planning that
effectively couples generative Flow Matching (FM) with Control Barrier Functions (CBFs). The
authors identify a key problem with existing generative planners: they either lack formal safety
guarantees or, if certification-based, suffer from local trap problems where interventions distort
the generative process and cause the plan to fail.

**Strengths:**

● Paper is very well written and easy to understand
● The key idea of a two-phase Prediction-Correction (PC) integrator is both simple and
highly effective. Decoupling the initial path generation from the subsequent safety
correction elegantly solves the local trap problem
● The paper provides a formal barrier certificate and a proof of finite-time convergence -
strong mathematical guarantees.
● The method is validated on maze navigation and high-dimensional locomotion tasks,
demonstrating superior performance across key metrics: safety, path quality, and
efficiency

**Weaknesses:**

● The paper's baselines are exclusively generative models (diffusion- and FM-based) .
While this is the direct field of contribution, classical sampling- or optimization-based
planners are still the gold standard in many robotics applications. A comparison against a
strong classical baseline (e.g., an optimization-based planner) would have provided a
better picture of the practical performance.

**Questions:**

● How does the framework handle dynamically changing environments? The CBF-QP is
solved at each control step, which seems well-suited for this, but the underlying FM
model is trained to generate paths based on a static map. How would the model perform
if an obstacle changed position mid-rollout?

---

> ### Author Response · Authors · 2025-11-24
> **Response to Reviewer Jozp**
>
> We thank the reviewer for their suggestion regarding comparisons with classical sampling- or optimization-based planners. We agree that such methods could indeed be strong baselines, especially in settings where a dynamics model or a well-specified reward map are available.
>
> Our current experimental domains (Maze2D, locomotion, and robot manipulations) follow the offline path-generation setting used in Diffuser and SafeDiffuser, where one learns from a fixed dataset of paths and does not assume access to a transition model or a task reward at test time.
> For this reason, our main comparison set focused on generative planners.
> The fact that the dynamics and reward map are both unavailable is a problem setup that we now see was not clearly described in our Section 3, and we will revise it accordingly.
>
> Moreover, we are currently working on additional comparisons for the Maze2D environment using PRM and RRT, which only require knowledge of the maze environment itself.
> We uploaded this comment in advance to adhere to the ICLR rebuttal timeline, but will update it and the paper PDF as soon as the experiments are completed.
>
> To answer the reviewer's question, SafeFlowMatcher can indeed be made compatible with dynamically changing environments.
> It is actually one of the main benefits of our prediction-correction decoupling, which ensures that the learned vector field remains agnostic to safety interventions, allowing seamless adaptation to changes in the unsafe (obstacle) set.
> Since both phases require only a few sampling steps, SafeFlowMatcher can be re-run at high frequency; the fact that SafeFlowMatcher can compute substantially faster than other methods is an aspect we highlighted in our revisions via the introduction of the total computation “T-Time’’ performance metric in Section 4.
>
> The fast computation speed of SafeFlowMatcher means that it is more applicable to use in a receding-horizon manner, which is more amenable to dynamic environments. For example, one can update the barrier function $b$ whenever the environment changes, and then replan from the current state.
> We have added additional related works in Appendix A to discuss CBFs for dynamic environments, which can be used as the method to update the barrier function $b$.

---

> > ### Author Response · Authors · 2025-11-29
> > **Updated Response to Reviewer Jozp — Comparison with Classical Planners**
> >
> > We have completed the additional comparison in the Maze2D environment using RRT* and A*, both of which require access to the full occupancy map. The full results are provided in the supplementary file “Classical_Methods_Comparison.pdf”.
> >
> > Briefly, although all methods satisfy the safety constraints, SafeFlowMatcher achieves a substantially higher task score (1.632 vs. 0.874 for RRT* and 0.854 for A*), despite operating *without* any map information. These results support our claim that generative planners such as SafeFlowMatcher can surpass classical planners even under weaker assumptions.
> >
> > Furthermore, unlike generative methods, classical planners cannot leverage the high-level strategies embedded in expert demonstrations, since they rely solely on explicit maps and handcrafted search heuristics. This fundamental difference limits their ability to reproduce expert-like behaviors in data-driven settings.

---

### Author Response · Authors · 2025-12-01
**Summary of Responses to Concerns from the reviewers (1/7)**

For clarity, we aggregate the reviewers’ questions and comments into thematic groups and provide concise responses to each summarized point below. Here, we provide a brief summary of responses. For detailed explanations, please refer to our point-by-point responses and the corresponding locations in the revised manuscript and supplementary materials.

>**Baselines & Experimental Scope (Classical Planners, Control Baselines, Complex Tasks)**
>
>The paper only evaluates on generative baselines (diffusion/flow matching) and relatively simple environments (one Maze2D and two locomotion tasks). Why not:
>
>* include strong classical planning/control baselines such as PRM/RRT, MPC+CBF-CLF-QP, Neural-ODE/CNF planners with a CBF filter, or a diffusion model with single-shot CBF projection; and
>    *sources: (Jozp; W1), (DQsF; W5), (DQsF; Q5)*
>* demonstrate performance on more complex and higher-dimensional robotic tasks (e.g., SafeDiffuser’s manipulation environment.)
>    *sources: (ZPuJ; W3), (gaEu; W8)*

**Response**

**(a) Offline path-generation setting and classical control baselines.**

All environments in our paper—Maze2D, locomotion, and manipulation—follow the **offline path-generation** setting introduced in Diffuser and SafeDiffuser: at test time, the planner has access only to

* a fixed dataset of demonstration paths (for training the flow matching model), and
* closed-form, differentiable functions $b(x)$,

and **does not** have access to a transition model. This is the standard setting for generative planners.

As a consequence, strong control baselines such as **MPC with CBF–CLF–QP** require explicit system dynamics and online rollouts, which are simply unavailable in this setting. For this reason, we did not include MPC-based baselines; we have made this assumption explicit in the revised paper (Section 3 and Appendix E.1).

Similarly, **Neural-ODE / CNF planners with CBF filters** have already been analyzed in prior work (e.g., On the Forward Invariance of Neural ODEs, as cited in SafeDiffuser), which highlights their limitations when combined with CBFs. So, we chose to focus on comparing SafeFlowMatcher against the strongest existing **generative safe planners** (SafeDiffuser, SafeFM/SafeDDIM).

**(b) Additional classical planning baselines in Maze2D.**

Following the reviewers’ suggestions, we **added experiments in Maze2D** comparing against **RRT\*** and **A\***. Both methods assume access to the full occupancy map, whereas SafeFlowMatcher does not use map information at test time. The new results (summarized in the supplementary file Classical\_Methods\_Comparison.pdf) show that:

* all methods satisfy the safety constraints, but
* **SafeFlowMatcher achieves a substantially higher task score** (1.632 vs. 0.874 for RRT* and 0.854 for A*), despite operating without maps.

This supports our claim that generative planners such as SafeFlowMatcher can **outperform classical planners even under weaker information assumptions**, by leveraging high-level strategies encoded in expert demonstration data.

**(c) More complex and higher-dimensional tasks.**

To address the concern about environment complexity, we added:

* a **“narrow-corridor” Maze2D variant** with **six CBF constraints** of varying sizes and shapes (Appendix F.1), and
* a **7-DoF Kuka block-stacking** task under joint-limit CBF constraints and object observations (Section 4.3).

In both settings, SafeFlowMatcher maintains **near-zero trap** rate and attains the **highest task score among safety-aware baselines** (SafeDiffuser, SafeFM), demonstrating that the prediction–correction integrator scales to higher-dimensional spaces and more intricate safety sets.

---

> ### Author Response · Authors · 2025-12-01
> **Summary of Responses to Concerns from the reviewers (2/7)**
>
> > **CBF Design, Barrier Function $b(x)$, and Assumed Environment Knowledge**
> >
> >The method seems to rely on closed-form, well-calibrated, differentiable barrier functions $b(x)$, while the CBF preliminaries are written in the general control-affine form $\dot{x}=f(x)+g(x)u$.
> >
> >* Exactly what environment information do you assume in experiments (transition dynamics known or not, obstacle shapes, closed-form $b$ and $\nabla b$?
> >
> >    *sources: (rRLo; W1), (rRLo; Q1)*
> >* Do you ever use the true simulator to compute $\nabla b$?
> >
> >    *sources: (rRLo; Q1)*
> >* How realistic is it to assume a known, smooth, correctly calibrated $b(x)$ in real system with perception noise, contacts, noncovex geometry, high-dimensional state spaces, or image-based observations?
> >
> >    *sources: (DQsF; W1)*
> >* How could $b$ be designed or learned in complex, dynamic or unstructured environments, and how would such learned/approximate CBFs fit into SafeFlowMatcher?
> >
> >    *sources: (ZPuJ; W6), (gaEu; W1)*
>
> **Response**
>
> **(a) Assumptions in our experiments.**
> In all experiments, SafeFlowMatcher operates without access to transition dynamics or rewards at test time. The environment assumptions are:
>
> * a dataset of expert demonstrations for conditional flow matching, and
> * simple, closed-form differentiable CBFs $b(x)$ and their gradients $\nabla b(x)$,
>
> We never use the true simulator to differentiate dynamics; gradients are computed analytically from the closed-form expressions of $b(x)$.
>
> We have clarified these assumptions in Section  and the environment setups for Maze2D, locomotion, and Kuka block stacking in Appendix E.1.
>
> **(b) Realistic CBFs in complex, uncertain environments.**
>
> We fully agree that handcrafting a smooth, perfectly calibrated barrier function is challenging in real systems with perception noise, contacts, nonconvex geometry, or high-dimensional observation spaces. This is not a limitation unique to SafeFlowMatcher but a **general challenge for all CBF-based safety filters.**
>
> Importantly, recent CBF literature shows that **perfect, hand-engineered barriers are not strictly necessary:**
>
> * **Nonconvex and nonsmooth safe sets:** discrete-time and duality-based CBFs can handle complex obstacle geometries.
> * **Perception uncertainty:** measurement-robust and probabilistic CBFs address state-estimation noise and uncertain constraints.
> * **Learned CBFs:** neural barrier functions can be learned from RGB-D images, LiDAR, or expert demonstrations.
>
> We have added a short survey of these methods in Appendix A.
>
> **(c) Compatibility with SafeFlowMatcher.**
> SafeFlowMatcher uses the CBF only as a constraint inside the correction QP. As long as we can evaluate $b(x)$ and its gradient (or subgradient) at the waypoints to be corrected, we can plug in robust/measurement-robust CBFs, probabilistic or chance-constrained CBFs, or learned CBFs from perception.
> Thus, SafeFlowMatcher is structurally compatible with these more realistic CBF formulations. Extending our experiments to fully perception-based or learned barriers is an exciting direction for future work; in this paper we focus on analytic CBFs for clarity of exposition.

---

> ### Author Response · Authors · 2025-12-01
> **Summary of Responses to Concerns from the reviewers (3/7)**
>
> > **Robustness to Uncertainty, Mismatch, and Dynamic Environments**
> >
> > How robust is SafeFlowMatcher’s safety guarantee in realistic setting with:
> >
> > * dynamically changing environments, such as moving or time-varying safety constraints?
> >
> >     *sources: (Jozp; Q1), (DQsF; W3), (DQsF; Q6)*
> > * state-estimation noise, delays, and dynamic mismatch?
> >
> >     *sources: (DQsF; W3), (DQsF; Q6)*
> Do you have ISS-type or chance-constrained analyses, or experiments that bound violation probabilities under such uncertainties?
> *sources: (DQsF; W3), (DQsF; Q6)*
>
> **Response**
>
> **(a) Dynamically changing environments and moving constraints.**
>
> SafeFlowMatcher’s prediction–correction integrator is particularly well-suited to dynamic environments. Because SafeFlowMatcher requires only a few inference steps, SafeFlowMatcher can be used in a receding-horizon manner: whenever obstacle positions or safety constraints change, we update $b(x,t)$ and replan from the current state. Also, we have added references on dynamic CBFs used with moving obstacles in Appendix A.
>
> **(b) State-estimation noise, dynamics mismatch, and delays.**
>
> Our current theoretical analysis is derived in an idealized deterministic setting — exact state information, static safety sets, and no model mismatch or delays.
>
> **(c) ISS-type or chance-constrained analyses.**
> We do not provide ISS-type or chance-constrained bounds in this work. However, the use of a **robust safe set** $\mathcal{C}_\delta$ already introduces a safety margin that can buffer small perturbations in practice.
>
> A rigorous treatment of estimation noise, dynamics mismatch, and time-varying constraints would require combining SafeFlowMatcher with **stochastic/robust CBF formulations and explicit modeling of uncertainty,** which we view as an important direction for follow-up work.
>
> > **Theoretical Assumptions on Prediction Error $\epsilon$**
> >
> > Lemma 2 and 3 rely on strong assumptions about the prediction error $\epsilon$: symmetric, zero-mean, with locally strong
>  convex $-\log p_\epsilon$.
> >
> > * Is there empirical evidence that FM’s actual integration error satisfies this?
> >
> >     *sources: (ZPuJ; W1)*
> > * How robust is the frame work when prediction errors are biased, skewed, or heavy-tailed?
> >
> >     *sources: (gaEu; W5)*
>
> **Response**
>
> **(a) Empirical validation of the error model.**
>
> Motivated by the reviewers’ comments, we added an empirical study of the **FM prediction error** in Appendix B.1. Concretely, we:
>
> * collect prediction errors from the FM model over multiple prediction horizons,
> * plot their empirical histograms and summary statistics.
>
> The results show that:
>
> * the error distribution is centered **near zero**,
> * is approximately **symmetric**, and
> * becomes increasingly **concentrated around zero** as the prediction horizon increases.
>
> These observations support modeling the error as approximately symmetric and zero-mean with a locally convex negative log-density, providing empirical justification for the assumptions used in Lemmas 2 and 3.
>
> **(b) Robustness to biased or heavy-tailed errors.**
> Even when these assumptions are violated, SafeFlowMatcher remains robust in the following sense:
>
> * If the prediction error is **biased**, the final refined path can exhibit a small bias relative to the nominal FM path, but **safety is unaffected**: the CBF-QP enforces forward invariance of the robust safe set regardless of the bias.
> * If the error is **heavy-tailed**, the local strong convexity of $-\log p_\epsilon$ weakens, which can **slow down the contraction rate** in the correction phase. This again affects only the **speed and quality of refinement**, not the safety guarantee.
>
> In such cases, increasing the vanishing scaling factor (denoted $\alpha$ in our paper) can deliberately introduce stronger correction against the prediction error at the expense of some path distortion, as seen in our updated Table 3 and Figure 6. We have clarified these trade-offs in the revised text.

---

> ### Author Response · Authors · 2025-12-01
> **Summary of Responses to Concerns from the reviewers (4/7)**
>
> > **Trap Rate and Distributional Drift, SafeFM vs. SafeFlowMatcher**
> >
> > Experiments report zero trap rate for SafeFlowMatcher and high trap rates for SafeFM/ SafeDiffuser.
> >
> > * What mechanism explains  or guarantees the low trap rate, especially in higher-dimensional or complex tasks?
> >
> >     *sources: (ZPuJ; W2)*
> > * Why does imposing safety from the very beginning (SafeFM/SafeDiffuser) lead to many local traps?
> >
> >     *sources: (gaEu; W6)*
> > * Does repeated CBF correction induce distributional drift, and can this be quantified (energy distance, path FID, etc.)?
> >
> >     *sources: (DQsF; W4), (DQsF; Q4)*
>
> **Response**
>
> **(a) Mechanism behind low trap rate.**
>
> We emphasize that we do not claim a formal theorem guaranteeing zero trap rate. Rather, the low trap rate arises from the structure of the prediction–correction integrator:
>
> * **SafeFM / SafeDiffuser**: start from i.i.d. noise and enforce CBF constraints at every sampling step and every waypoint. Neighboring waypoints can be far apart, and each waypoint may receive very different CBF corrections. This breaks path continuity and can cause local trap near safety boundaries.
> * **SafeFlowMatcher**: first performs a **prediction phase** that produces a **semi-continuous nominal path** (neighboring waypoints are already close). The **correction phase** then applies small CBF adjustments that depend continuously on the waypoint along the path. As a result, path continuity is preserved and local-trap phenomena are dramatically reduced.
>
> We have added a detailed explanation and visualization of this mechanism in Appendix F.5.
>
> To test whether this behavior persists in more complex settings, we added:
>
> * a Maze2D environment with **six CBF constraints** (Appendix F.1), and
> * a **Kuka block-stacking manipulation task** (Section 4.3).
>
> Across all these environments, SafeFlowMatcher exhibits **essentially zero trap rate**, while SafeFM and SafeDiffuser continue to suffer from local traps. This suggests that the PC integrator is effective at mitigating traps beyond simple 2D mazes.
>
> **(b) Distributional drift induced by CBF corrections.**
>
> We agree that repeated corrections could, in principle, introduce **distributional drift** away from the nominal flow and reduce path diversity. To quantify this effect, we added a new analysis based on **energy distance** in Appendix F.4 (Fig. 15).
>
> For each pair of models (FlowMatcher vs. SafeFlowMatcher, FM vs. SafeFM, Diffuser vs. SafeDiffuser), we generate 100 matched rollouts and compute the path-level energy distance:
>
> * SafeFlowMatcher: **0.061**
> * SafeFM: **0.097**
> * SafeDiffuser: **0.229**
>
> SafeFlowMatcher introduces the **smallest drift**, due to:
>
> 1. the prediction–correction integrator, which avoids perturbing intermediate latent states, and
> 2. the vanishing time-scaled dynamics, which confines significant corrections to small, safety-critical segments.
>
> Figure 15 further shows that SafeFlowMatcher’s per-waypoint drift is localized around constraint-violating portions of the path, whereas SafeFM and SafeDiffuser exhibit drift over much larger portions of the path.

---

> ### Author Response · Authors · 2025-12-01
> **Summary of Responses to Concerns from the reviewers (5/7)**
>
> > **Vanishing Time-Scaling & Hyperparameters Sensitivity $\epsilon, \rho, \delta, \alpha, t_w$**
> >
> > SafeFlowMatcher introduces several hyperparameters in correction dynamics and vanishing time-scaling.
> >
> > * How sensitive are performance and safety to these choices?
> >
> >     *sources: (gaEu; W2), (DQsF; Q3)*
> > * Are there principled heuristics or adaptive strategies to tune them?
> >
> >     *sources: (gaEu; W2)*
> > * Why choose $\alpha=2$, and what is the meaningful range of $\alpha$ that balances fast contraction and stability?
> >
> >     *sources: (gaEu; W7), (DQsF; Q3)*
>
> **Response**
>
> We denote here the robust margin ($\delta$), finite-time CBF parameters ($\rho, \varepsilon$), vanishing scale ($\alpha$), and the time $t_w$ at which slack is switched off.
>
> **(a) Overall robustness due to prediction–correction.**
>
> The **prediction phase** brings the nominal path close to the target distribution, so the correction phase operates on **small residual errors**, which makes SafeFlowMatcher **less sensitive** to hyperparameter choices than methods that correct from pure noise. This is theoretically valid according to Proposition 1, and we described this Remark 1 and Appendix F.2.
>
> We studied sensitivity in detail via ablations reported in Appendix F.2. For example, in Maze2D we sweep $\rho \in $ {0.1, 0.3, 0.5, 0.7, 0.9}, and $\varepsilon \in $ {0.25, 0.50, 1.00, 2.50, 5.00, 10.00}.
>
> Across this grid, SafeFlowMatcher maintains high scores and safety, revealing a **broad feasible region** for the hyperparameters, whereas SafeFM is much more fragile. We have added these results and discussed them in Appendix F.2.
>
> **(b) Role of individual hyperparameters.**
>
> * **Robust margin $\delta$**: sets the thickness of the robust safe set $\mathcal{C}_\delta$. It is problem-specific; larger $\delta$ trades off some path optimality for additional safety margin.
> * **Finite-time CBF parameters ($\rho, \varepsilon$)**: control the speed of convergence to the safe set. Smaller $\rho$ or larger $\varepsilon$ induce more aggressive safety corrections, which can help fast convergence to a safe set but may also increase curvature. When excessively strong, these corrections can even lead to unstable or oscillatory behavior.
> * **Time $t_w$** determines when the relaxation (slack) is turned off; a larger $t_w$ allows more gradual convergence, but we require $t_w < 1$. Empirically, while relaxation can help improve path quality (e.g., score, curvature, and acceleration) and reduce local traps, SafeFlowMatcher can also operate without relaxation and still outperforms other methods. To show this more clearly, we have added SafeFlowMatcher without relaxation to Table 1.
>
> **(c) Vanishing time-scaling constant $\alpha$.**
>
> We clarify that $\alpha$ in our paper is a constant scaling factor in the vanishing time-scaled flow dynamics, not a class-$\mathcal{K}$ function.
>
> Lemma 2 and Lemma 3 provide intuition:
>
> * increasing $\alpha$ **accelerates contraction** of the prediction error, while
> * the vanishing factor $(1-t)$ **stabilizes behavior near $t=1$**.
>
> In the revised manuscript, we expanded Table 3 and added Figure 6 to show the empirical effect of $\alpha$. We observe that:
>
> * larger $\alpha$ systematically **reduces curvature and acceleration**, producing smoother paths, but
> * excessively large $\alpha$ introduce **distortion**.
>
> In practice, we found that SafeFlowMatcher is robust over a broad range of $\alpha$ values, and $\alpha=2$ works well across all reported tasks. Our tuning heuristic is:
>
> 1. start from $\alpha = 1$,
> 2. gradually increase $\alpha$,
> 3. stop just before sharp distortions appear near $t \approx 1$ (as visible in Fig. 6).
>
> We describe this guideline in the revised Section 4.2 (“Effect of Vanishing Time-Scale”).

---

> ### Author Response · Authors · 2025-12-01
> **Summary of Responses to Concerns from the reviewers (6/7)**
>
> > **QP Feasibility, Slack, and Computational Efficiency**
> >
> > The correction phase solves a CBF-QP per waypoint.
> >
> > * What are the computation costs of prediction vs. correction?
> >
> >     *sources: (ZPuJ; W4)*
> > * How does QP overhead scale with number of waypoints $H$ and with more complex constraints?
> >
> >     *sources: (gaEu; W3), (DQsF; Q1)*
> > * Have you seen QP infeasibility, and under what conditions is feasibility guaranteed?
> >
> >     *sources: (ZPuJ; W5), (DQsF; W2), (DQsF; Q2)*
>
> **Response**
>
> **(a) Computational costs of prediction vs. correction.**
> To make the efficiency of SafeFlowMatcher more transparent, we have updated Section 4.1. In the Maze2D experiment:
>
> * **FlowMatcher inference only (prediction phase)** takes **3.51 ms** on average,
> * **computing the closed-form CBF solution** (when applicable) takes **1.14 ms**, and
> * **solving the CBF-QP** (general case) takes **33.21 ms** on average.
>
> Thus, the dominant overhead comes from the QP in the correction phase, but the overall runtime remains in the tens-of-milliseconds range, since we perform only a single prediction–correction pass rather than many denoising steps as in diffusion-based planners.
>
> **(b) QP overhead scaling with the number of waypoints $H$ or constraints.**
>
> QP correction inevitably introduces computational overhead; this is a known limitation of safety-filter–based generative planners (e.g., SafeDiffuser). SafeFlowMatcher is designed specifically to mitigate this via the prediction–correction integrator:
>
>
> * The **prediction phase** already produces a path close to the target in a few steps (even in one step), so
> * the **correction phase** only needs to compensate a small prediction error (see Fig. 4, where SafeFlowMatcher attains high performance even with a short sampling horizon $T$).
>
> As a result, the QP in Eq. (17) is small across all our experiments, and we did not observe it becoming a runtime bottleneck.
> As with many path-planning algorithms, increasing $H$ scales the cost roughly proportionally, and adding more CBF constraints increases the QP size.
>
> In Appendix F.1, we also report results on a Maze2D variant with six CBF constraints; as expected, the computation time increases but remains manageable and can be offset by shortening the correction horizon.
>
> **(c) QP feasibility and the role of slack.**
> The constraint in Eq.(15) contains a nonnegative slack variable $r_t^k \ge 0$ with a strictly positive weight $w_t^k > 0$ for all $t < t_w$. Hence, for $t < t_w$ the constraint set is always nonempty, since any violation of the barrier inequality can be absorbed by the slack variable.
>
> For $t \ge t_w$, the relaxation term vanishes ($w_t^k = 0$) and the condition reduces to the finite-time CBF inequality. Proposition 1 yields an upper bound
> $$
> T \le t_w + \frac{(\delta - b(\tau_{t_w}^{c,k}))^{1-\rho}}{\epsilon(1-\rho)}.
> $$
> By selecting $(\rho, \varepsilon)$ so that the right-hand side is at most $1$, we guarantee that the path enters the robust safe set $C_\delta$ within the time interval $[0,1]$. The prediction phase is crucial here: because it already moves $\tau^c_0$ (and thus $\tau^c_{t_w}$) close to the target path, $b(\tau^{c,k}_{t_w})$ is typically close to $\delta$, which significantly enlarges the admissible region of $(\rho, \varepsilon)$. In contrast, methods without the PC integrator have much more negative $b(\tau_0)$, making this condition harder to satisfy.
>
> We further study the sensitivity of $(\rho, \varepsilon)$ in Appendix F.2 by sweeping
> $\rho \in $ {0.1, 0.3, 0.5, 0.7, 0.9} and $\varepsilon \in$ {0.25, 0.50, 1.00, 2.50, 5.00, 10.00}.
> These experiments show that the PC integrator admits a **broad feasible region** for $(\rho, \varepsilon)$, which makes hyperparameter selection easier in practice.
>
> Empirically, across all our experiments, we **did not observe QP infeasibility** with these settings, so no additional recovery mechanism (e.g., backtracking or horizon extension) was required in this work.

---

> ### Author Response · Authors · 2025-12-01
> **Summary of Responses to Concerns from the reviewers (7/7)**
>
> > **Clarity on Notation, Conditioning, Environment Description**
> >
> > Some parts of the exposition are unclear:
> >
> > * The generic CBF definitions (Def. 1, Lemma 1) are not clearly connected to the actual vanishing-scaled FM dynamics.
> >
> >     *sources: (rRLo; W2)*
> > * Distribution $q$, environment state spaces, and dataset assumptions require clearer introduction.
> >
> >     *sources: (rRLo; W3)*
> > * In Maze2D, what exactly is given as conditioning (start only vs start+goal)?
> >
> >     *sources: (rRLo; Q2)*
> > * There is a typo: $\tau_0^p$ should $\tau_1^p$.
> >
> >     *sources: (ZPuJ; W7)*
>
> **Response**
>
> **(a) CBF preliminaries vs. vanishing time-scaled FM dynamics.**
>
> Section 2.2 presents the CBF framework for general control-affine systems. In our method, the “dynamics” to which we apply the CBF conditions are not the physical environment dynamics, but the vanishing time-scaled FM vector field used in the correction phase (Eq. (10) in the revised paper).
>
> We have revised Sections 2.2 and 3.2 to:
>
> * explicitly state that the general-form dynamics in Eq. (5) correspond to the VTFD in Eq. (14),
> * clarify that Definitions 1 and Lemma 1 are applied to this vector field, and
> * add brief transition sentences explaining their intuitive meaning, while deferring detailed CBF background to Appendix A due to space constraints.
>
> **(b) Path distribution $q$ and environment/dataset assumptions.**
>
> In our setting, $q$ is the **empirical data distribution**, i.e., the dataset of expert demonstrations used for conditional flow matching. We have now explicitly defined this distribution at its first occurrence in the main text.
>
> We have also expanded Appendix E.1 to describe environment assumptions (Maze2D, locomotion tasks, and a manipulation task)
>
> **(c) Conditioning schemes.**
> We clarify the conditioning used in each task:
>
> * **Maze2D**: the planner is conditioned on both the start and goal state observations.
> * **Locomotion**: the planner conditions on the current state observation at every environment step, and uses the task score as an optional guidance signal to encourage forward progress.
> * **Kuka block stacking**: the context includes the initial robot joint configuration and the observed states of the four blocks.
>
> These details are now summarized in Appendix E.1.
>
> **(d) Typos and minor notation issues.**
> We have corrected all typos pointed out by the reviewers, including:
>
> * replacing an incorrect $\tau_0^p$ with $\tau_1^p$,
> * re-checking the manuscript for similar mistakes.
>
> > **Guidance-Free / Unconditional Version**
> >
> > The paper mentions a possible guidance-free version of SafeFlowMatcher.
> >
> > * What are the challenges in removing guidance?
> >
> >     *sources: (gaEu; W4)*
> > * How would the PC integrator extend to unconditional settings (e.g., unconditional image generation)?
> >
> >     *sources: (gaEu; W4)*
> > * Any initial ideas on how to approach such a version?
> >
> >     *sources: (gaEu; W4)*
>
> **Response**
>
> Our initial submission briefly mentioned a **guidance-free version** of the PC integrator as future work. We now realize, thanks to the reviewers’ comments and our own follow-up experiments, that this direction requires careful treatment.
>
> SafeFlowMatcher as presented in the paper is evaluated in **conditional settings**: the FM model is conditioned at least on the current or initial observation (and sometimes on goal/task information), which provides a strong structural prior and focuses the generative process around relevant modes.
>
> In contrast, in fully **unconditional domains** such as CIFAR-10 image generation, there is:
>
> * no explicit conditioning signal,
> * a highly multimodal target distribution, and
> * a strong need for exploration to cover diverse modes.
>
> In the revised supplementary material (CIFAR10\_Uncond\_Verify.pdf), we report preliminary experiments where we simply attach the PC integrator to an unconditional FM model. This naive approach leads to **severe instability and poor sample quality**, suggesting that:
>
> * strong contraction from the vanishing time-scaled correction can oppose the need for exploration and diversity, and
> * the assumptions on the prediction error may fail in highly multimodal, unconditional settings.

---

### Meta-Review · Area_Chair_csQN · 2026-01-05

**Summary:**

This paper proposes to incorporate safety into flow-based planning algorithms by introducing control barrier functions (CBFs) to correct the planners' predicted trajectories using a fast QP-based algorithm. The authors present many theoretical underpinnings of their approach to justify it along with empirical results on 3 robotics tasks with superior accuracy and speed versus the competing approaches.

Five reviewers provided ratings of (4, 6, 8, 4, 6). Most of the reviewers' concerns were addressed by the author's rebuttal. However, the main criticism of the approach is its reliance on CBFs which are hard to define for complex tasks. However, the authors point to much on-going work on learn CBFs for complex tasks, which can be combined with the current work in the future.

All things considered, the AC feels that the timely novel contribution of the work of coupling CBFs with flow-based path planners is a useful contribution for the research community and hence recommends acceptance. The authors should revise their final manuscript as promised in the rebuttal.

Congratulations!

**Reviewer Concerns:**

Concerns not addressed:
* comparisons to classical planners besides diffusion-based ones
* results on robot manipulation tasks
* Assumptions about assumed zero-mean model for the prediction error

Concerns not addressed:
* reliance on well-defined CBFs, which are challenging to define in dynamic, or unstructured real-world robotic environments.

**Reviewer Scores:**

1. Reviewer Jozp (Rating: 6: marginally above the acceptance threshold. But would not mind if paper is rejected)

This reviewer's primary concern was comparisons to optimization-based planner as well besides diffusion-based planners. The authors provided numerical comparisons showing that their method performs better than these baselines, which addresses the reviewer's concern.

2. Reviewer ZPuJ (Rating: 4: marginally below the acceptance threshold. But would not mind if paper is accepted)

The reviewers's concerns were that (a) Lemma 2 and 3 make strong assumptions, which are not verified, (b) can SafeFlow achieve zero trap rates in more complex robot manipulation tasks, (c) additional results on robot manipulation tasks, (d) providing additional timing information, (e) questions regards QP always being solvable and (f) how CBFs can used for complex environments. While most the reviewer's concerns were addressed the authors acknowledged that challenges of using CBFs in complex, unstructured environments. This point, limits the utility of the proposed approach in practical settings.

3. Reviewer rRLo (Rating: 8: accept, good paper (poster))

This reviewer mostly had clarification questions and suggestions for improving the presentation of the paper based on them. The author's adequately addressed these concerns by appropriately revising the paper.

4. Reviewer gaEu (Rating: 4: marginally below the acceptance threshold. But would not mind if paper is accepted)

This reviewer also raised concerns about (a) the reliance of CBF that are difficult to define in complex settings, (b) had questions about how hyper parameters should be chosen for new tasks, (c) how does QP scale with longer horizons or fail, (d) questions about the proposed guidance-free version as future work, (e) questions about the zero mean prediction error assumptions for Lemma 1 and 2. Most of these reviewers' comments were addressed by the author's rebuttal.

5. Reviewer DQsF (Rating: 6: marginally above the acceptance threshold. But would not mind if paper is rejected)

The reviewer raised many concerns about the novelty and methodology initially. However, they were all addressed by the authors' rebuttal and the reviewer gave a final positive rating.

---

### Decision · Program_Chairs · 2026-01-26

Accept (Poster)